# The Catastrophic Failure of *the* $k$-Means Algorithm in High Dimensions, and How Hartigan's Algorithm Avoids It

Roy R. Lederman [1]  David Silva-Sánchez [2]  Ziling Chen [1]  Gilles Mordant [2]  Amnon Balanov [3]  Tamir Bendory [3]

## Abstract

Lloyd's $k$-means algorithm is one of the most widely used clustering methods. We prove that in high-dimensional, high-noise settings, the algorithm exhibits catastrophic failure: with high probability, essentially every partition of the data is a fixed point. Consequently, Lloyd's algorithm simply returns its initial partition — even when the underlying clusters are trivially recoverable by other methods. In contrast, we prove that Hartigan's $k$-means algorithm does not exhibit this pathology. Our results show the stark difference between these algorithms and offer a theoretical explanation for the empirical difficulties often observed with $k$-means in high dimensions.

## 1. Introduction

Clustering is a core problem in statistics and machine learning. One of the most common formulations of this problem is $k$-means (MacQueen, 1967), which aims to minimize intra-cluster variance; see Bock (2008) for a historical account. Solutions to the $k$-means problem are typically approximated using Lloyd's iterative $k$-means algorithm (Lloyd, 1982; Forgy, 1965), which is so synonymous with the problem that it is referred to as *the* $k$-means algorithm. To this day, the latter is still considered one of the most important algorithms in data analysis (Wu et al., 2008).

However, with the rise of high-dimensional data analysis applications, such as gene expression, text analysis, and imaging, it has been observed that Lloyd's $k$-means algorithm encounters difficulties in high-dimensional settings (e.g., Hartigan, 1975; Steinley, 2006; Zha et al., 2001; Ding

[1]Department of Statistics and Data Science, Yale University, New Haven, CT [2]Department of Applied and Computational Mathematics, Yale University, New Haven, CT [3]School of Electrical and Computer Engineering, Tel Aviv University, Tel Aviv, Israel. Correspondence to: Roy R. Lederman <roy.lederman@yale.edu>.

*Proceedings of the 43rd International Conference on Machine Learning*, Seoul, South Korea. PMLR 306, 2026. Copyright 2026 by the author(s).

& He, 2004). In this paper, we prove that these difficulties reflect a critical problem that leads to a *catastrophic failure* of the algorithm *even in very easy problems*. To complete the picture, we also prove that Hartigan's $k$-means algorithm (Hartigan, 1975), a greedy variant of Lloyd's algorithm, succeeds where Lloyd's algorithm fails.

### 1.1. Main Results

The following is an abridged version of the results, omitting some of the nuances that are discussed in the standard background in Appendix A and in the full statements in Section 2.1, Corollary 3.8, and Corollary 3.12.

**Theorem 1.1** (Informal: high-noise, high-dimensional, finite-sample behavior of Lloyd vs. Hartigan)**.** *Consider* $n \in \mathbb{N}$ *observed samples* $x_1, \ldots, x_n$ *from a two-cluster* ($K = 2$) *Gaussian mixture model in* $\mathbb{R}^d$, *with standard normally distributed means* $\mu_1^\star, \mu_2^\star \in \mathbb{R}^d$ *and isotropic noise covariance* $\sigma^2 I_d$. *Let* $\mathcal{F}_{\mathrm{Lloyd}}$ *denote the event that* all (but exceptionally unbalanced) partitions *are fixed points of Lloyd's algorithm, and let* $\mathcal{F}_{\mathrm{Hart}}$ *denote the event that* there exists an incorrect partition *that is a fixed point of Hartigan's algorithm. Then, for* $\sigma^2 > n$,

$$1 - \mathbb{P}(\mathcal{F}_{\mathrm{Lloyd}}) \lesssim 2^n\, n \left(1 - \frac{1}{n^2}\right)^{d/4}, \qquad (1)$$

$$\mathbb{P}(\mathcal{F}_{\mathrm{Hart}}) \lesssim 2^n \left(1 - \frac{1}{4\sigma^4}\right)^{d/4}, \qquad (2)$$

*which yield the contrasting behaviors as* $d, n \to \infty$:

$$\mathbb{P}(\mathcal{F}_{\mathrm{Lloyd}}) \to 1 \quad \text{if} \quad d \gtrsim n^3, \qquad (3)$$

$$\mathbb{P}(\mathcal{F}_{\mathrm{Hart}}) \to 0 \quad \text{if} \quad d \gtrsim n\sigma^4, \qquad (4)$$

*where* $a \lesssim b$ *means that there exists some* $C > 0$ *such that* $a \leq Cb$.

That is, in the high-noise, high-dimensional regime of Theorem 1.1 and Corollary 3.8, *nearly every partition is already a fixed point of Lloyd's update map.* Since Lloyd's algorithm terminates at fixed points, this has a direct algorithmic implication: with high probability, for essentially any initialization (except extremely unbalanced ones), Lloyd's algorithm halts after the first update and returns the same

partition, i.e., it makes essentially no progress beyond its initialization.

It is tempting to attribute this phenomenon to an inherent geometric degeneracy in high dimensions (e.g., "all distances are nearly equal"), suggesting that meaningful clustering is impossible in this regime. However, our results for Hartigan's algorithm and the accompanying experiments show that this conclusion is incorrect: in the appropriate regime, even when Lloyd's algorithm becomes stuck at its initialization, a greedy local-improvement dynamics can still avoid spurious fixed points and recover the correct clustering with high probability.

In particular, since Hartigan's algorithm monotonically decreases the *k*-means objective and is guaranteed to terminate at a fixed point, Theorem 1.1 and Corollary 3.12 imply that, in the high-noise, high-dimensional regime, with high probability *there are no incorrect fixed points*. Consequently, from essentially any initialization, Hartigan's algorithm terminates at the correct partition.

### 1.2. Main Empirical Results

To place the phenomena in wider context, we compare Lloyd's and Hartigan's algorithms with several alternative clustering approaches: (i) PCA+ *k*-means, which first applies PCA and then runs Lloyd in the reduced space (Zha et al., 2001; Ding & He, 2004); (ii) a semidefinite-programming (SDP) relaxation from the modern family of *k*-means SDPs (Peng & Wei, 2007; Mixon et al., 2016); and (iii) a spectral clustering algorithm (Shi & Malik, 2000; Ng et al., 2001). We note that spectral clustering is not designed for the same objective, and we present comparisons with

it primarily for context and completeness. Both spectral clustering and the SDPs are less computationally scalable with $n$ than Lloyd's and Hartigan's iterations.

Figure 1 presents a summary of numerical experiments comparing these algorithms; see detailed description in Section 4.1. Across all $K$, Lloyd's algorithm exhibits a pronounced *failure region* (low normalized mutual information (NMI)) that persists well into regimes where the other algorithms succeed. Interestingly, Hartigan's algorithm appears to perform comparably to state-of-the-art alternative algorithms beyond the regimes where our theorems apply.

### 1.3. Prior Art

Over the years, *k*-means clustering and the behavior of Lloyd's *k*-means algorithm have been extensively studied. A large body of work provides statistical and computational conditions under which Lloyd's algorithm, or closely related procedures, recover the correct clustering or the cluster means (e.g., Lu & Zhou, 2016; Gao & Zhang, 2022; Ndaoud, 2022; Chen & Yang, 2021). In addition, previous comparisons between Lloyd's and Hartigan's algorithms (Telgarsky & Vattani, 2010; Slonim et al., 2013) have shown that the fixed points of Hartigan's algorithm are a subset of those of Lloyd's.

Our results complement these lines of work by identifying a broad high-noise, high-dimensional finite-sample regime in which Lloyd's algorithm exhibits a fixed-point abundance phenomenon, so that, with high probability, it cannot improve upon its initialization, while Hartigan's algorithm avoids this pathology by having no incorrect fixed points with high probability.

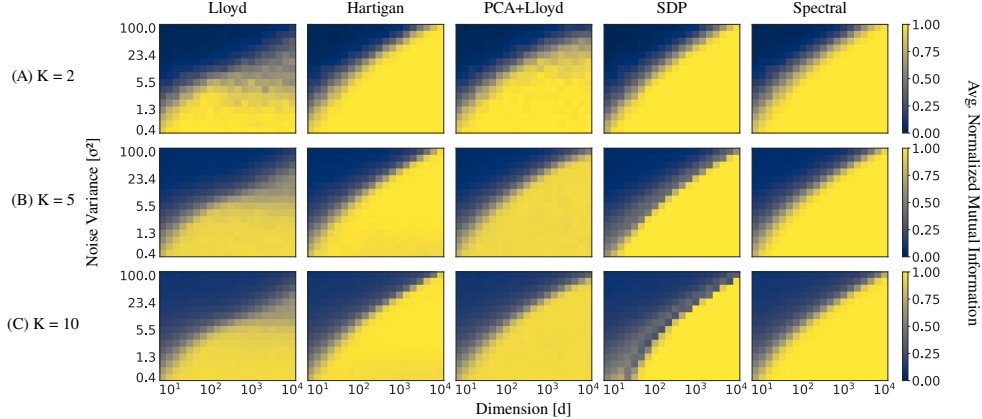

*Figure 1.* Normalized mutual information (NMI; see Definition A.13) between the ground-truth partition and the output of each clustering algorithm. Each entry reports the mean over 100 independent trials: in each trial, we sample data from the Gaussian mixture model (GMM) in Model 2.1 (generalized to $K \geq 2$) with $\tau^2 = 1.0$ and 20 samples per class, and run each algorithm until convergence, with Lloyd's algorithm initialized using *k*-means++ and Hartigan's algorithm initialized with a random partition; Supplementary Figure 3 reports results for both algorithms under different initialization strategies. The results illustrate that in the high-noise, high-dimensional regime, Lloyd's *k*-means performs poorly relative to the other methods. In contrast, Hartigan's algorithm achieves performance comparable to spectral clustering and semidefinite-programming (SDP) based clustering. See Section 4.1 for details.

The remainder of the paper is organized as follows. Section 2 summarizes the observational model, the $k$-means problem, and the precise statements of Lloyd's and Hartigan's update rules used throughout the paper. Section 3 introduces the analytical apparatus and states the formal versions of the results summarized in Theorem 1.1 (Theorems 3.4, 3.6 and Corollary 3.8 for Lloyd's algorithm, and Theorem 3.9 together with the corollaries that follow for Hartigan's algorithm). Section 4 reports numerical experiments on a Gaussian mixture model and on real datasets that illustrate the failure region and the empirical comparison with PCA+$k$-means, the SDP relaxation, and spectral clustering. The conclusions are summarized in Section 5. Standard facts used in the proofs, the detailed pseudocode of both algorithms, the full proofs, and additional experiments are deferred to Appendix A–Appendix F.

## 2. Preliminaries

This section collects the definitions and notation used throughout the paper. We denote the index set $\{1, 2, \ldots, n\}$ by $[n]$. We use $\|\cdot\|$ for the vector Euclidean norm.

### 2.1. The Observational Model

We analyze the $k$-means problem in the two-component GMM ($K = 2$). The random variables considered throughout this analysis are formalized in the following classic model, where there are $K = 2$ centers $\mu_1^\star$ and $\mu_2^\star$, and each observation is a noisy version of one of the centers.

**Model 2.1** (Two-component isotropic GMM). The observations $X = \{x_i\}_{i=1}^n$ are drawn according to $x_i := \mu_{z_i^\star}^\star + \xi_i$, where the underlying random variables are as follows:

(a) *The ground-truth class centers* are random and i.i.d.:
$\mu_1^\star, \mu_2^\star \overset{\text{i.i.d.}}{\sim} \mathcal{N}\left(0, \tau^2 I_d\right)$ with $\tau \in \mathbb{R}^+$.

(b) *The sample noise* is random, i.i.d., and independent of the ground-truth centers: $\xi_i \overset{\text{i.i.d.}}{\sim} \mathcal{N}\left(0, \sigma^2 I_d\right)$.

(c) *The ground-truth class assignment* is given by latent labels $z_1^\star, \ldots, z_n^\star \in \{1, 2\}$. We denote the ground-truth classes by $S_\ell^\star := \{i \in [n] : z_i^\star = \ell\}$. At this point, we do not assume a specific distribution of the class assignment, only that neither class is empty.

We assume that (a), (b) and (c) are independent.

### 2.2. The $k$-Means Problem

The $k$-means problem admits several equivalent formulations; we adopt the standard partition-based one. Given samples $x_1, \ldots, x_n \in \mathbb{R}^d$ and an integer $K \geq 2$, the $k$-*means problem* asks for a partition of the index set $[n]$ into $K$ nonempty clusters $S = \{S_1, \ldots, S_K\}$ that minimizes

the within-cluster sum of squared distances,

$$\arg\min_S \sum_{k=1}^K \sum_{i \in S_k} \|x_i - \mu_k\|^2, \tag{5}$$

where $\mu_k = |S_k|^{-1} \sum_{i \in S_k} x_i$. This loss is often referred to as Inertia, Within-Cluster Sum of Squares (WCSS or WSS), Sum of Squared Errors (SSE), or Distortion. The $k$-means problem is known to be NP-hard (Aloise et al., 2009).

We find the following definitions useful for presenting and analyzing Lloyd's and Hartigan's algorithms for $K = 2$.

**Definition 2.2** (Current assignment, clusters, and partition). At iteration $t$, the *current cluster assignment* is a labeling vector $z^{(t)} = (z_1^{(t)}, \ldots, z_n^{(t)}) \in \{1, 2\}^n$. It induces the current clusters $C_j^{(t)} := \{i \in [n] : z_i^{(t)} = j\}$ and the corresponding (current) partition $\mathcal{P}^{(t)} := \{C_1^{(t)}, C_2^{(t)}\}$. Unless stated otherwise, we restrict attention to iterates for which both clusters are nonempty, i.e., $|C_1^{(t)}|, |C_2^{(t)}| > 0$. We omit the iteration index $t$ where it is not needed. For notational convenience, we reserve the term "class" for the ground truth $S_\ell^\star$ and the term "cluster" for the current state of the algorithm $C_j^{(t)}$, and we use the term "partition" in the context of clusters.

**Definition 2.3** (Centroids). Let us consider a bipartition $\mathcal{P}^{(t)} = \{C_1^{(t)}, C_2^{(t)}\}$; the associated empirical centroids are

$$\widehat{\mu}_j^{(t)} := \frac{1}{|C_j^{(t)}|} \sum_{i \in C_j^{(t)}} x_i, \qquad j \in \{1, 2\}. \tag{6}$$

#### 2.2.1. LLOYD'S ALGORITHM

Lloyd's algorithm for $k$-means (Lloyd, 1982) is an alternating-minimization heuristic for approximately minimizing the objective in Equation (5). Starting from an initial nonempty partition, the algorithm iterates the following two steps.

*Assignment step.* Given the current centroids $\widehat{\mu}_1^{(t)}, \widehat{\mu}_2^{(t)}$, reassign each sample to the nearest centroid:

$$z_i^{(t+1)} \in \arg\min_{j \in \{1,2\}} \|x_i - \widehat{\mu}_j^{(t)}\|^2, \qquad i \in [n]. \tag{7}$$

*Averaging step.* Given the updated partition, recompute the centroids as empirical means:

$$\widehat{\mu}_j^{(t+1)} = \frac{1}{|C_j^{(t+1)}|} \sum_{i \in C_j^{(t+1)}} x_i, \qquad j \in \{1, 2\}. \tag{8}$$

Lloyd's algorithm is monotonically decreasing in the loss (Equation (5)) and guaranteed to converge (see, for example, Slonim et al., 2013, p. 1678). More detailed pseudocode, specialized to the setting of this paper, is provided in Appendix B.

### 2.2.2. HARTIGAN'S ALGORITHM

Hartigan's algorithm (Hartigan, 1975) is a greedy algorithm for minimizing the $k$-means loss (Equation (5)). In contrast to Lloyd's batch reassignment, Hartigan updates the partition one sample at a time. For each individual sample, Hartigan's algorithm reassigns the sample to the nearest centroid based on the *Hartigan weighted distance*:

$$
\Delta_{\mathrm{H}}^2(x_i, C_j^{(t)}) := \begin{cases} \frac{|C_j^{(t)}|}{|C_j^{(t)}|-1} \, \|x_i - \widehat{\mu}_j^{(t)}\|^2, & \text{if } i \in C_j^{(t)}, \\ \frac{|C_j^{(t)}|}{|C_j^{(t)}|+1} \, \|x_i - \widehat{\mu}_j^{(t)}\|^2, & \text{if } i \notin C_j^{(t)}. \end{cases}
$$
(9)

The algorithm repeatedly sweeps through the samples until no relocation is accepted, at which point the output partition is locally optimal with respect to single-sample moves (i.e., a 1-swap local optimum).

Hartigan's algorithm is monotonically decreasing in the loss (Equation (5)) and guaranteed to converge (Slonim et al., 2013, p. 1678). More detailed pseudocode, specialized to the setting of this paper, is provided in Appendix B.

### 2.2.3. ADDITIONAL NOTATION

To state the main results in the next section, it is convenient to introduce the following definition.

**Definition 2.4** (Class proportions, purity, and correctness). Let $S_1^\star, S_2^\star \subseteq [n]$ denote the ground-truth classes such that $S_1^\star \cap S_2^\star = \emptyset$ and $S_1^\star \cup S_2^\star = [n]$. Let $\mathcal{P} = \{C_1, C_2\}$ be a (current) partition of $[n]$ into two nonempty clusters. We define

(i) *Class proportions*: $R^1 := \frac{|S_1^\star|}{n}, R^2 := \frac{|S_2^\star|}{n} = 1 - R^1$.

(ii) *Purity coefficient* of cluster $C_j$ with respect to class $\ell$: $R_j^\ell := |C_j \cap S_\ell^\star|/|C_j|$.

(iii) The partition $\mathcal{P}$ is *correct* if it agrees with the ground-truth classes up to permutation, i.e., either it holds that $(C_1, C_2) = (S_1^\star, S_2^\star)$ or $(C_1, C_2) = (S_2^\star, S_1^\star)$.

For a cluster $j$ or class $\ell$, we denote by $\overline{j}$ or $C_{\overline{j}}$ the other cluster, and $\overline{\ell}$ or $S_{\overline{\ell}}^\star$ the other ground-truth class, so that $\overline{1} = 2, \overline{2} = 1, C_{\overline{1}} = C_2, C_{\overline{2}} = C_1, S_{\overline{1}}^\star = S_2^\star$ and $S_{\overline{2}}^\star = S_1^\star$.

### 2.3. Approximately Balanced Partitions

A large fraction of all the partitions are mostly "balanced" in the sense that the size of each set in the partition is close to $n/2$. This can be proved by a probabilistic argument, see e.g., Fact A.10 in the appendix. We fix $n$ and examine all the partitions with cluster sizes within $q$ standard deviations of $n/2$.

**Definition 2.5** (*q*-approximately balanced partitions). Fix $n \in \mathbb{N}$ and a parameter $q > 0$. A bipartition $\mathcal{P} = \{C_1, C_2\}$ of $[n]$ (i.e., $C_1 \cap C_2 = \emptyset$ and $C_1 \cup C_2 = [n]$) is said to be *q-approximately balanced* if both clusters have sizes $|C_k| > 2$ and within $q$ standard deviations of $n/2$, namely, $n/2 - q\sqrt{n/4} < |C_k| < n/2 + q\sqrt{n/4}$, for $k \in \{1, 2\}$.

For large $q$, the set of all $q$-approximately balanced bipartitions of $[n]$ contains *all but an exponentially small fraction of bipartitions*, and thus captures the "typical" case. The arguments can be adapted to allow for $q$ to depend on $n$ such that the statement holds for a proportion of partitions converging to 1 as $n \to \infty$, see Remark C.4.

## 3. Analytical Apparatus

The goal of this paper is to exhibit a basic high-dimensional regime in which Lloyd's algorithm fails with high probability, despite the simplicity of the underlying two-cluster model. Theorem 3.4 isolates the case of a single sample's probability of being reassigned to a new cluster by Lloyd's algorithm. The intuition behind the proof in Appendix C.3 is to consider a partition that agrees with the ground truth except for a single misclassified sample; one may intuitively expect such a near-correct initialization to be immediately repaired by the next Lloyd assignment step. Instead, we prove that in the high-noise, high-dimensional regime, the misclassified sample remains misclassified, because it is (with high probability) closer to its current empirical centroid in the wrong cluster than to the empirical centroid of the correct cluster. Intuitively, this single misclassification is the "most favorable case" for the algorithm, and therefore can serve as a bound: the proof in Appendix C.3 turns this intuition into a rigorous argument. Having established this one-sample persistence phenomenon, we extend it uniformly over broad families of partitions in Theorem 3.6, and then use union bounds to argue about all samples in all approximately balanced partitions (which are almost all partitions) in Corollary 3.8.

### 3.1. Distances to the Two Cluster Centers

Our analysis of both Lloyd's and Hartigan's algorithms hinges on the distribution of squared distances between a sample and the empirical cluster centroids. Under the isotropic Gaussian model, such squared distances are (up to deterministic scaling) chi-squared with $d$ degrees of freedom. The next two lemmas formalize these distance laws under our model. Lemma 3.1 characterizes the distribution of the distance from a sample to the centroid of its currently assigned cluster, while Lemma 3.2 characterizes the distance to the centroid of the other (competing) cluster. Proofs are deferred to Appendix C.2.

**Lemma 3.1** (Distance to the current cluster centroid). *Consider the setting of Model 2.1. Fix $i \in [n]$. Let $j \in \{1, 2\}$ be the (current) cluster index such that $i \in C_j$, and let*

$\ell \in \{1, 2\}$ be the ground-truth class index such that $i \in S_\ell^\star$. Then, the distance between the sample $x_i$ and the centroid $\widehat{\mu}_j$ of a cluster $C_j$ is distributed as

$$\|x_i - \widehat{\mu}_j\|^2 \sim \alpha_{\mathrm{cur}}\chi_d^2 \ ,$$
$$\alpha_{\mathrm{cur}} = 2\tau^2(1 - R_j^\ell)^2 + (1 - 1/|C_j|)\,\sigma^2 \ . \tag{10}$$

where $R_j^\ell$ is the cluster purity defined in Definition 2.4.

**Lemma 3.2** (Distance to the other cluster centroid). *In the settings of Lemma 3.1, the distance between the sample $x_i$ and the centroid $\widehat{\mu}_{\overline{j}}$ of a cluster $C_{\overline{j}}$ is distributed as*

$$\left\|x_i - \widehat{\mu}_{\overline{j}}\right\|^2 \sim \alpha_{\mathrm{alt}}\chi_d^2 \ ,$$
$$\alpha_{\mathrm{alt}} = 2\tau^2\left(1 - R_{\overline{j}}^\ell\right)^2 + \left(1 + 1/|C_{\overline{j}}|\right)\sigma^2. \tag{11}$$

We emphasize that, in both lemmas, the ground-truth latent class labels (Model 2.1(c)) and the current cluster assignment (and therefore the bipartition $\mathcal{P} = \{C_1, C_2\}$ in Definition 2.2) are treated as fixed. The only relevant sources of randomness are the ground-truth centers and additive noise (Model 2.1(a)–(b)), which are independent of the class and cluster labels.

In the sequel, comparisons of the distances in the lemmas above reduce to events involving differences of scaled $\chi_d^2$ random variables (with scaling factors given by $\alpha_{\mathrm{cur}}$ and $\alpha_{\mathrm{alt}}$). The following lemma provides a convenient tail bound for such differences; its proof, based on the Chernoff method (Chernoff, 1952), appears in Appendix C.1.

**Lemma 3.3.** *Fix $b_1 > b_2 > 0$, and $m \in \mathbb{R}$. Let $Y_1 \sim b_1\chi_d^2$ and $Y_2 \sim b_2\chi_d^2$ be scaled chi-squared distributed with $d$ degrees of freedom, not necessarily independent. Then,*

$$\mathbb{P}\left(Y_1 - Y_2 \le m\right) \le \exp\left(m\frac{b_1 - b_2}{8b_1 b_2}\right)\rho^{d/4}, \tag{12}$$

*where $\rho = 1 - \left((b_1 - b_2)/(b_1 + b_2)\right)^2 < 1$.*

### 3.2. Analysis of Lloyd's Algorithm

Having introduced the technical machinery, we proceed according to the strategy described at the beginning of this section.

#### 3.2.1. SINGLE-SAMPLE REASSIGNMENT PROBABILITY

In this section, we study the probability that a fixed sample $x_i$ changes its assignment in the next Lloyd iteration. Let $j \in \{1, 2\}$ be its current cluster index (so $i \in C_j$) and let $\overline{j}$ denote the other cluster. Lloyd reassigns $x_i$ to $C_{\overline{j}}$ if and only if $\|x_i - \widehat{\mu}_{\overline{j}}\|^2 < \|x_i - \widehat{\mu}_j\|^2$, equivalently, if the difference $\|x_i - \widehat{\mu}_{\overline{j}}\|^2 - \|x_i - \widehat{\mu}_j\|^2$ is negative; otherwise the sample remains in $C_j$ for the next iteration.

**Theorem 3.4** (Lloyd's algorithm: single sample). *We consider the setting of Model 2.1, a fixed $i \in [n]$ with current cluster $C_j$ and other cluster $C_{\overline{j}}$ such that $i \in C_j$. We denote the cluster sizes $c := |C_j|$ and $\overline{c} := |C_{\overline{j}}|$.*

*If the noise level $\sigma > 0$ satisfies*

$$\sigma > \frac{\sqrt{2\overline{c}}\,\tau\,(c - 1)}{\sqrt{c(c + \overline{c})}} \ , \tag{13}$$

*then, we have*

$$\mathbb{P}\left(\left\|x_i - \widehat{\mu}_{\overline{j}}\right\|^2 < \left\|x_i - \widehat{\mu}_j\right\|^2\right) \le \rho^{d/4}, \tag{14}$$

*where $\rho$ is defined by*

$$\rho(\sigma, \tau, c, \overline{c}) =$$
$$\frac{4\sigma^2(c-1)c^2\overline{c}(\overline{c}+1)\left(c(\sigma^2+2\tau^2)-2\tau^2\right)}{(-c(\sigma^2+4\tau^2)\overline{c}+c^2(\sigma^2+2(\sigma^2+\tau^2)\overline{c})+2\tau^2\overline{c})^2}, \tag{15}$$

*and satisfies $0 \le \rho < 1$.*

The proof of Theorem 3.4 is given in Appendix C.3. It combines the distance characterizations in Lemmas 3.1 and 3.2 with the tail bound for differences of scaled chi-squared variables in Lemma 3.3. The resulting estimate is uniform over all partitions with the prescribed cluster sizes and, in particular, applies to the "most favorable" incorrect initialization in which the partition agrees with the ground truth except for a single misclassified sample.

*Remark* 3.5. The condition on the noise in Equation (13) is the threshold where the expected distance from a sample to its current cluster exceeds the expected distance to the other cluster, even in the "most favorable," or "easiest to fix" incorrect initialization. On the technical level, it is the condition required to satisfy the requirements of Lemma 3.3 (see proof for details).

#### 3.2.2. SAMPLES IN APPROXIMATELY BALANCED PARTITIONS

The following theorem, which is proved in Appendix C.4, provides a uniform bound over all $q$-approximately balanced partitions (Definition 2.5), obviating the need for explicit cluster sizes. To simplify the notation in this theorem, we fix the signal scale $\tau = 1$ without loss of generality.

**Theorem 3.6** (Uniform bound for $q$-approximately balanced partitions). *Consider the setting of Model 2.1, with a fixed $i \in [n]$ with current cluster index $j$ and other cluster $\overline{j}$. Fix a partition imbalance factor $q > 1$ and assume a partition $\mathcal{P} = \{C_1, C_2\}$ that is $q$-approximately balanced (Definition 2.5). Fix $\tau = 1$, and fix $\beta > 1$, such that*

$$\sigma = \beta\frac{(\sqrt{n}\,q + n - 2)}{\sqrt{2}\sqrt{\sqrt{n}\,q + n}}. \tag{16}$$

*Then,*

$$\mathbb{P}\left(\left\|x_i - \widehat{\mu}_{\overline{j}}\right\|^2 - \left\|x_i - \widehat{\mu}_j\right\|^2 < 0\right) \le \rho_q^{d/4}, \qquad (17)$$

*where*

$$\rho_q = \frac{\sigma^2\left(\sqrt{n}q+n-2\right)\left(\sqrt{n}q+n\right)\left(\sqrt{n}q+n+2\right)\left(\sqrt{n}\left(\sigma^2+2\right)\left(\sqrt{n}+q\right)-4\right)}{\left(n\sigma^2\left(\sqrt{n}+q\right)^2+\left(\sqrt{n}q+n-2\right)^2\right)^2}. \tag{18}$$

*Remark* 3.7 (Asymptotics of Theorem 3.6). The expressions in Theorem 3.6 become more interpretable for large $n$ (with fixed $q$ and $\beta$). Squaring the expression in Equation (16) and expanding it in $n$ yields

$$\sigma^2 = \frac{\beta^2 n}{2} + \frac{\beta^2 q\sqrt{n}}{2} - 2\beta^2 + \frac{2\beta^2}{n} + O\left(n^{-3/2}\right). \quad (19)$$

Substituting Equation (16) into Equation (18) and expanding it in $n$ yields

$$\rho_q = 1 - \frac{4\left(\beta^2-1\right)^2 n^{-2}}{\beta^4} + \frac{8\left(\beta^2-1\right)^2 n^{-5/2}q}{\beta^4} + O\left(n^{-3}\right). \tag{20}$$

Theorem 3.6 implies that in the appropriate regime, the probability that a given sample in an approximately balanced partition would switch over to a different cluster in the next iteration of Lloyd's $k$-means algorithm is small and decreases as the dimension $d$ grows.

### 3.2.3. ALL APPROXIMATELY BALANCED PARTITIONS ARE FIXED POINTS OF LLOYD'S ALGORITHM

Recall from Section 2.3 that an overwhelming proportion of partitions are nearly balanced, and by setting the appropriate $q$, all partitions are $q$-approximately balanced partitions (with the exception of partitions with clusters of size 2 or less). The following generalizes Theorem 3.6 to a statement about the probability that any partition is not a fixed point of Lloyd's algorithm (with the same exclusions as before).

**Corollary 3.8** (Main result: Lloyd's algorithm). *Consider the setting of Model 2.1. Fix an imbalance parameter $q > 1$ and assume the noise level satisfies Equation (16). Then the probability that there exists a $q$-approximately balanced partition (see Definition 2.5) that is* not *a fixed point of Lloyd's $k$-means update scheme is upper bounded by*

$$\mathbb{P}\left(\exists \text{ approx. balanced partition that is not a fixed point}\right)$$
$$\le 2^n n\rho_q^{d/4}, \tag{21}$$

*where $\rho_q$ is defined in* (18).

The proof is presented in Appendix C.5. The idea is to extend Theorem 3.6 using the union bound over multiple samples in a single partition, and then over multiple partitions.

Examining the distance distributions in the proofs clarifies the mechanism behind Lloyd's failure. When $i \in C_j$,

the centroid $\widehat{\mu}_j$ is computed from an average that includes $x_i$, so $x_i$ exerts a non-negligible "self-influence" and pulls $\widehat{\mu}_j$ toward itself by an amount on the order of $1/|C_j|$. In the high-noise, high-dimensional regime, this self-influence can dominate the weak class-separation signal: a misassigned sample can remain closer to the centroid of its current (wrong) cluster than to the competing centroid, even when the rest of the partition is correct. In sufficiently extreme settings, this effect is strong enough that Lloyd's update fails to repair even a single misclassification with high probability. We mention that the same conclusion holds under centroid initialization: the first assignment step induces a partition that is already a fixed point (w.h.p., and with the same exclusions), so no further progress occurs.

### 3.3. Analysis of Hartigan's Algorithm

For Hartigan's algorithm, we apply similar tools, but in the opposite direction. In Theorem 3.9, we bound the probability that a sample *fails* to relocate when its current cluster contains no greater proportion of its ground-truth class than the other cluster. We then bound the probability that an incorrect partition is a fixed point in Corollary 3.10, establish a uniform bound over a wider range of parameters in Corollary 3.11, and infer that w.h.p., no incorrect partition is a fixed point of Hartigan's dynamics in Corollary 3.12.

**Theorem 3.9** (Hartigan's algorithm: single sample). *In the setting of Model 2.1, fix an index $i \in [n]$ and let $\ell \in \{1,2\}$ be its ground-truth class (so $i \in S_\ell^\star$). Let $j \in \{1,2\}$ be its current cluster index (so $i \in C_j$) and let $\overline{j}$ denote the other cluster. If the cluster purity coefficient (Definition 2.4) satisfies*

$$0 < R_j^\ell \le R_{\overline{j}}^\ell \le 1, \tag{22}$$

*then,*

$$\mathbb{P}\left(\Delta_H^2(x_i, C_j) \le \Delta_H^2(x_i, C_{\overline{j}})\right) \le \rho^{d/4}, \qquad (23)$$

*where $\Delta_H$ is defined in Equation (9) and $\rho$ is given by*

$$\rho =$$
$$1 - \left(\frac{\tau^2\left(\frac{|C_j|}{|C_j|-1}(1-R_j^\ell)^2 - \frac{|C_{\overline{j}}|}{|C_{\overline{j}}|+1}(1-R_{\overline{j}}^\ell)^2\right)}{\tau^2\left(\frac{|C_j|}{|C_j|-1}(1-R_j^\ell)^2 + \frac{|C_{\overline{j}}|}{|C_{\overline{j}}|+1}(1-R_{\overline{j}}^\ell)^2\right)+\sigma^2}\right)^2. \tag{24}$$

*and satisfies $0 \le \rho < 1$.*

The proof is deferred to Appendix D.2. It follows the same outline as the Lloyd analysis, but replaces the usual squared distances by Hartigan's rescaled distances $\Delta_H^2(\cdot, \cdot)$ defined in Equation (9); accordingly, we use analogues of Lemmas 3.1 and 3.2 tailored to the Hartigan weighting.

In particular, a partition $\mathcal{P} = \{C_1, C_2\}$ can be a fixed point of Hartigan's dynamics only if *every* sample prefers (in the

Hartigan sense) its current cluster over the other one; thus, it suffices to exhibit a single index $i$ for which a Hartigan move is strictly improving. The next corollary is therefore an immediate consequence of Theorem 3.9.

**Corollary 3.10.** *If the conditions of Theorem 3.9 are satisfied, then the probability that the partition $\mathcal{P}$ is a fixed point of Hartigan's algorithm is bounded by*

$$\mathbb{P}(\mathcal{P} \text{ is a fixed point}) \leq \rho^{d/4}, \tag{25}$$

*where $\rho$ is given in Equation (24).*

The bound in Theorem 3.9 depends on the specific cluster sizes and the corresponding purity coefficients. The following corollary, proved in Appendix D.3, generalizes this pointwise estimate to a uniform statement: it provides a bound that holds simultaneously over all admissible cluster sizes and over all purity configurations corresponding to partitions that do not coincide with the ground-truth classes (up to permutation).

**Corollary 3.11.** *Consider the setting of Model 2.1. Further assume that $n \geq 4$. If the current partition $\mathcal{P}$ is nonempty and is an incorrect partition (Definition 2.4), then, the probability that $\mathcal{P}$ is a fixed point is bounded by*

$$\mathbb{P}\left(\mathcal{P} \text{ is a fixed point}\right) \leq \rho_h^{d/4}, \tag{26}$$

*where $\rho_h$ is given by*

$$\rho_h = 1 - \left(\frac{4\tau^2(R^\star)^2 n^{-1}}{3\tau^2 + \sigma^2}\right)^2 < 1, \tag{27}$$

*and $R^\star = \min(R^1, R^2)$ is the relative size of the smallest ground-truth class.*

This uniform bound in Corollary 3.11 enables a union bound over the family of non-correct bipartitions and leads to the next corollary (proved in Appendix D.4), which bounds the probability that *any* partition that does not match the ground truth (up to permutation) is a fixed point of the algorithm.

**Corollary 3.12** (Main result: Hartigan's algorithm)**.** *Consider the setting of Model 2.1. Assume that $n \geq 4$. Then, the probability that there is any nonempty incorrect partition (Definition 2.4) that is a fixed point of Hartigan's algorithm is bounded by*

$$\mathbb{P}\left(\exists \mathcal{P} \text{ a nonempty incorrect partition}\right) \leq 2^n \rho_h^{d/4}, \tag{28}$$

*where $\rho_h$ is given by (27).*

## 4. Numerical Results

In this section, we report numerical experiments illustrating the failure of Lloyd's *k*-means algorithm in the high-noise,

high-dimensional regimes studied in this paper. Alongside Lloyd's method, we evaluate Hartigan's algorithm and several modern alternatives: the common high-dimensional heuristic PCA+*k*-means, which applies PCA and then runs Lloyd in the reduced space (Zha et al., 2001; Ding & He, 2004); an SDP relaxation from the modern family of *k*-means SDPs (Mixon et al., 2016) and spectral clustering (Shi & Malik, 2000; Ng et al., 2001), run via the standard scikit-learn implementation (Pedregosa et al., 2011).

Additional numerical results are presented in Appendix F. Implementation details are available in Appendix E. Our code is freely available at `https://github.com/Lederman-Group/Catastrophic_Failure_KMeans`, and the scripts for reproducing the results in this paper are available at `https://zenodo.org/records/20435365`.

### 4.1. Synthetic GMM

The first set of experiments illustrates the connection between the theoretical findings in this paper and the performance of Hartigan's and Lloyd's *k*-means, and compares them with other clustering algorithms. We sample data from the GMM defined in Model 2.1 (generalized to $K \geq 2$) at different dimensions $d$ and noise variances $\sigma^2$, for $K = 2, 5$ and 10 clusters, $n = 20 \times K$ samples, and $\tau^2 = 1$. We measure the performance of each algorithm in two ways: i) how well it recovers the ground-truth in terms of the NMI score (Definition A.13), and ii) the *k*-means loss of the solution.

In each experiment, we generate a new dataset and run each clustering algorithm with a single random initialization. We repeat the experiment 100 times for each combination of values of $d$, $\sigma^2$, and $K$. In the case of Lloyd's and Hartigan's algorithms, as well as the PCA+*k*-means algorithm, we evaluate the performance for three different initialization strategies: i) a "random partition" initialization, which randomly selects equal-size clusters and sets the initial centroids as the corresponding clusters' averages; ii) a "random centers" initialization, which designates $K$ random samples as centers; and iii) the popular *k-means++* initialization (Arthur & Vassilvitskii, 2006).

In Figure 1, we present the NMI (see Definition A.13) between each method and the ground-truth partition; in this figure, we restrict our attention to *k-means++* initialization in the case of Lloyd's algorithm and random balanced partition initialization in the case of Hartigan's algorithm; a discussion of the initialization strategy follows below. Overall, Hartigan's *k*-means and spectral clustering achieve the highest accuracy across all values of $K$; we note again, as context for this comparison, that spectral clustering is not designed for the same objective. The SDP relaxation performs well for $K = 2$, but degrades more noticeably at higher noise levels as $K$ increases. Lloyd's algorithm

performs worst in the high-noise, high-dimensional regime, although a simple PCA preprocessing step (PCA+$k$-means) substantially improves its accuracy. Additional metrics are presented in Supplementary Figure 2 in Appendix F.2.

We note that $k$-means++, used for Lloyd's algorithm in the results in Figure 1, provides the algorithm with centroids as initial guesses, whereas our analysis of Lloyd's algorithm considers partitions. A preliminary empirical evaluation of alternative initialization strategies is presented in Appendix F.2, demonstrating that $k$-means++ outperforms random partitions as initialization for Lloyd's algorithm. While we defer the detailed study of initialization by centroids to future work, informally, we note that in some cases, good initial centers can lead to a good first partition, which is already a fixed point of the algorithm. We consider this a case where the algorithm itself "does not do anything" beyond partitioning directly based on the initialization. Figure 1 illustrates that even with this advantageous initialization, Lloyd's algorithm is outperformed by other algorithms.

### 4.2. Real-World Datasets

We compare the clustering algorithms on the following real-world datasets: the *Olivetti* faces dataset (Samaria & Harter, 1994), which contains images of 40 individuals in 10 different poses, the Amazon Commerce Reviews dataset (Liu, 2011), which contains 1,500 documents authored by $K = 50$ distinct users, and four datasets derived from the 20 newsgroups dataset (Mitchell, 1997; Lang, 1995), denoted by 20NG-A, 20NG-B, 20NG-C, and 20NG-D with $K = 2, 5, 10$, and $20$, respectively. Further details on these datasets are provided in Appendix E.

For each dataset, we run Lloyd's and Hartigan's $k$-means, spectral clustering, and SDP. For each algorithm (except the deterministic SDP), we select the output that results in the smallest $k$-means loss (Equation (5)) out of 500 independent initializations. SDP is run only once since it is deterministic; it is run until convergence. In all experiments, we initialize Lloyd's $k$-means and spectral clustering with $k$-means++, and Hartigan's $k$-means with a random balanced partition. Table 1 summarizes the results obtained for each dataset. Supplementary Table 3 reports results for larger versions of the 20NG datasets, restricted to Lloyd's and Hartigan's $k$-means and spectral clustering, since SDP does not scale well to this regime.

We note that while spectral clustering outperformed Hartigan's algorithm in terms of NMI in the Olivetti example, Hartigan's algorithm achieved a better loss, which is the criterion it is designed to optimize.

## 5. Discussion and Conclusions

Our analysis of Lloyd's $k$-means offers a concrete explanation for its empirical breakdown in high-dimensional, high-noise regimes. In this setting, with high probability over the sampled dataset, every approximately balanced partition is already a fixed point of Lloyd's update map. Consequently, except for extremely unbalanced initializations, the algorithm halts after the first update and returns the initial partition (or, under centroid initialization, the partition induced by the initial centers), making essentially no progress beyond its starting point. While sensitivity to initialization is well known for $k$-means (Balanov et al., 2025), our results identify an extreme regime in which the initialization is essentially the outcome: Lloyd's algorithm makes essentially no progress beyond its starting point and returns the initial partition.

The theoretical bounds we obtain are conservative and are intended to certify the existence of this phenomenon rather than pinpoint sharp thresholds. An interesting direction for future work is to understand how this fixed point proliferation weakens as noise or dimension decreases, and to quantify the probability that Lloyd's dynamics become trapped in suboptimal fixed points even when not all partitions are fixed points.

In sharp contrast, Hartigan's algorithm does not exhibit the fixed point proliferation that traps Lloyd's updates. In our two-component Gaussian model, once the dimension is sufficiently large (a regime in which the clustering task becomes easier), we show that Hartigan's greedy local-improvement dynamics has no incorrect fixed points with high probability and therefore terminates at the correct partition (up to label permutation).

As with the Lloyd analysis, our guarantees are conservative, and the dimension thresholds are likely far from tight. Empirically, Hartigan's algorithm remains substantially more robust at much lower dimensions, and its performance appears to be often comparable to powerful modern alternatives to Lloyd's method. Many of these alternatives are considerably more difficult to scale; for instance, SDP relaxations operate on $n \times n$ matrices, whereas Hartigan's algorithm, like Lloyd's, relies only on repeated comparisons of samples to empirical centroids. A careful runtime/accuracy tradeoff study depends on implementation details and is outside the scope of this work, but our findings motivate the theoretical question of whether one can obtain sharper complexity guarantees and SDP-like recovery guarantees for Hartigan's algorithm in high-dimensional mixture models and in broader settings, and whether the conservative bounds here can be significantly tightened. We note that the computational complexity of Lloyd's and Hartigan's algorithms has been studied, for example, in (Vattani, 2011; Manthey et al., 2024; Arthur et al., 2011; Manthey & van Rhijn, 2024).

The comparison to spectral clustering demonstrates that Hartigan's algorithm can, in some cases and some metrics, be comparable to spectral clustering when Lloyd's algorithm is not. This is an interesting subject for future investigation, but this paper does not imply that Hartigan's algorithm outperforms spectral clustering in general; spectral clustering outperforms Hartigan's algorithm in other cases, and the comparison is nuanced because spectral clustering does not optimize the same objective.

Although our theory focuses on the two-cluster case, the empirical results in Section 4 indicate that the same qualitative separation between Lloyd and Hartigan persists for larger numbers of clusters. Extending the analysis to general $K$ appears feasible: For Lloyd, the fixed point mechanism is expected to carry over with minor modifications, whereas a corresponding extension for Hartigan requires additional care (e.g., handling multiple competing moves and cluster-size effects across $K$ clusters). We defer a full $K > 2$ theoretical treatment of both algorithms to future work.

As noted, for example, in Bottou & Bengio (1994); Bishop (2006), Lloyd's algorithm is closely related to the expectation–maximization (EM) algorithm (Dempster et al., 1977), another fundamental tool in statistical learning and data analysis (Wu et al., 2008). The effect studied appears to carry over to the EM algorithm; in high-noise, high-dimensional settings, EM can likewise become trapped near its initialization. We also defer a detailed analysis of this question to future work.

## Impact Statement

This paper presents work whose goal is to advance the field of machine learning. There are many potential societal consequences of our work, none of which we feel must be specifically highlighted here.

## Acknowledgements

The authors would like to thank Adar Pfeffer, Amit Singer, Fred Sigworth, Sheng Xu, Zhou Fan, and Yihong Wu for helpful discussions. The authors would like to thank the Yale Center for Research Computing (YCRC) for providing computing resources and support. The work was supported by NIH/NIGMS (1R35GM157226), the Alfred P. Sloan Foundation (FG-2023-20853), and the Simons Foundation (1288155).

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

*Table 1.* **Clustering results for real-world datasets using Lloyd's and Hartigan's *k*-means, spectral clustering, and SDP.** 500 random initializations per dataset for Lloyd's *k*-means, Hartigan's *k*-means, and spectral clustering; SDP is run only once since it is deterministic. As the ratio between $d$ and $n$ increases, Lloyd's algorithm performs worse than the others. For each algorithm, we report the values for the *k*-means loss (Equation (5)) and the NMI (see Definition A.13) corresponding to the partition that achieves the lowest *k*-means loss.

| DATASET PARAMETERS | | | | *k*-MEANS LOSS | | | | NMI | | | |
|---|---|---|---|---|---|---|---|---|---|---|---|
| | $n$ | $d$ | $K$ | LLOYD | HART. | SDP | SPEC. | LLOYD | HART. | SDP | SPEC. |
| OLIVETTI | 400 | 4096 | 40 | 161.3 | **157.3** | 165.4 | 166.2 | 0.76 | 0.80 | 0.77 | **0.83** |
| AMAZON | 1500 | 10000 | 50 | 1386.8 | **1356.3** | 1360.3 | 1363.3 | 0.42 | **0.52** | 0.44 | 0.38 |
| 20NG-A | 200 | 5000 | 2 | 193.7 | **193.5** | 193.5 | 193.5 | 0.26 | **0.54** | 0.18 | 0.52 |
| 20NG-B | 500 | 5000 | 5 | 483.8 | **481.7** | 484.3 | 482.9 | 0.25 | **0.44** | 0.39 | 0.37 |
| 20NG-C | 1000 | 5000 | 10 | 957.1 | **952.0** | 956.9 | 953.7 | 0.23 | **0.30** | 0.27 | 0.27 |
| 20NG-D | 2000 | 5000 | 20 | 1897.5 | **1887.0** | 1902.5 | 1890.0 | 0.25 | **0.29** | 0.21 | 0.26 |

Chen, X. and Yang, Y. Cutoff for exact recovery of gaussian mixture models. *IEEE Transactions on Information Theory*, 67(6):4223–4238, 2021.

Chernoff, H. A measure of asymptotic efficiency for tests of a hypothesis based on the sum of observations. *The Annals of Mathematical Statistics*, pp. 493–507, 1952.

Dempster, A. P., Laird, N. M., and Rubin, D. B. Maximum likelihood from incomplete data via the em algorithm. *Journal of the royal statistical society: series B (methodological)*, 39(1):1–22, 1977.

Diamond, S. and Boyd, S. CVXPY: A Python-embedded modeling language for convex optimization. *Journal of Machine Learning Research*, 17(83):1–5, 2016.

Ding, C. and He, X. K-means clustering via principal component analysis. In *Proceedings of the twenty-first international conference on Machine learning*, pp. 29, 2004.

Forgy, E. W. Cluster analysis of multivariate data: efficiency versus interpretability of classifications. *biometrics*, 21: 768–769, 1965.

Gao, C. and Zhang, A. Y. Iterative algorithm for discrete structure recovery. *The Annals of Statistics*, 50(2):1066–1094, 2022.

Hartigan, J. A. *Clustering algorithms*. John Wiley & Sons, Inc., 1975.

Hartigan, J. A. and Wong, M. A. Algorithm AS 136: A k-means clustering algorithm. *Journal of the Royal Statistical Society. Series C (Applied Statistics)*, 28(1):100–108, 1979.

Lam, S. K., Pitrou, A., and Seibert, S. Numba: A llvm-based python jit compiler. In *Proceedings of the Second Workshop on the LLVM Compiler Infrastructure in HPC*, pp. 1–6, 2015.

Lang, K. Newsweeder: Learning to filter netnews. In *Machine learning proceedings 1995*, pp. 331–339. Elsevier, 1995.

Liu, Z. Amazon commerce reviews, 2011. URL https://doi.org/10.24432/C55C88.

Lloyd, S. Least squares quantization in pcm. *IEEE transactions on information theory*, 28(2):129–137, 1982.

Lu, Y. and Zhou, H. H. Statistical and computational guarantees of lloyd's algorithm and its variants. *arXiv preprint arXiv:1612.02099*, 2016.

MacQueen, J. Multivariate observations. In *Proceedings ofthe 5th Berkeley Symposium on Mathematical Statisticsand Probability*, volume 1, pp. 281–297, 1967.

Manthey, B. and van Rhijn, J. Worst-Case and Smoothed Analysis of the Hartigan-Wong Method for k-Means Clustering. In Beyersdorff, O., Kanté, M. M., Kupferman, O., and Lokshtanov, D. (eds.), *41st International Symposium on Theoretical Aspects of Computer Science (STACS 2024)*, volume 289 of *Leibniz International Proceedings in Informatics (LIPIcs)*, pp. 52:1–52:16, Dagstuhl, Germany, 2024. Schloss Dagstuhl – Leibniz-Zentrum für Informatik. ISBN 978-3-95977-311-9. doi: 10.4230/LIPIcs.STACS.2024.52. URL https://drops.dagstuhl.de/entities/document/10.4230/LIPIcs.STACS.2024.52.

Manthey, B., Morawietz, N., van Rhijn, J., and Sommer, F. Complexity of Local Search for Euclidean Clustering Problems. In Mestre, J. and Wirth, A. (eds.), *35th International Symposium on Algorithms and Computation (ISAAC 2024)*, volume 322 of *Leibniz International Proceedings in Informatics (LIPIcs)*, pp. 48:1–48:16, Dagstuhl, Germany, 2024. Schloss Dagstuhl – Leibniz-Zentrum für Informatik. ISBN 978-3-95977-354-6. doi: 10.4230/LIPIcs.ISAAC.2024.48. URL https://drops.dagstuhl.de/entities/document/10.4230/LIPIcs.ISAAC.2024.48.

Mitchell, T. Twenty newsgroups, 1997. URL https://doi.org/10.24432/C5C323.

Mixon, D. G., Villar, S., and Ward, R. Clustering subgaussian mixtures by semidefinite programming, May 2016.

Mousavi, N. How tight is chernoff bound. *Unpublished manuscript*, 2010. URL https://ece.uwaterloo.ca/~nmousavi/Papers/Chernoff-Tightness.pdf.

Ndaoud, M. Sharp optimal recovery in the two component Gaussian mixture model. *The Annals of Statistics*, 50(4): 2096–2126, 2022.

Ng, A., Jordan, M., and Weiss, Y. On spectral clustering: Analysis and an algorithm. *Advances in neural information processing systems*, 14, 2001.

Pedregosa, F., Varoquaux, G., Gramfort, A., Michel, V., Thirion, B., Grisel, O., Blondel, M., Prettenhofer, P., Weiss, R., Dubourg, V., Vanderplas, J., Passos, A., Cournapeau, D., Brucher, M., Perrot, M., and Duchesnay, E. Scikit-learn: Machine learning in Python. *Journal of Machine Learning Research*, 12:2825–2830, 2011.

Peng, J. and Wei, Y. Approximating k-means-type clustering via semidefinite programming. *SIAM journal on optimization*, 18(1):186–205, 2007.

Samaria, F. S. and Harter, A. C. Parameterisation of a stochastic model for human face identification. In *Proceedings of 1994 IEEE workshop on applications of computer vision*, pp. 138–142. IEEE, 1994.

Shi, J. and Malik, J. Normalized cuts and image segmentation. *IEEE Transactions on Pattern Analysis and Machine Intelligence*, 22(8):888–905, August 2000. ISSN 1939-3539. doi: 10.1109/34.868688.

Slonim, N., Aharoni, E., and Crammer, K. Hartigan's k-means vs. lloyd's k means–is it time for a change? In *Proceedings of the 23rd International Joint Conference on Artificial Intelligence (IJCAI)*, 2013.

Steinley, D. K-means clustering: a half-century synthesis. *British Journal of Mathematical and Statistical Psychology*, 59(1):1–34, 2006.

Telgarsky, M. and Vattani, A. Hartigan's method: k-means clustering without Voronoi. In *Proceedings of the thirteenth international conference on artificial intelligence and statistics*, pp. 820–827. JMLR Workshop and Conference Proceedings, 2010.

Vattani, A. K-means Requires Exponentially Many Iterations Even in the Plane. *Discrete & Computational Geometry*, 45(4):596–616, June 2011. ISSN 1432-0444. doi: 10.1007/s00454-011-9340-1.

von Luxburg, U. A tutorial on spectral clustering. *Statistics and computing*, 17(4):395–416, 2007.

Wu, X., Kumar, V., Ross Quinlan, J., Ghosh, J., Yang, Q., Motoda, H., McLachlan, G. J., Ng, A., Liu, B., Yu, P. S., et al. Top 10 algorithms in data mining. *Knowledge and information systems*, 14(1):1–37, 2008.

Zha, H., He, X., Ding, C., Gu, M., and Simon, H. Spectral relaxation for k-means clustering. *Advances in neural information processing systems*, 14, 2001.

# Appendix

**Appendix organization.** We begin by collecting several standard results in Appendix A, for completeness and later reference. Appendix B presents the clustering algorithms studied in this work in pseudocode form. The proofs of the main results related to Lloyd's algorithm are provided in Appendix C, while the corresponding proofs for Hartigan's algorithm appear in Appendix D. Additional implementation details, including the definitions of the auxiliary clustering methods and the dataset preprocessing steps, appear in Appendix E. Finally, Appendix F reports further numerical experiments and supplementary discussion that complement the results in Section 4.

## A. Standard Results and Definitions

The following are standard textbook facts in statistics.

*Fact* A.1 (Adding Gaussian Variables). Let $\xi_1$ and $\xi_2$ be i.i.d. with a normal distribution $\xi_1, \xi_2 \sim \mathcal{N}(0, 1)$ and let $a_1, a_2, b_1, b_2 \in \mathbb{R}$. Then,

$$a_1 + b_1\xi_1 + a_2 + b_2\xi_2 \sim \mathcal{N}(a_1 + a_2, b_1^2 + b_2^2) \tag{29}$$

*Fact* A.2. Let $X \sim \mathcal{N}(0, \sigma^2)$ have a normal distribution. Let $-\infty < t < 1/2$. Then,

$$\mathbb{E}\left(\exp\left(tX^2\right)\right) = \frac{1}{\sqrt{1 - 2t}}. \tag{30}$$

*Fact* A.3 (Special Case of Cochran's Theorem). Let $X \sim \mathcal{N}(0, \sigma^2 I_d)$ be a $d$-dimensional Gaussian random vector with mean zero and covariance matrix $\sigma^2 I_d$, where $I_d$ is the $d \times d$ identity matrix. Then

$$\|X\|^2 \sim \sigma^2 \chi_d^2, \tag{31}$$

where $\chi_d^2$ denotes the chi-squared distribution with $d$ degrees of freedom.

The above facts can be used to compute the moment-generating function of the $\chi^2$ distribution.

*Fact* A.4 (The Moment Generating Function of $\chi_d^2$). Let $X \sim a\chi_d^2$. Then $\mathbb{E}(X) = da$ and $\mathrm{Var}(X) = 2da^2$. Let $t < 1/2$. Then

$$M_X(t) = \mathbb{E}\left(\exp\left(tX\right)\right) = (1 - 2t)^{-d/2}. \tag{32}$$

*Fact* A.5 (Markov's Inequality). Let $X$ be a nonnegative random variable. Then for any $a > 0$,

$$\mathbb{P}(X \geq a) \leq \frac{\mathbb{E}(X)}{a}. \tag{33}$$

*Fact* A.6 (Cauchy-Schwarz Inequality). Let $X$ and $Y$ be random variables. Then,

$$\mathbb{E}(XY)^2 \leq \mathbb{E}(X^2)\mathbb{E}(Y^2). \tag{34}$$

*Fact* A.7 (Chebyshev's Inequality). Let $X$ be an integrable random variable with finite variance $\sigma^2 > 0$ and a finite mean. Then for any $a > 0$,

$$\mathbb{P}(|X - \mathbb{E}(X)| \geq a\sigma) \leq \frac{1}{a^2}. \tag{35}$$

*Fact* A.8 (Hoeffding's Inequality). Let $X_1, X_2, \ldots, X_n$ be i.i.d. random variables with $a_i \leq X_i \leq b_i$ almost surely. Let $S = \sum_{i=1}^n X_i$. Then for any $t > 0$,

$$\mathbb{P}(|S - \mathbb{E}(S)| \geq t) \leq 2\exp\left(-\frac{2t^2}{\sum_{i=1}^n (b_i - a_i)^2}\right). \tag{36}$$

*Fact* A.9 (Counting Partitions). There are $2^n$ ways to partition $n$ elements into two labeled sets. The fraction of partitions with exactly $S$ elements in the first set (and $n - S$ in the other) is $\binom{n}{S}/2^n$. Thus, the distribution of the cluster size $S$ under a uniformly random partition is $\mathrm{Binomial}(n, 1/2)$, with mean $n/2$ and variance $n/4$. Equivalently, choosing a partition uniformly at random is the same as assigning each element independently to the first set with probability $1/2$ (and otherwise to the second set).

*Fact* A.10 (Counting Typical Partitions). The fractions of the partitions of $n$ into 2 identified sets with exactly $S$ elements in the first set (and $n - S$ in the other) are bounded by Hoeffding's inequality (Fact A.8) applied to the binomial distribution:

$$\mathbb{P}(|S - n/2| \geq q\sqrt{n}/2) \leq 2\exp\left(-\frac{q^2}{2}\right) \tag{37}$$

*More informally, for a large $n$, almost all the partitions of $n$ into 2 identified sets have about $n/2$ elements in each set.*

*Fact* A.11 (Union Bound). Let $A_1, A_2, \ldots, A_n$ be events. Then,

$$\mathbb{P}\left(\bigcup_{i=1}^{n} A_i\right) \leq \sum_{i=1}^{n} \mathbb{P}(A_i). \tag{38}$$

**Definition A.12** (Wilson's Interval for Confidence Interval of Binomial Proportions). The error bars for estimates of proportions in this paper are computed using Wilson's interval. We choose this method for computing confidence intervals as it is robust to cases where the predicted proportion is close to 1 or 0, a case where other methods for computing the confidence interval give a zero-width interval regardless of the number of samples. The confidence interval is defined as:

$$CI = (\text{center} - \text{width}, \text{center} + \text{width}) \tag{39}$$

$$\text{center} = \frac{n_s + \frac{1}{2}z_\alpha^2}{n + z_\alpha^2} \tag{40}$$

$$\text{width} = \frac{z_\alpha}{n + z_\alpha^2}\sqrt{\frac{n_s n_f}{n} + \frac{z_\alpha^2}{4}}, \tag{41}$$

where $n$ is the number of experiments, with $n_s$ and $n_f$ being the number of successes and failures, respectively. The value $z_\alpha$ is the $1 - \frac{\alpha}{2}$ for a standard normal distribution. In plots that use Wilson's interval, we plot the actual estimated ratio $n_s/n$, and omit Wilson's center.

**Definition A.13** (Normalized Mutual Information). Mutual information quantifies the dependence of two random variables. In our context, we apply it to discrete random variables, although a definition for continuous random variables also exists. Let $X, Y$ be two discrete random variables with a joint probability density function $P_{(X,Y)}$. For the discrete case, the mutual information is defined as:

$$I(X;Y) = \sum_{x \in \mathcal{Y}} \sum_{x \in \mathcal{X}} \mathbb{P}_{(X,Y)}(x,y) \log\left(\frac{\mathbb{P}_{(X,Y)}(x,y)}{\mathbb{P}_X(x)\mathbb{P}_Y(y)}\right), \tag{42}$$

where $P_X$ and $P_Y$ are the marginal probability density functions of $X$ and $Y$, respectively.

To compare the partitions obtained through $k$-means to the true partitions, we use the Normalized Mutual Information (NMI). We calculate the NMI using the implementation provided by scikit-learn, which normalizes the mutual information (Equation (42)) to range from 0 to 1, where 1 indicates perfect correlation, and 0 indicates no dependence.

## B. k-Means Algorithms

### B.1. Lloyd's $k$-Means Algorithm

Algorithm 1 is a description of Lloyd's $k$-means algorithm (Lloyd, 1982).

*Remark* B.1 (Lloyd's Algorithm Initialization). We note that there are two approaches to initializing the algorithm: using an initial partition (as discussed in this paper) or using initial guess centers. In the case of initialization based on initial centers, the first iteration of the algorithm would produce partitions: the argument in this paper proves that in the settings of this paper, the first partition produced by the algorithm is a fixed point of the algorithm (with high probability, with the possible exception of very unbalanced partitions). So, unless the initial guess is sufficiently good to produce the correct partition immediately (or, possibly, through some unusually unbalanced partitions), the algorithm will not converge to the correct partition.

---

**Algorithm 1** Lloyd's $k$-Means Algorithm

---

1: **Input:** Dataset $X = \{x_1, x_2, \ldots, x_n\}$, number of clusters $K$, initial partition $C = \{C_1, C_2, \ldots, C_K\}$ based on the indices.
2: Compute initial cluster centroids: $\widehat{\mu}_j = \frac{1}{|C_j|} \sum_{x_i \in C_j} x_i$ for $j = 1, \ldots, K$
3: **repeat**
4:     changed $\leftarrow$ false
5:     **for** each sample $i$ in dataset $X$ **do**
6:         Let $C_m$ be the cluster containing $i$
7:         **for** each cluster $C_j$ **do**
8:             Compute the distance to the cluster centroid
9:             $\Delta^2_{\text{Lloyd}}(x_i, C_j) = \|x_i - \widehat{\mu}_j\|^2$
10:         **end for**
11:         Find $j_{\text{new}} = \arg\min_j \Delta^2_{\text{Lloyd}}(x_i, C_j)$
12:         **if** $j_{\text{new}} \neq m$ **then**
13:             Move $i$ from $C_m$ to $C_{j_{\text{new}}}$
14:             changed $\leftarrow$ true
15:         **end if**
16:     **end for**
17:     Update cluster centroids: $\widehat{\mu}_j = \frac{1}{|C_j|} \sum_{x_i \in C_j} x_i$ for $j = 1, \ldots, K$
18: **until** changed = false
19: **Return:** Clusters $C_1, C_2, \ldots, C_K$ and centers $\widehat{\mu}_1, \widehat{\mu}_2, \ldots, \widehat{\mu}_K$

---

## B.2. Hartigan's $k$-Means Algorithm

Hartigan's $k$-means algorithm is described in Algorithm 2. We note that the algorithm is not defined when one of the subsets is empty; unlike Lloyd's algorithm, Hartigan's algorithm cannot reach an empty-cluster state if the initialization has no empty clusters.

Hartigan's assignment criterion based on the weighted distance in Equation (9) is, in fact, equivalent to a greedy reassignment that minimizes the $k$-means loss (Equation (5)) across all possible assignments available with the current clusters (without reassigning any other samples); Hartigan's criterion is computationally more efficient than a direct naive computation of the loss. For details, see (Hartigan, 1975). A more efficient version of the algorithm is available in (Hartigan & Wong, 1979).

---

**Algorithm 2** Hartigan's $k$-Means Algorithm

---

 1: **Input:** Dataset $X = \{x_1, x_2, \ldots, x_n\}$, number of clusters $K$, initial partition $C = \{C_1, C_2, \ldots, C_K\}$ based on the indices.
 2: Compute initial cluster centroids: $\widehat{\mu}_j = \frac{1}{|C_j|} \sum_{x_i \in C_j} x_i$ for $j = 1, \ldots, K$
 3: **repeat**
 4:     changed $\leftarrow$ false
 5:     **for** each sample $i$ in dataset $X$ **do**
 6:         Let $C_m$ be the cluster containing $i$
 7:         **if** $|C_m| > 1$ **then**
 8:             **for** each cluster $C_j$ **do**
 9:                 **if** $m = j$ **then**
10:                     Compute the Hartigan weighted distance with respect to *current* cluster $j$
11:                     $\Delta^2_{\text{Hartigan}}(x_i, C_j) = \frac{|C_j|}{|C_j|-1} \|x_i - \widehat{\mu}_j\|^2$
12:                 **else**
13:                     Compute the Hartigan weighted distance with respect to *alternative* cluster $j$
14:                     $\Delta^2_{\text{Hartigan}}(x_i, C_j) = \frac{|C_j|}{|C_j|+1} \|x_i - \widehat{\mu}_j\|^2$
15:                 **end if**
16:             **end for**
17:         Find $j_{\text{new}} = \arg\min_j \Delta^2_{\text{Hartigan}}(x_i, C_j)$
18:         **if** $j_{\text{new}} \neq m$ and $\Delta^2_{\text{Hartigan}}(x_i, C_{j_{\text{new}}}) < \Delta^2_{\text{Hartigan}}(x_i, C_m)$ **then**
19:             Move $i$ from $C_m$ to $C_{j_{\text{new}}}$
20:             Update cluster centroids $\widehat{\mu}_m$ and $\widehat{\mu}_{j_{\text{new}}}$
21:             changed $\leftarrow$ true
22:         **end if**
23:         **end if**
24:     **end for**
25: **until** changed = false
26: **Return:** Clusters $C_1, C_2, \ldots, C_K$ and centers $\widehat{\mu}_1, \widehat{\mu}_2, \ldots, \widehat{\mu}_K$

---

# C. Lloyd's Algorithm Proofs

This section contains the proofs of the theorems, lemmas, and corollaries presented in the main text related to Lloyd's algorithm, along with some additional auxiliary results and remarks. For convenience, we restate the theorems, lemmas, and corollaries before their proofs. This results in some redundancy in text and numbering—some equation numbers might seem to be out of sequence—but it may make the proofs easier to follow.

## C.1. Proof of Lemma 3.3

We restate Lemma 3.3 and provide a proof.

**Lemma 3.3.** *Fix $b_1 > b_2 > 0$, and $m \in \mathbb{R}$. Let $Y_1 \sim b_1 \chi^2_d$ and $Y_2 \sim b_2 \chi^2_d$ be scaled chi-squared distributed with $d$ degrees*

*of freedom, not necessarily independent. Then,*

$$\mathbb{P}\left(Y_1 - Y_2 \leq m\right) \leq \exp\left(m\frac{b_1 - b_2}{8b_1 b_2}\right) \rho^{d/4}, \tag{12}$$

*where $\rho = 1 - \left((b_1 - b_2)/(b_1 + b_2)\right)^2 < 1$.*

*Proof.* It holds that

$$
\begin{aligned}
&\mathbb{P}\left(Y_1 - Y_2 - m \leq 0\right) \\
&= \mathbb{P}\left(-(Y_1 - Y_2 - m) \geq 0\right) \\
&= \mathbb{P}\left(\exp\left(-t(Y_1 - Y_2 - m)\right) \geq 1\right) \quad \text{for all } t > 0.
\end{aligned} \tag{43}
$$

Using Markov's inequality (Equation (33)):

$$
\begin{aligned}
\mathbb{P}\left(Y_1 - Y_2 - m \leq 0\right) &\leq \mathbb{E}\left(\exp\left(-t(Y_1 - Y_2 - m)\right)\right) \\
&= \exp\left(tm\right) \mathbb{E}\left(\exp\left(-tY_1\right)\exp\left(tY_2\right)\right)
\end{aligned} \tag{44}
$$

Using the Cauchy-Schwarz inequality (Equation (34)):

$$\mathbb{P}\left(Y_1 - Y_2 - m \leq 0\right) \leq \exp\left(tm\right) \sqrt{\mathbb{E}\left(\exp\left(-2tY_1\right)\right)\mathbb{E}\left(\exp\left(2tY_2\right)\right)}. \tag{45}$$

Next, using Equation (32), if we further assume $-2tb_1 < 1/2$ and $2tb_2 < 1/2$ (which we will show are satisfied at the optimal $t$), we have:

$$
\begin{aligned}
\mathbb{P}\left(Y_1 - Y_2 - m \leq 0\right) &\leq \exp\left(tm\right) \sqrt{(1 + 4tb_1)^{-d/2}(1 - 4tb_2)^{-d/2}} \\
&= \exp\left(tm\right) \left((1 + 4tb_1)(1 - 4tb_2)\right)^{-d/4}
\end{aligned} \tag{46}
$$

The expression $(4b_1 t + 1)(1 - 4b_2 t)$ is maximized at

$$t_{\max} = \frac{b_1 - b_2}{8b_1 b_2}. \tag{47}$$

Using simple algebra for the expression at $t_{\max}$, we have,

$$\left((1 + 4t_{\max}b_1)(1 - 4t_{\max}b_2)\right) = \frac{(b_1 + b_2)^2}{4b_1 b_2}, \tag{48}$$

and

$$\left(\frac{(b_1 + b_2)^2}{4b_1 b_2}\right)^{-1} = 1 - \left(\frac{b_1 - b_2}{b_1 + b_2}\right)^2. \tag{49}$$

Substituting Equations (47), (48) and (49) into Equation (46), we obtain the desired result.

It remains to check that the assumptions for Equation (46) are satisfied at $t_{\max}$. The first assumption $-2tb_1 < 1/2$ holds because $b_1 > b_2 > 0$:

$$-2t_{\max}b_1 = -\frac{b_1 - b_2}{4b_2} = \frac{b_2 - b_1}{4b_2} = \frac{1}{4} - \frac{b_1}{4b_2} < 1/2. \tag{50}$$

The second assumption $2tb_2 < 1/2$ holds because $b_1 > b_2 > 0$:

$$2t_{\max}b_2 = \frac{b_1 - b_2}{4b_1} < \frac{b_1}{4b_1} < 1/2. \tag{51}$$

$\square$

*Remark* C.1 (Intuition). Under the assumptions of Lemma 3.3, we have the variables $Y_1 \sim b_1 \chi_d^2$ and $Y_2 \sim b_2 \chi_d^2$. The expected value of $Y_1$ is $\mathbb{E}(Y_1) = d \cdot b_1$ and the expected value of $Y_2$ is $\mathbb{E}(Y_2) = d \cdot b_2$. Since $b_1 > b_2$, the mean of $Y_1$ is greater than that of $Y_2$. As $d$ increases, the distributions are more concentrated around their means, and therefore we expect that the difference $Y_1 - Y_2$ will be positive with a probability that increases with $d$. The Lemma states that the probability of a negative value decreases like $\left(1 - \left(\frac{b_1 - b_2}{b_1 + b_2}\right)^2\right)^{d/4}$. In other words, since $\left(1 - \left(\frac{b_1 - b_2}{b_1 + b_2}\right)^2\right) < 1$, we can obtain any desired small probability of $Y_1 - Y_2 \leq m$ simply by increasing $d$.

## C.2. Proofs of Section 3.1: the Distribution of Distances to the Cluster Centers

### C.2.1. DISTANCE TO THE CURRENT CLUSTER'S CENTER

We restate Lemma 3.1 and provide a proof.

**Lemma 3.1** (Distance to the current cluster centroid). *Consider the setting of Model 2.1. Fix $i \in [n]$. Let $j \in \{1, 2\}$ be the (current) cluster index such that $i \in C_j$, and let $\ell \in \{1, 2\}$ be the ground-truth class index such that $i \in S_\ell^\star$. Then, the distance between the sample $x_i$ and the centroid $\widehat{\mu}_j$ of a cluster $C_j$ is distributed as*

$$\|x_i - \widehat{\mu}_j\|^2 \sim \alpha_{\mathrm{cur}} \chi_d^2 \ ,$$
$$\alpha_{\mathrm{cur}} = 2\tau^2(1 - R_j^\ell)^2 + (1 - 1/|C_j|)\, \sigma^2 \ . \tag{10}$$

*where $R_j^\ell$ is the cluster purity defined in Definition 2.4.*

*Proof.* Consider the setting of Model 2.1, and fix the true cluster assignments $\{z_i^\star\}$ as in Model 2.1(c) together with a current cluster assignment $z$ as in Definition 2.2. Without loss of generality (W.L.O.G.), assume sample $i$ which is currently assigned to cluster $C_j$, belongs to ground-truth class $\ell$, i.e. $z_i^\star = \ell$.

The difference between the sample $x_i$ and the centroid $\widehat{\mu}_j$ of cluster $C_j$ to which $x_i$ is currently assigned is:

$$x_i - \widehat{\mu}_j = \mu_\ell^\star + \xi_i - \widehat{\mu}_j \tag{52}$$

$$= \mu_\ell^\star + \xi_i - \frac{1}{|C_j|} \sum_{k \in C_j} x_k \tag{53}$$

$$= \mu_\ell^\star + \xi_i - \frac{1}{|C_j|} \sum_{k \in C_j} \left( \mu_{z_k^\star}^\star + \xi_k \right) \tag{54}$$

$$= \mu_\ell^\star - \frac{1}{|C_j|} \sum_{k \in C_j} \mu_{z_k^\star}^\star + \xi_i - \frac{1}{|C_j|} \sum_{k \in C_j} \xi_k \tag{55}$$

$$= \mu_\ell^\star - \frac{|C_j \cap S_\ell^\star|}{|C_j|} \mu_\ell^\star - \frac{|C_j \cap S_{\bar{\ell}}^\star|}{|C_j|} \mu_{\bar{\ell}}^\star + \xi_i - \frac{1}{|C_j|} \sum_{k \in C_j} \xi_k \tag{56}$$

$$= \left( 1 - \frac{|C_j \cap S_\ell^\star|}{|C_j|} \right) \mu_\ell^\star - \frac{|C_j \cap S_{\bar{\ell}}^\star|}{|C_j|} \mu_{\bar{\ell}}^\star + \xi_i - \frac{1}{|C_j|} \sum_{k \in C_j} \xi_k \tag{57}$$

Using the notation $R_j^\ell = |C_j \cap S_\ell^\star|/|C_j|$ introduced in Definition 2.4, Equation (57) becomes

$$x_i - \widehat{\mu}_j = (1 - R_j^\ell)\mu_\ell^\star - R_j^{\bar{\ell}}\mu_{\bar{\ell}}^\star + \xi_i - \frac{1}{|C_j|} \sum_{k \in C_j} \xi_k \tag{58}$$

$$= (1 - R_j^\ell)\mu_\ell^\star - (1 - R_j^\ell)\mu_{\bar{\ell}}^\star + \xi_i - \frac{1}{|C_j|} \sum_{k \in C_j} \xi_k \tag{59}$$

$$= (1 - R_j^\ell)(\mu_\ell^\star - \mu_{\bar{\ell}}^\star) + \left( 1 - \frac{1}{|C_j|} \right) \xi_i - \frac{1}{|C_j|} \sum_{k \in C_j \setminus \{i\}} \xi_k. \tag{60}$$

In Equation (59) we used $R_j^{\bar{\ell}} = 1 - R_j^\ell$, and Equation (60) follows by splitting the sum as $\sum_{k \in C_j} \xi_k = \xi_i + \sum_{k \in C_j \setminus \{i\}} \xi_k$.

By the definition of $\mu_\ell^\star$ in Model 2.1(a), the first term in Equation (60) has variance

$$\mathrm{Var}\left[ (1 - R_j^\ell)(\mu_\ell^\star - \mu_{\bar{\ell}}^\star) \right] = (1 - R_j^\ell)^2 \, \mathrm{Var}\left( \mu_\ell^\star - \mu_{\bar{\ell}}^\star \right) = 2(1 - R_j^\ell)^2 \tau^2. \tag{61}$$

Moreover, since $\xi_i$ defined in Model 2.1 and the noise terms are i.i.d. with variance $\sigma^2$, the second (noise) term in

Equation (60) satisfies

$$\text{Var}\left[\left(1 - \frac{1}{|C_j|}\right)\xi_i - \frac{1}{|C_j|}\sum_{k \in C_j \setminus \{i\}} \xi_k\right] = \left(1 - \frac{1}{|C_j|}\right)^2 \sigma^2 + \frac{1}{|C_j|^2}(|C_j| - 1)\sigma^2$$

$$= \left(1 - \frac{1}{|C_j|}\right)\sigma^2. \tag{62}$$

Therefore, by Equations (29), (61)–(62), and the independence assumptions in Model 2.1(a)–(b), the random vector in (60) is Gaussian with zero mean and isotropic covariance. In particular,

$$x_i - \widehat{\mu}_j \sim \mathcal{N}\left(0, \ \left(2(1 - R_j^\ell)^2 \tau^2 + \left(1 - \frac{1}{|C_j|}\right)\sigma^2\right) I_d\right). \tag{63}$$

Using Equation (31), we obtain the Lemma. $\qquad\square$

The following lemma identifies the case of a single misassignment at "the most favorable case" in the sense that it maximizes the value of $\alpha_{\text{cur}}$.

**Lemma C.2** ("Most favorable case")**.** *Consider the setting of Model 2.1. Fix $i \in [n]$ and let $i \in C_j \cap S_\ell^\star$, and recall $R_j^\ell$ from Definition 2.4, and $\alpha_{\text{cur}}$ from Equation (10). Then, $\alpha_{\text{cur}}$ is maximized at the smallest feasible purity $R_j^\ell = 1/|C_j|$ (equivalently $|C_j \cap S_\ell^\star| = 1$). In this case,*

$$\|x_i - \widehat{\mu}_j\|^2 \sim \omega_{\text{cur}} \chi_d^2, \qquad where \quad \omega_{\text{cur}} := \left(1 - \frac{1}{|C_j|}\right)\sigma^2 + \frac{2\tau^2(|C_j| - 1)^2}{|C_j|^2}, \tag{64}$$

*and for all feasible $R_j^\ell$, we have*

$$\alpha_{\text{cur}} \leq \omega_{\text{cur}} = \sigma^2\left(1 - \frac{1}{|C_j|}\right) + \frac{2\tau^2(|C_j| - 1)^2}{|C_j|^2}. \tag{65}$$

*Proof.* Since $i \in C_j \cap S_\ell^\star$ implies $R_j^\ell \geq 1/|C_j|$, and $r \mapsto (1 - r)^2$ is decreasing on $[0, 1]$, we get $(1 - R_j^\ell)^2 \leq (1 - 1/|C_j|)^2$, hence $\alpha_{\text{cur}} \leq \omega_{\text{cur}}$. Substituting $R_j^\ell = 1/|C_j|$ yields the expression for $\omega_{\text{cur}}$. $\qquad\square$

### C.2.2. THE DISTANCE TO A DIFFERENT CLUSTER'S CENTER

First, we consider the distribution of distances from a sample to a cluster to which it does not belong. We restate Lemma 3.2 and provide a proof.

**Lemma 3.2** (Distance to the other cluster centroid)**.** *In the settings of Lemma 3.1, the distance between the sample $x_i$ and the centroid $\widehat{\mu}_{\bar{j}}$ of a cluster $C_{\bar{j}}$ is distributed as*

$$\left\|x_i - \widehat{\mu}_{\bar{j}}\right\|^2 \sim \alpha_{\text{alt}} \chi_d^2 \ ,$$

$$\alpha_{\text{alt}} = 2\tau^2\left(1 - R_{\bar{j}}^\ell\right)^2 + \left(1 + 1/|C_{\bar{j}}|\right)\sigma^2. \tag{11}$$

*Proof.* Consider the setting of Model 2.1, and fix the true cluster assignments $\{z_i^\star\}$ as in Model 2.1(c) together with a current cluster assignment $z$ as in Definition 2.2. W.L.O.G., assume sample $i$ which is currently assigned to cluster $C_j$, belongs to ground-truth class $\ell$, i.e. $z_i^\star = \ell$, and consider cluster $C_{\bar{j}}$ which does not contain sample $i$, so $i \notin C_{\bar{j}}$.

The difference from sample $x_i$ to the centroid $\widehat{\mu}_{\bar{j}}$ of cluster $C_{\bar{j}}$ is:

$$x_i - \widehat{\mu}_{\bar{j}} = \mu_{\ell}^{\star} + \xi_i - \widehat{\mu}_{\bar{j}} \tag{66}$$

$$= \mu_{\ell}^{\star} + \xi_i - \frac{1}{|C_{\bar{j}}|} \sum_{k \in C_{\bar{j}}} x_k \tag{67}$$

$$= \mu_{\ell}^{\star} + \xi_i - \frac{1}{|C_{\bar{j}}|} \sum_{k \in C_{\bar{j}}} \left( \mu_{z_k^{\star}}^{\star} + \xi_k \right) \tag{68}$$

$$= \mu_{\ell}^{\star} - \frac{1}{|C_{\bar{j}}|} \sum_{k \in C_{\bar{j}}} \mu_{z_k^{\star}}^{\star} + \xi_i - \frac{1}{|C_{\bar{j}}|} \sum_{k \in C_{\bar{j}}} \xi_k \tag{69}$$

$$= \mu_{\ell}^{\star} - \frac{|C_{\bar{j}} \cap S_{\ell}^{\star}|}{|C_{\bar{j}}|} \mu_{\ell}^{\star} - \frac{|C_{\bar{j}} \cap S_{\bar{\ell}}^{\star}|}{|C_{\bar{j}}|} \mu_{\bar{\ell}}^{\star} + \xi_i - \frac{1}{|C_{\bar{j}}|} \sum_{k \in C_{\bar{j}}} \xi_k \tag{70}$$

$$= \left( 1 - \frac{|C_{\bar{j}} \cap S_{\ell}^{\star}|}{|C_{\bar{j}}|} \right) \mu_{\ell}^{\star} - \frac{|C_{\bar{j}} \cap S_{\bar{\ell}}^{\star}|}{|C_{\bar{j}}|} \mu_{\bar{\ell}}^{\star} + \xi_i - \frac{1}{|C_{\bar{j}}|} \sum_{k \in C_{\bar{j}}} \xi_k. \tag{71}$$

Using the notation $R_j^{\ell} = |C_j \cap S_{\ell}^{\star}|/|C_j|$ introduced in Definition 2.4, this becomes

$$x_i - \widehat{\mu}_{\bar{j}} = (1 - R_{\bar{j}}^{\ell}) \mu_{\ell}^{\star} - R_{\bar{j}}^{\bar{\ell}} \mu_{\bar{\ell}}^{\star} + \xi_i - \frac{1}{|C_{\bar{j}}|} \sum_{k \in C_{\bar{j}}} \xi_k$$

$$= (1 - R_{\bar{j}}^{\ell})(\mu_{\ell}^{\star} - \mu_{\bar{\ell}}^{\star}) + \xi_i - \frac{1}{|C_{\bar{j}}|} \sum_{k \in C_{\bar{j}}} \xi_k. \tag{72}$$

where we have used $R_{\bar{j}}^{\bar{\ell}} = 1 - R_{\bar{j}}^{\ell}$.

It follows, using Equations (29), (61) and the independence of the random variables as defined in Model 2.1 (a)–(b), that the difference is distributed as

$$x_i - \widehat{\mu}_{\bar{j}} \sim \mathcal{N}\left( 0, \left( 2(1 - R_{\bar{j}}^{\ell})^2 \tau^2 + \left( 1 + \frac{1}{|C_{\bar{j}}|} \right) \sigma^2 \right) I_d \right), \tag{73}$$

using the assumption $i \notin C_{\bar{j}}$. Using Equation (31), we obtain the claim. $\square$

**Lemma C.3** ("Most favorable case" for the other centroid). *Consider the setting of Model 2.1. Fix $i \in [n]$ and suppose $i \in C_j \cap S_{\ell}^{\star}$, and recall $R_j^{\ell}$ from Definition 2.4, and $\alpha_{\mathrm{alt}}$ from Equation (11). Then, for fixed $|C_{\bar{j}}|$, the scale $\alpha_{\mathrm{alt}}$ is minimized at the largest feasible purity $R_{\bar{j}}^{\ell} = 1$ (equivalently $C_{\bar{j}} \subseteq S_{\ell}^{\star}$). In this case,*

$$\|x_i - \widehat{\mu}_{\bar{j}}\|^2 \sim \omega_{\mathrm{alt}} \chi_d^2, \qquad \text{where} \quad \omega_{\mathrm{alt}} := \left( 1 + \frac{1}{|C_{\bar{j}}|} \right) \sigma^2, \tag{74}$$

*and for all feasible $R_{\bar{j}}^{\ell}$ we have*

$$\alpha_{\mathrm{alt}} \geq \omega_{\mathrm{alt}} = \left( 1 + \frac{1}{|C_{\bar{j}}|} \right) \sigma^2. \tag{75}$$

*Proof.* Clearly, we have, $\tau^2 \left( 1 - R_{\bar{j}}^{\ell} \right)^2 \geq 0$, and therefore

$$\alpha_{\mathrm{alt}} = 2\tau^2 \left( 1 - R_{\bar{j}}^{\ell} \right)^2 + \left( 1 + \frac{1}{|C_{\bar{j}}|} \right) \sigma^2 \geq \left( 1 + \frac{1}{|C_{\bar{j}}|} \right) \sigma^2 = \omega_{\mathrm{alt}}. \tag{76}$$

Substituting $R_{\bar{j}}^{\ell} = 1$ yields the stated expression for $\omega_{\mathrm{alt}}$, and the claimed distribution follows from $\|x_i - \widehat{\mu}_{\bar{j}}\|^2 \sim \alpha_{\mathrm{alt}} \chi_d^2$ (Equation (11)). $\square$

## C.3. Proof of Theorem 3.4

We restate Theorem 3.4 and provide a proof.

**Theorem 3.4** (Lloyd's algorithm: single sample). *We consider the setting of Model 2.1, a fixed $i \in [n]$ with current cluster $C_j$ and other cluster $C_{\bar{j}}$ such that $i \in C_j$. We denote the cluster sizes $c := |C_j|$ and $\bar{c} := |C_{\bar{j}}|$.*

*If the noise level $\sigma > 0$ satisfies*

$$\sigma > \frac{\sqrt{2\bar{c}}\,\tau\,(c-1)}{\sqrt{c(c+\bar{c})}}, \tag{13}$$

*then, we have*

$$\mathbb{P}\Big(\big\|x_i - \widehat{\mu}_{\bar{j}}\big\|^2 < \big\|x_i - \widehat{\mu}_j\big\|^2\Big) \leq \rho^{d/4}, \tag{14}$$

*where $\rho$ is defined by*

$$\rho(\sigma,\tau,c,\bar{c}) =$$
$$\frac{4\sigma^2(c-1)c^2\bar{c}(\bar{c}+1)\big(c(\sigma^2+2\tau^2)-2\tau^2\big)}{(-c(\sigma^2+4\tau^2)\bar{c}+c^2(\sigma^2+2(\sigma^2+\tau^2)\bar{c})+2\tau^2\bar{c})^2}, \tag{15}$$

*and satisfies $0 \leq \rho < 1$.*

*Proof.* Substituting $b_1 = \alpha_{\text{alt}}$ (Equation (11)) and $b_2 = \alpha_{\text{cur}}$ (Equation (10)) into Lemma 3.3, would yield an expression that is somewhat more complicated and depends on additional parameters. We observe that the expression $\rho = 1 - \left(\frac{b_1 - b_2}{b_1 + b_2}\right)^2$ in Lemma 3.3 increases when $b_1$ decreases and when $b_2$ increases since

$$\frac{\partial}{\partial b_1}\left\{-\left(\frac{b_1 - b_2}{b_1 + b_2}\right)^2\right\} = -2\frac{b_1 - b_2}{b_1 + b_2}\frac{2b_2}{(b_1 + b_2)^2} < 0, \tag{77}$$

as well as

$$\frac{\partial}{\partial b_2}\left\{-\left(\frac{b_1 - b_2}{b_1 + b_2}\right)^2\right\} = 4\frac{(b_1 - b_2)b_1}{(b_1 + b_2)^3} > 0. \tag{78}$$

Owing to the monotonicity properties of Equations (77) and (78), we can replace $b_1$ and $b_2$ by their "most favorable case" values $\omega_{\text{alt}} \leq \alpha_{\text{alt}}$ and $\omega_{\text{cur}} \geq \alpha_{\text{cur}}$ (Equations (75) and (65) in the auxiliary lemmas ), and obtain a bound that is *looser* than the one we would have obtained by substituting $b_1$ and $b_2$. In particular, plugging-in $b_1 = \omega_{\text{alt}}$ and $b_2 = \omega_{\text{cur}}$ into $\rho = 1 - \left(\frac{b_1 - b_2}{b_1 + b_2}\right)^2$ gives

$$\mathbb{P}\big(\|x_i - \widehat{\mu}_{\bar{j}}\|^2 < \|x_i - \widehat{\mu}_j\|^2\big) \leq \rho^{d/4}, \tag{79}$$

with

$$\rho = \frac{4\sigma^4(1 + |C_{\bar{j}}|^{-1})(1 - |C_j|^{-1}) + 8\tau^2\sigma^2(|C_j| - 1)^2(1 + |C_{\bar{j}}|^{-1})|C_j|^{-2}}{\Big(\sigma^2(1 + |C_{\bar{j}}|^{-1}) + \sigma^2(1 - |C_j|^{-1}) + 2\tau^2(|C_j| - 1)^2|C_j|^{-2}\Big)^2}. \tag{80}$$

which is exactly Equation (15).

The requirement of Lemma 3.3 that $b_1 > b_2 > 0$ (with $m = 0$) is satisfied by the condition on the noise given in Equation (13), as the latter is crafted for that purpose. Concretely, Equation (13) corresponds to the condition that $\omega_{\text{alt}} > \omega_{\text{cur}}$, and hence the requirement $b_1 > b_2 > 0$ in Lemma 3.3 holds. $\qquad\square$

## C.4. Proof of Theorem 3.6

We now restate Theorem 3.6 and provide a proof.

**Theorem 3.6** (Uniform bound for $q$-approximately balanced partitions). *Consider the setting of Model 2.1, with a fixed $i \in [n]$ with current cluster index $j$ and other cluster $\bar{j}$. Fix a partition imbalance factor $q > 1$ and assume a partition $\mathcal{P} = \{C_1, C_2\}$ that is $q$-approximately balanced (Definition 2.5). Fix $\tau = 1$, and fix $\beta > 1$, such that*

$$\sigma = \beta\frac{(\sqrt{n}\,q + n - 2)}{\sqrt{2}\sqrt{\sqrt{n}\,q + n}}. \tag{16}$$

*Then,*

$$\mathbb{P}\left(\left\|x_i - \widehat{\mu}_{\bar{j}}\right\|^2 - \left\|x_i - \widehat{\mu}_j\right\|^2 < 0\right) \le \rho_q^{d/4}, \tag{17}$$

*where*

$$\rho_q = \frac{\sigma^2\left(\sqrt{n}q + n - 2\right)\left(\sqrt{n}q + n\right)\left(\sqrt{n}q + n + 2\right)\left(\sqrt{n}\left(\sigma^2 + 2\right)\left(\sqrt{n} + q\right) - 4\right)}{\left(n\sigma^2\left(\sqrt{n} + q\right)^2 + \left(\sqrt{n}q + n - 2\right)^2\right)^2}. \tag{18}$$

*Proof.* This theorem aims to reproduce a version of Theorem 3.4 that applies uniformly to all $q$-approximately balanced partitions. The idea is to derive the threshold noise in Equation (16) (with $\beta = 1$) which is larger than the threshold in Equation (13) in Theorem 3.4 for cluster sizes in $q$-approximately balanced partitions, and similarly to derive the $\rho_q$ in Equation (18) which is larger than $\rho$ in Equation (15) in Theorem 3.4 for cluster sizes in $q$-approximately balanced partitions. To do this, we will partially repeat the proof of Theorem 3.4 from its components with small modifications.

We recall from the proof of Theorem 3.4 that $\rho$ is obtained by substituting $b_1 = \omega_{\text{alt}}$ (Equation (75)) and $b_2 = \omega_{\text{cur}}$ (Equation (65)) into Lemma 3.3, yielding Equation (80). We will repeat the derivation while generalizing it to $q$-approximately balanced partitions.

Substituting the maximum values of $|C_{\bar{j}}| = n/2 + q\sqrt{n/4}$ and $|C_j| = n/2 + q\sqrt{n/4}$ into $b_1 = \omega_{\text{alt}}$ (Equation (75)) and $b_2 = \omega_{\text{cur}}$ (Equation (65)) and then into Equation (12) of Lemma 3.3 yields Equation (18); it remains to show that $\rho_q$ yields an upper bound on the value of $\rho$. To establish this, we will show that the expression in Equation (15) is monotone in both $|C_j|$ and $|C_{\bar{j}}|$ in the range $\frac{n}{2} - q\sqrt{\frac{n}{4}} < |C_k| < \frac{n}{2} + q\sqrt{\frac{n}{4}}$, for $k \in \{1, 2\}$, even *without* requiring $n = |C_j| + |C_{\bar{j}}|$. This fact enables us to replace $|C_{\bar{j}}|$ and $|C_j|$ by upper bounds.

We then need to show that the choice of the noise in Equation (16), which does not depend on any cluster size, is valid. In particular, as the noise plays a role in establishing the monotonicity of $\rho$, it is important to show that substituting the maximum values of $|C_{\bar{j}}| = n/2 + q\sqrt{n/4}$ and $|C_j| = n/2 + q\sqrt{n/4}$ into $b_1 = \omega_{\text{alt}}$ (Equation (75)) and $b_2 = \omega_{\text{cur}}$ (Equation (65)) leaves the monotonicity statement unscathed.

To establish this, we will show that the cluster size-dependent lower bound for $\sigma$ in Equation (13) is monotone in both $|C_j|$ and $|C_{\bar{j}}|$ in the relevant domain. It can thus be maximized and compared with the value

$$\beta \frac{\left(\sqrt{n}\, q + n - 2\right)}{\sqrt{2}\sqrt{\sqrt{n}\, q + n}} \tag{81}$$

which we fixed in the statement of Theorem 3.6.

Monotonicity of $\rho$. We will compute the derivative of $\rho$ with respect to $|C_{\bar{j}}|$ and $|C_j|$. To simplify our approach, we rely on the chain rule, building up on the fact that $\rho$ comes from plugging in $b_1 = \omega_{\text{alt}}$ (Equation (75)) and $b_2 = \omega_{\text{cur}}$ (Equation (65)) into Lemma 3.3. This enables us to keep shorter, more transparent expressions. Recall from the proof of Theorem 3.4 that, for $i \in \{1, 2\}$,

$$\partial_{b_i}\rho(b_1, b_2) = \partial_{b_i}\left(1 - \frac{(b_1 - b_2)^2}{(b_1 + b_2)^2}\right) = \frac{4(b_{\bar{i}} - b_i)b_{\bar{i}}}{(b_1 + b_2)^3}, \tag{82}$$

with $\bar{i} = 2$ if $i = 1$ and vice versa. Further, using the exact form of $\omega_{\text{alt}}$ and $\omega_{\text{cur}}$,

$$\partial_{|C_{\bar{j}}|}\omega_{\text{alt}}(|C_{\bar{j}}|) = \partial_{|C_{\bar{j}}|}\left(1 + |C_{\bar{j}}|^{-1}\right)\sigma^2 = \frac{-\sigma^2}{|C_{\bar{j}}|^{-2}} \tag{83}$$

and

$$\partial_{|C_{\bar{j}}|}\omega_{\text{cur}}(|C_{\bar{j}}|) = \partial_{|C_j|}\left\{\sigma^2(1 - |C_j|^{-1}) + 2\tau^2(|C_j| - 1)^2|C_j|^{-2}\right\} = |C_j|^{-2}(\sigma^2 + 4\tau^2 - 4\tau^2|C_j|^{-1}). \tag{84}$$

These are all the parts needed for the chain rule as

$$\partial_{|C_{\bar{j}}|}\rho = \partial_{\omega_{\text{alt}}}\rho(\omega_{\text{cur}}, \omega_{\text{alt}})\,\partial_{|C_{\bar{j}}|}\omega_{\text{alt}}(|C_{\bar{j}}|). \tag{85}$$

As $\partial_{|C_{\bar{j}}|}\omega_{\text{alt}}(|C_{\bar{j}}|) = -\sigma^2|C_{\bar{j}}|^2$ is negative, to prove the fact that $\rho$ increases as a function of $|C_{\bar{j}}|$, we need that $\partial_{\omega_{\text{alt}}}\rho(\omega_{\text{cur}}, \omega_{\text{alt}})$ is negative as well. From (82) this translates into the condition

$$\sigma^2(1 - |C_j|^{-1}) + 2\tau^2(|C_j| - 1)^2|C_j|^{-2} < \sigma^2(1 + |C_{\bar{j}}|^{-1}). \tag{86}$$

A simple reorganization yields,

$$\frac{2\tau^2(|C_j| - 1)^2 |C_j|^{-2}}{|C_{\bar{j}}|^{-1} + |C_j|^{-1}} < \sigma^2. \tag{87}$$

Note that this was the previously stated condition on the variance. However, in the context of the present theorem, we aim at having this bound hold for any $q$-approximately balanced partition. Therefore, we want to find the worst case scenario in terms of $|C_j|$ and $|C_{\bar{j}}|$ and show that the choice of $\sigma$ in the statement of the theorem is sufficient.

Let us now prove the fact that $\rho$ increases as a function of $|C_j|$. The chain rule in that case is

$$\partial_{|C_j|}\rho = \partial_{\omega_{\mathrm{cur}}}\rho(\omega_{\mathrm{cur}}, \omega_{\mathrm{alt}}) \, \partial_{|C_j|}\omega_{\mathrm{cur}}(|C_j|). \tag{88}$$

We note that $\partial_{|C_j|}\omega_{\mathrm{cur}}(|C_j|) = |C_j|^{-2}(\sigma^2 + 4\tau^2 - 4\tau^2|C_j|^{-1})$ is positive, as $1 - |C_j|^{-1} > 0$. As the difference appearing in the numerator of Equation (82) for the case $\partial_{\omega_{\mathrm{cur}}}\rho(\omega_{\mathrm{cur}}, \omega_{\mathrm{alt}})$ is the opposite of that of $\partial_{\omega_{\mathrm{alt}}}\rho(\omega_{\mathrm{cur}}, \omega_{\mathrm{alt}})$, one has $\partial_{\omega_{\mathrm{cur}}}\rho(\omega_{\mathrm{cur}}, \omega_{\mathrm{alt}}) > 0$ thanks to the condition in Equation (87) again. This yields the sought monotonicity.

Monotonicity of $\sigma$'s lower bound. As it is clear from (87) that the lower bound on $\sigma$ increases as a function of $|C_{\bar{j}}|$, we turn to the other argument. Compute

$$\partial_{|C_j|}\frac{(|C_j| - 1)^2 |C_j|^{-2}}{|C_{\bar{j}}|^{-1} + |C_j|^{-1}} = \frac{\big(2(|C_j| - 1)|C_j|^{-2} - 2(|C_j| - 1)^2|C_j|^{-3}\big)\big(|C_{\bar{j}}|^{-1} + |C_j|^{-1}\big) + (|C_j| - 1)^2|C_j|^{-4}}{(|C_{\bar{j}}|^{-1} + |C_j|^{-1})^2}$$

and observe that

$$\big(2(|C_j| - 1)|C_j|^{-2} - 2(|C_j| - 1)^2|C_j|^{-3}\big)\big(|C_{\bar{j}}|^{-1} + |C_j|^{-1}\big) + (|C_j| - 1)^2|C_j|^{-4}$$
$$= (|C_j| - 1)|C_j|^{-2}\Big(2|C_{\bar{j}}|^{-1} + 2|C_j|^{-1} - (|C_j| - 1)|C_j|^{-1}\big(2|C_{\bar{j}}|^{-1} + 2|C_j|^{-1} - |C_j|^{-1}\big)\Big)$$
$$= (|C_j| - 1)|C_j|^{-2}\Big(|C_j|^{-1} + (|C_j| - 1)|C_j|^{-1}\big(2|C_{\bar{j}}|^{-1} + |C_j|^{-1}\big)\Big) > 0.$$

We thus have shown that, under the assumption of approximately balanced partitions, recall Definition 2.5, substituting the maximum values of $|C_{\bar{j}}| = n/2 + q\sqrt{n/4}$ and $|C_j| = n/2 + q\sqrt{n/4}$ into Equations (13) and (15) yield an upper bound and further yields the expressions in the current theorem.

We note that slightly better expressions can be obtained by maximizing $\rho$ and $\sigma$ within the range of $|C_{\bar{j}}|$ and $|C_j|$, and not maximizing the values of $|C_{\bar{j}}|$ and $|C_j|$ separately. $\qquad\square$

*Remark* C.4. We emphasize that our statements exclude *exceptionally unbalanced* partitions in a precise sense: we require each cluster to contain at least two samples. Therefore, the imbalance tolerance parameter $q$ could grow with $n$. In that case, Fact A.10 yields

$$\mathbb{P}(|S - n/2| \geq q\sqrt{n}/2) \leq 2\exp\left(-\frac{q^2}{2}\right) \tag{89}$$

would ensure that one covers (as $n \to \infty$) a fraction of the partitions converging to 1.

To ensure that we still do not take $q$ too large (to enforce the requirement that the partitions should contain at least 2 samples), taking $q \leq 0.6\sqrt{n}$ would suffice (asymptotically as $n \to \infty$). Indeed, the order of the number of partitions of size at most 2 is $O(n^2)$; their proportion is thus around $O(n^2 2^{-n})$. One can then use an anti-concentration result like that of Mousavi (2010) claiming that

$$\mathbb{P}\left((S_n - n/2) > t\right) \geq \frac{1}{4}\exp(-2t^2/n). \tag{90}$$

It remains to plug in our choice $q \leq 0.6\sqrt{n}$ in the formula above to see that the problematic partitions are correctly avoided (asymptotically). Further remark that $S_n - n/2$ is symmetric around zero.

## C.5. Proof of Corollary 3.8

We restate Corollary 3.8 and provide a proof.

**Corollary 3.8** (Main result: Lloyd's algorithm). *Consider the setting of Model 2.1. Fix an imbalance parameter $q > 1$ and assume the noise level satisfies Equation (16). Then the probability that there exists a $q$-approximately balanced partition (see Definition 2.5) that is* not *a fixed point of Lloyd's $k$-means update scheme is upper bounded by*

$$\mathbb{P}\left(\exists \; approx. \; balanced \; partition \; that \; is \; not \; a \; fixed \; point\right)$$
$$\leq 2^n n \rho_q^{d/4} \;, \tag{21}$$

*where $\rho_q$ is defined in (18).*

*Proof.* First, we consider a single partition $\mathcal{P}$ that satisfies the conditions of the corollary. By the union bound (Equation (38)), we have

$$\mathbb{P}\left(\exists i \in [n] : \left\|x_i - \widehat{\mu}_{\overline{j}}\right\|^2 < \left\|x_i - \widehat{\mu}_j\right\|^2 | \mathcal{P}\right) \leq \sum_{i=1}^n \mathbb{P}\left(\left\|x_i - \widehat{\mu}_{\overline{j}}\right\|^2 < \left\|x_i - \widehat{\mu}_j\right\|^2 | \mathcal{P}\right). \tag{91}$$

Substituting Equation (17), we obtain

$$\mathbb{P}\left(\exists i \in [n] : \left\|x_i - \widehat{\mu}_{\overline{j}}\right\|^2 < \left\|x_i - \widehat{\mu}_j\right\|^2 | \mathcal{P}\right) \leq n \rho_q^{d/4} \;. \tag{92}$$

Next, we consider all partitions that satisfy the conditions of the corollary. Using the union bound again, we have

$$\mathbb{P}\left(\exists i \in [n], \mathcal{P} : \left\|x_i - \widehat{\mu}_{\overline{j}}\right\|^2 < \left\|x_i - \widehat{\mu}_j\right\|^2\right) \leq \sum_{\mathcal{P}} \mathbb{P}\left(\exists i \in [n] : \left\|x_i - \widehat{\mu}_{\overline{j}}\right\|^2 < \left\|x_i - \widehat{\mu}_j\right\|^2 | \mathcal{P}\right) \leq \sum_{\mathcal{P}} n \rho_q^{d/4} \;. \tag{93}$$

Recalling Fact A.9, there are at most $2^n$ possible partitions of the samples and therefore at most $2^n$ partitions that satisfy the requirements of the theorem, which yields the desired result. □

## D. Hartigan's Algorithm Proofs

This section contains the proofs of the theorems, lemmas, and corollaries presented in the main text related to Hartigan's algorithm, along with some additional auxiliary results and remarks.

### D.1. Auxiliary Lemmas: The Distribution of Hartigan Weighted Distances

Applying the Hartigan weighted distance in Equation (9) to the distances in Lemma 3.1 and Lemma 3.2 immediately yields the following lemmas.

**Corollary D.1** (Hartigan Weighted Distance to a Different Cluster). *In the setting of Lemma 3.2, consider a sample $x_i \in S_\ell^\star$ from ground-truth class $\ell$, and let $C_{\overline{j}}$ be the cluster to which $x_i$ is* not *currently assigned ($i \notin C_{\overline{j}}$). Then, the Hartigan weighted distance defined in Equation (9) is distributed as*

$$\Delta_H^2(x_i, C_{\overline{j}}) = \frac{\|x_i - \hat{\mu}_{\overline{j}}\|^2}{1 + 1/|C_{\overline{j}}|} \sim \eta_{C_{\overline{j}}} \chi_d^2, \tag{94}$$

*with the scale parameter*

$$\eta_{C_{\overline{j}}} = 2\tau^2 \frac{|C_{\overline{j}}|}{|C_{\overline{j}}| + 1}(1 - R_{\overline{j}}^\ell)^2 + \sigma^2. \tag{95}$$

*The randomization here is over the noise and ground-truth class centers, as specified in Model 2.1(a)–(b).*

**Corollary D.2** (Hartigan Weighted Distance to the Current Cluster). *In the setting of Lemma 3.1, consider the weighted distance between the sample $x_i \in S_\ell^\star$ and the centroid of the cluster $C_j$ to which it is* currently assigned *($i \in C_j$). The Hartigan weighted distance defined in Equation (9) is distributed as*

$$\Delta_H^2(x_i, C_j) = \frac{|C_j|}{|C_j| - 1} \|x_i - \widehat{\mu}_j\|^2 \sim \eta_{C_j} \chi_d^2, \tag{96}$$

*with the scale parameter*

$$\eta_{C_j} = 2\tau^2 \frac{|C_j|}{|C_j| - 1}(1 - R_j^\ell)^2 + \sigma^2. \tag{97}$$

*The randomization here is again over the noise and ground-truth class centers, as specified in Model 2.1(a)–(b).*

*Observation* D.3. Based on Corollary D.2 and Corollary D.1, the expected values of the Hartigan weighted squared distances are

$$\mathbb{E}\Delta_H^2(x_i, C_j) = \begin{cases} d\left(2\tau^2 \frac{|C_j|}{|C_j|-1}(1-R_j^\ell)^2 + \sigma^2\right) & \text{if } i \in C_j \\ d\left(2\tau^2 \frac{|C_j|}{|C_j|+1}(1-R_j^\ell)^2 + \sigma^2\right) & \text{otherwise} \end{cases} \tag{98}$$

Interestingly, the expected difference between the weighted squared distance to the current cluster and the weighted squared distance to another cluster does not depend on the noise level $\sigma$. Indeed,

$$\mathbb{E}\left(\Delta_H^2(x_i, C_j) - \Delta_H^2(x_i, C_{\bar{j}})\right) = \mathbb{E}\left(\Delta_H^2(x_i, C_j)\right) - \mathbb{E}\left(\Delta_H^2(x_i, C_{\bar{j}})\right)$$

$$= 2\tau^2 d\left(\frac{|C_j|}{|C_j|-1}(1-R_j^\ell)^2 - \frac{|C_{\bar{j}}|}{|C_{\bar{j}}|+1}(1-R_{\bar{j}}^\ell)^2\right). \tag{99}$$

This property does not hold for the unweighted distances or other weights that might appear natural. Recalling that $\chi_d^2$ becomes concentrated around the mean as $d$ grows, a closer look at Equation (99) suggests that a sample would tend to be reassigned by Hartigan's algorithm from the current cluster to the other cluster if the other cluster has a higher ratio of samples from the sample's ground-truth class. This observation is made more rigorous in the subsequent steps of the proof.

## D.2. Proof of Theorem 3.9

We restate Theorem 3.9 and provide a proof.

**Theorem 3.9** (Hartigan's algorithm: single sample). *In the setting of Model 2.1, fix an index $i \in [n]$ and let $\ell \in \{1, 2\}$ be its ground-truth class (so $i \in S_\ell^\star$). Let $j \in \{1, 2\}$ be its current cluster index (so $i \in C_j$) and let $\bar{j}$ denote the other cluster. If the cluster purity coefficient (Definition 2.4) satisfies*

$$0 < R_j^\ell \leq R_{\bar{j}}^\ell \leq 1, \tag{22}$$

*then,*

$$\mathbb{P}\left(\Delta_H^2(x_i, C_j) \leq \Delta_H^2(x_i, C_{\bar{j}})\right) \leq \rho^{d/4}, \tag{23}$$

*where $\Delta_H$ is defined in Equation (9) and $\rho$ is given by*

$$\rho = 1 - \left(\frac{\tau^2\left(\frac{|C_j|}{|C_j|-1}(1-R_j^\ell)^2 - \frac{|C_{\bar{j}}|}{|C_{\bar{j}}|+1}(1-R_{\bar{j}}^\ell)^2\right)}{\tau^2\left(\frac{|C_j|}{|C_j|-1}(1-R_j^\ell)^2 + \frac{|C_{\bar{j}}|}{|C_{\bar{j}}|+1}(1-R_{\bar{j}}^\ell)^2\right) + \sigma^2}\right)^2. \tag{24}$$

*and satisfies $0 \leq \rho < 1$.*

*Proof.* By Corollary D.2 and Corollary D.1,

$$\Delta_H^2(x_i, C_j) \sim b_1 \chi_d^2, \qquad \Delta_H^2(x_i, C_{\bar{j}}) \sim b_2 \chi_d^2, \tag{100}$$

with

$$b_1 = 2\tau^2 \frac{|C_j|}{|C_j|-1}(1-R_j^\ell)^2 + \sigma^2, \qquad b_2 = 2\tau^2 \frac{|C_{\bar{j}}|}{|C_{\bar{j}}|+1}(1-R_{\bar{j}}^\ell)^2 + \sigma^2, \tag{101}$$

Since $\frac{|C_j|}{|C_j|-1} > 1$ and $\frac{|C_{\bar{j}}|}{|C_{\bar{j}}|+1} < 1$, and in light of the assumption in Equation (22), we can conclude that $b_1 - b_2 > 0$. Indeed,

$$b_1 - b_2 = 2\tau^2 \frac{|C_j|}{|C_j|-1}(1-R_j^\ell)^2 - 2\tau^2 \frac{|C_{\bar{j}}|}{|C_{\bar{j}}|+1}(1-R_{\bar{j}}^\ell)^2$$

$$> 2\tau^2\left[(1-R_j^\ell)^2 - (1-R_{\bar{j}}^\ell)^2\right]$$

$$= 2\tau^2\left[\left(2 - R_j^\ell - R_{\bar{j}}^\ell\right)\left(R_{\bar{j}}^\ell - R_j^\ell\right)\right] \geq 0.$$

Therefore, the distributions satisfy the requirements of Lemma 3.3. Substituting (101) into Lemma 3.3 yields

$$\mathbb{P}\left(\Delta_H^2(x_i, C_j) - \Delta_H^2(x_i, C_{\bar{j}}) < 0\right) \leq \rho^{d/4} \tag{102}$$

where

$$\rho = 1 - \left(\frac{b_1 - b_2}{b_1 + b_2}\right)^2 = 1 - \left(\frac{\tau^2\left[\frac{|C_j|}{|C_j|-1}(1 - R_j^\ell)^2 - \frac{|C_{\bar{j}}|}{|C_{\bar{j}}|+1}(1 - R_{\bar{j}}^\ell)^2\right]}{\tau^2[\frac{|C_j|}{|C_j|-1}(1 - R_j^\ell)^2 + \frac{|C_{\bar{j}}|}{|C_{\bar{j}}|+1}(1 - R_{\bar{j}}^\ell)^2] + \sigma^2}\right)^2 \tag{103}$$

and $\rho < 1$. $\qquad\square$

A direct consequence of Theorem 3.9 is the following corollary.

**Corollary 3.10.** *If the conditions of Theorem 3.9 are satisfied, then the probability that the partition $\mathcal{P}$ is a fixed point of Hartigan's algorithm is bounded by*

$$\mathbb{P}(\mathcal{P} \text{ is a fixed point}) \leq \rho^{d/4}, \tag{25}$$

*where $\rho$ is given in Equation (24).*

### D.3. Proof of Corollary 3.11

We restate Corollary 3.11 and provide a proof.

**Corollary 3.11.** *Consider the setting of Model 2.1. Further assume that $n \geq 4$. If the current partition $\mathcal{P}$ is nonempty and is an incorrect partition (Definition 2.4), then, the probability that $\mathcal{P}$ is a fixed point is bounded by*

$$\mathbb{P}(\mathcal{P} \text{ is a fixed point}) \leq \rho_h^{d/4}, \tag{26}$$

*where $\rho_h$ is given by*

$$\rho_h = 1 - \left(\frac{4\tau^2(R^\star)^2 n^{-1}}{3\tau^2 + \sigma^2}\right)^2 < 1, \tag{27}$$

*and $R^\star = \min(R^1, R^2)$ is the relative size of the smallest ground-truth class.*

*Proof.* We recall the proportions notation from Definition 2.4,

$$R_j^\ell = \frac{|S_\ell^\star \cap C_j|}{|C_j|}, \quad R^\ell = \frac{|S_\ell^\star|}{n}. \tag{104}$$

**Step 1: Choice of $C_j$ and $S_\ell^\star$.** Since $\mathcal{P}$ is not equal to the ground-truth partition, at least one cluster is not pure. Hence, there exists $j \in \{1, 2\}$ such that $0 < R_j^1 < 1$ and $0 < R_j^2 < 1$. Fix such an index $j$. Moreover, because $R_j^1 + R_j^2 = 1$ and $R^1 + R^2 = 1$, it cannot be that $R_j^1 > R^1$ and $R_j^2 > R^2$ simultaneously. Therefore, for this (mixed) cluster $C_j$ there exists an index $\ell \in \{1, 2\}$ such that

$$R_j^\ell \leq R^\ell. \tag{105}$$

We henceforth fix such an $\ell$ and take a sample $i \in C_j \cap S_\ell^\star$ accordingly, with $C_j \cap S_\ell^\star \neq \emptyset$ (which holds since $C_j$ contains points from both classes).

**Step 2: Expression of $R_{\bar{j}}^\ell$.** For convenience, set $n_j := |C_j|$ and $n_{\bar{j}} := |C_{\bar{j}}|$, so that $n = n_j + n_{\bar{j}}$.

By Definition 2.4, we have $|S_\ell^\star| = R^\ell n$ and $|C_j \cap S_\ell^\star| = R_j^\ell n_j$. Since

$$|C_{\bar{j}} \cap S_\ell^\star| = |S_\ell^\star| - |C_j \cap S_\ell^\star| = R^\ell n - R_j^\ell n_j, \tag{106}$$

it follows that

$$R_{\bar{j}}^\ell = \frac{|C_{\bar{j}} \cap S_\ell^\star|}{n_{\bar{j}}} = \frac{R^\ell n - R_j^\ell n_j}{n - n_j}. \tag{107}$$

We note that by Equation (107), the choice of Equation (105) implies $R_{\bar{j}}^\ell \geq R^\ell$ (since $n - n_j > 0$). This pair of inequalities,

$$R_j^\ell \leq R^\ell \leq R_{\bar{j}}^\ell, \tag{108}$$

will be used below.

**Step 3: Derivation of $\rho$.** The sample $i$ satisfies the conditions of Theorem 3.9 and Corollary 3.10, and in particular satisfies Equation (22) due to Equation (108). Therefore, the probability that the partition is a fixed point also satisfies Equation (25) with

$$
\begin{aligned}
\rho &= 1 - \left( \frac{\tau^2 \left( \frac{|C_j|}{|C_j|-1}(1-R_j^\ell)^2 - \frac{|C_{\bar{j}}|}{|C_{\bar{j}}|+1}(1-R_{\bar{j}}^\ell)^2 \right)}{\tau^2 \left( \frac{|C_j|}{|C_j|-1}(1-R_j^\ell)^2 + \frac{|C_{\bar{j}}|}{|C_{\bar{j}}|+1}(1-R_{\bar{j}}^\ell)^2 \right) + \sigma^2} \right)^2 \\
&= 1 - \left( \frac{\tau^2 \left( \frac{n_j}{n_j-1}(1-R_j^\ell)^2 - \frac{n_{\bar{j}}}{n_{\bar{j}}+1}(1-\frac{R^\ell n - R_j^\ell n_j}{n-n_j})^2 \right)}{\tau^2 \left( \frac{n_j}{n_j-1}(1-R_j^\ell)^2 + \frac{n_{\bar{j}}}{n_{\bar{j}}+1}(1-\frac{R^\ell n - R_j^\ell n_j}{n-n_j})^2 \right) + \sigma^2} \right)^2 \\
&= 1 - \left( \frac{\tau^2 \left[ \left(\frac{n_j}{n_j-1}\right)(1-R_j^\ell)^2 - \frac{n-n_j}{n-n_j+1}\left(1-\frac{R^\ell n - R_j^\ell n_j}{n-n_j}\right)^2 \right]}{\tau^2 \left[ \frac{n_j}{n_j-1}(1-R_j^\ell)^2 + \frac{n-n_j}{n-n_j+1}\left(1-\frac{R^\ell n - R_j^\ell n_j}{n-n_j}\right)^2 \right] + \sigma^2} \right)^2 .
\end{aligned}
\tag{109}
$$

**Step 4: Upper bound for $\rho$.** First, we observe that the denominator in Equation (109) satisfies

$$
\tau^2 \left( \frac{|C_j|}{|C_j|-1}(1-R_j^\ell)^2 + \frac{|C_{\bar{j}}|}{|C_{\bar{j}}|+1}(1-R_{\bar{j}}^\ell)^2 \right) + \sigma^2 \le 3\tau^2 + \sigma^2 .
\tag{110}
$$

Next, we consider the numerator in (109).

Then, together with the fact that $R_j^\ell \le R^\ell \le R_{\bar{j}}^\ell$ in Equation (108), we obtain a lower bound for the numerator

$$
\begin{aligned}
\tau^2 \left( \frac{|C_j|}{|C_j|-1}(1-R_j^\ell)^2 - \frac{|C_{\bar{j}}|}{|C_{\bar{j}}|+1}(1-R_{\bar{j}}^\ell)^2 \right) &\ge \tau^2 \left( \frac{|C_j|}{|C_j|-1}(1-R^\ell)^2 - \frac{|C_{\bar{j}}|}{|C_{\bar{j}}|+1}(1-R^\ell)^2 \right) \\
&= \tau^2 \left( \frac{n_j}{n_j-1}(1-R^\ell)^2 - \frac{n-n_j}{n-n_j+1}(1-R^\ell)^2 \right) \\
&= \tau^2(1-R^\ell)^2 \frac{n}{(n_j-1)(n-n_j+1)} \\
&\ge \tau^2(1-R^\ell)^2 \frac{4n}{n^2}.
\end{aligned}
\tag{111}
$$

To obtain the last inequality in the display above, we have used the bound

$$
(n_j-1)(n-n_j+1) \le n^2/4.
\tag{112}
$$

Indeed, for any non-trivial partition $1 \le n_j \le n-1$, we set $a := n_j - 1 \in [0, n-2]$ and note that $(n_j-1)(n-n_j+1) = a(n-a)$. Since $f(a) := a(n-a)$ is concave, its maximum over $[0, n-2]$ is attained at $a = n/2$ if $n/2 \le n-2$ or at the endpoint $a = n-2$. Owing to our assumption (recall the statement of the corollary) that $n \ge 4$, the maximum is thus attained at $n/2$ so that $(n_j-1)(n-n_j+1) \le n/2\,(n-n/2)$.

**Step 5: Uniform upper bound of $\rho$.** The class $\ell$, as defined in this proof, may be different for different partitions. In order to obtain a bound that applies to all relevant partitions, we replace $1 - R^\ell$ with $R^\star := \min(R^1, R^2)$, which yields

$$
\rho \le 1 - \left( \frac{4\tau^2 (R^\star)^2 n^{-1}}{3\tau^2 + \sigma^2} \right)^2 =: \rho_h.
$$

We recall that $0 < R^\star < 1$ for any non-trivial class assignment, and thus complete the proof. $\qquad\square$

### D.4. Proof of Corollary 3.12

We restate Corollary 3.12 and provide a proof.

**Corollary 3.12** (Main result: Hartigan's algorithm). *Consider the setting of Model 2.1. Assume that $n \geq 4$. Then, the probability that there is any nonempty incorrect partition (Definition 2.4) that is a fixed point of Hartigan's algorithm is bounded by*

$$\mathbb{P}\left(\exists \mathcal{P} \text{ a nonempty incorrect partition}\right) \leq 2^n \rho_h^{d/4}, \tag{28}$$

*where $\rho_h$ is given by (27).*

*Proof.* We denote the set of all partitions that are not correct partitions and not empty by $\mathcal{Q}$.

By the Union Bound in Equation (38), the probability that at least one of the partitions is not a fixed point is

$$\mathbb{P}\left(\exists \text{ non-fixed point in } \mathcal{Q}\right) = \mathbb{P}\left(\cup_{\mathcal{P} \in \mathcal{Q}}\left(\mathcal{P} \text{ is not a fixed point}\right)\right) \tag{113}$$

$$\leq \sum_{\mathcal{P} \in \mathcal{Q}} \mathbb{P}\left(\mathcal{P} \text{ is not a fixed point}\right) \tag{114}$$

The number of partitions that are not correct and not empty is smaller than the number of all possible partitions $2^n$ (ignoring symmetries), which completes the proof. □

# E. Additional Implementation Details

In this section, we provide additional details about the implementation and benchmarks. The code is written in Python using JAX (Bradbury et al., 2018) for Lloyd's $k$-means and Hartigan's $k$-means, Python's scikit-learn (Pedregosa et al., 2011) for spectral clustering, and CVXPY (Diamond & Boyd, 2016) for the SDP relaxation. We run all our experiments on a Quadro RTX 4000 GPU with 8GB of RAM.

## E.1. Clustering Algorithms

Here, we provide additional details on the main benchmark algorithms we compared with Lloyd's and Hartigan's $k$-means algorithms.

**"PCA + Lloyd".** A popular heuristic for $k$-means in high-dimensions, where PCA is applied to the data as a preprocessing step before $k$-means (Ding & He, 2004). In our experiments, we set the target dimension to $\min(\max(4, K), d)$, where $K$ is the number of clusters and $d$ is the data dimension. We note that when presenting a result involving the $k$-means loss, we use the partition produced by the algorithm to calculate the loss for the *original* data, not the reduced-dimension data.

**SDP relaxation of the $k$-means problem.** We implemented the SDP proposed in (Mixon et al., 2016), which is one of a family of state-of-the-art SDPs. Formal optimality results have been obtained for some of the algorithms in this family (Peng & Wei, 2007). We note that SDPs are notoriously difficult to scale.

**Spectral clustering.** A popular clustering algorithm that makes use of the spectral decomposition of a similarity matrix of the data (Shi & Malik, 2000; Ng et al., 2001). We note that some spectral clustering methods do not aim to solve the $k$-means minimization problem as formulated in Equation (5). Scaling spectral clustering is also challenging. In our experiments, we use the spectral clustering algorithm implemented in Python's scikit-learn library (Pedregosa et al., 2011) with default parameters (RBF affinity with $\gamma = 1.0$) and Lloyd's $k$-means for the final clustering step. We acknowledge that the default parameters are not optimal for every dataset and every metric. We observed that different parameter choices yield better results in some settings and worse in others; a comparison between the default parameters and a nearest-neighbors graph construction is provided in Section F.7 for reference. A thorough optimization of spectral clustering parameters is beyond the scope of this work.

## E.2. Olivetti Faces

We use the Olivetti faces dataset (Samaria & Harter, 1994), available in Python's scikit-learn library (Pedregosa et al., 2011). We standardized each feature to have zero mean and unit variance across the dataset, and then normalized each sample to unit Euclidean norm.

### E.3. Amazon Commerce Reviews

We use the Amazon Commerce Reviews dataset (Liu, 2011), obtained from OpenML (dataset ID 1457), which consists of $n = 1{,}500$ documents authored by $K = 50$ distinct users, with exactly 30 documents per user. Each document is represented by $d = 10{,}000$ dense stylometric and lexical features. We standardized each feature to have zero mean and unit variance across the dataset, and then normalized each sample to unit Euclidean norm.

### E.4. 20 newsgroups Datasets

Here, we describe the preprocessing steps applied to the 20 Newsgroups dataset (Mitchell, 1997; Lang, 1995) to create the datasets used for benchmarking clustering algorithms in Section 4. First, we removed the headers, footers, and quotes from the 20 newsgroups data. Subsequently, we extracted the documents (i.e., samples) corresponding to specific categories for each dataset:

- **20NG-A**: "alt.atheism" and "comp.graphics".

- **20NG-B**: "alt.atheism", "comp.graphics", "comp.windows.x", "misc.forsale", and "rec.autos".

- **20NG-C**: "alt.atheism", "comp.graphics", "comp.windows.x", "misc.forsale", "rec.autos", "rec.motorcycles", "rec.sport.baseball", "sci.crypt", "sci.med", and "sci.space".

- **20NG-D**: all 20 newsgroups.

Next, we tokenized each sample, omitting English stopwords and using standard TF-IDF with 5000 features, following the implementation in Python's scikit-learn (Pedregosa et al., 2011). We then discarded samples whose TF-IDF vector had zero norm. Finally, we randomly sampled $n$ samples with equal probability and without replacement for each dataset: 200 for 20NG-A, 500 for 20NG-B, 1000 for 20NG-C, and 2000 for 20NG-D. In Appendix F.6, we additionally consider versions of these datasets that include all available samples; we denote these by appending the suffix "(f)" (for "full") to the dataset name, e.g., 20NG-A (f).

## F. Additional Numerical Results

In this section, we present supplementary results to those described in Section 4. We begin by introducing the $k$-Means Win Rate, a metric for comparing partitions based on the $k$-means loss (Equation (5)). Next, we provide additional comparisons for the experiments performed in Section 4. Subsequently, we evaluate Lloyd's $k$-means against a simple algorithm based on Principal Component Analysis. We then benchmark the computing time of Lloyd's $k$-means and Hartigan's $k$-means. We then conduct numerical experiments for Theorems 3.4 and 3.9. Finally, we present results for larger versions of the datasets considered in Section 4, and provide additional comparisons of spectral clustering under different graph construction strategies.

### F.1. $k$-Means Win Rate: Comparing partitions through the $k$-Means loss

In the following sections, we compare clustering partitions using an alternative metric to the Normalized Mutual Information, based on the $k$-means loss (Equation (5)), to which we refer as the $k$-Means Win Rate. Since the raw loss is difficult to interpret across different parameter settings (e.g., data dimension or noise variance), we defined a score that assesses whether each algorithm produces a partition better or worse than the ground-truth partition with respect to the $k$-means loss. If the loss for the algorithm's output partition is close to the ground-truth loss (up to a relative difference of $10^{-6}$), the score is zero; if the output is better than the ground truth (lower loss), the score is one; if the ground truth is better, the score is negative one. By running multiple experiments for each parameter setting, we average these scores across all generated instances to obtain an overall performance measure. We note that the $k$-Means Win Rate is more sensitive to misclassification than the NMI, as a single misclassified point can leave the NMI largely unchanged while causing an individual experiment to receive the minimum score of $-1$, which can substantially impact the average $k$-Means Win Rate.

### F.2. Additional Results for Synthetic GMM Experiments

In this section, we revisit the experiments conducted in Section 4.1 and present the results in terms of the metric based on the $k$-Means Win Rate (see Section F.1). We note that when dimensionality reduction is used ("PCA + $k$-means"), we use the

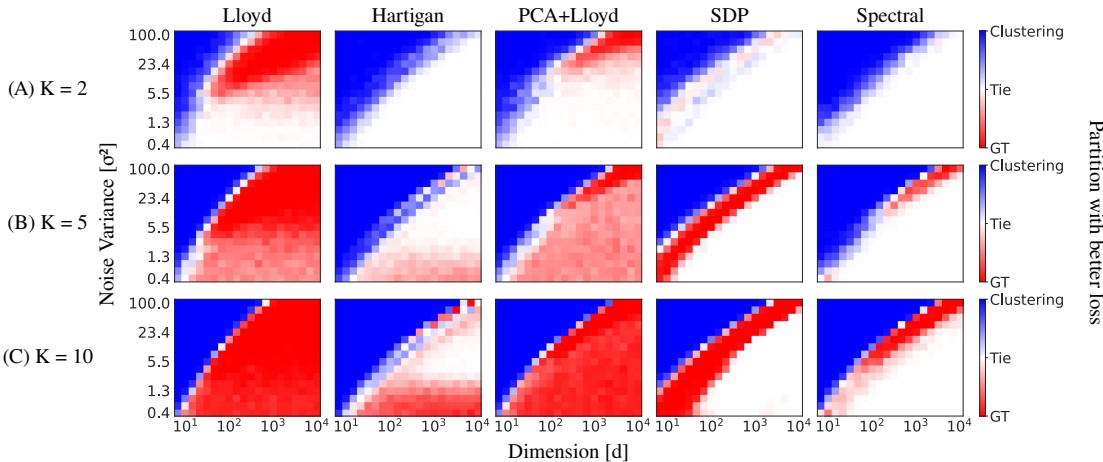

*Figure 2.* Comparison of the *k*-Means Win Rate (see Section F.1) obtained with each algorithm for the synthetic GMM dataset for different values of $K$. Each value corresponds to the average of 100 independent experiments, where, for each instance, we sample data from the Gaussian mixture model defined in Model 2.1 (generalized to $K \geq 2$) with $\tau^2 = 1$ and 20 samples per class, and run each algorithm until convergence using the same initialization strategy as in Figure 1.

partition produced by the algorithm to calculate the loss for the original data, not the reduced-dimension data. Additionally, we present complementary results to Figure 1, where we compare the performance of Lloyd's and Hartigan's *k*-means in terms of the NMI for different initialization strategies.

Figure 2 illustrates the average scores for each clustering algorithm, Lloyd's and Hartigan's *k*-means, and those described in Section E, against the ground truth in terms of the *k*-Means Win Rate. For each algorithm, results are shown for the same initialization strategy as in Figure 1. These results are consistent with the NMI scores shown in Figure 1 and demonstrate that, at high dimensions, Lloyd's *k*-means algorithm fails to minimize Equation (5), and converges to suboptimal fixed points.

Figure 3 and Figure 4 compare the performance of Lloyd's and Hartigan's *k*-means algorithm in terms of the NMI and *k*-Means Win Rate, respectively, for each initialization strategy. We observe that in this regime Hartigan's algorithm is sensitive to the initialization method in $K > 2$, but still achieves high NMI across the different initializations, suggesting that it finds good partitions, but not always the optimal ones. Lloyd's performance is significantly reduced with the "random partition" initialization; we attribute this to the fact that, when using random centers or *k-means++* for initialization, it is possible that the initial centers are associated with distinct ground-truth centers, with higher probability at lower values of $K$. In that case, assigning samples to the nearest cluster in a single iteration of Lloyd's algorithm can be effective. However, when $K$ is larger, it becomes less likely to obtain centers from distinct ground-truth clusters. When using a random balanced partition for initialization, the initial centers tend to be a balanced mix of samples, which results in a higher probability that the initial partition is a fixed point of Lloyd's *k*-means, as implied by Theorem 3.6. This is illustrated further in the next figure.

Figure 5 shows how many iterations, on average, Lloyd's *k*-means takes to reach convergence for each value of $d$ and $\sigma^2$. The figure shows that, at high dimensions, Lloyd's *k*-means converges in one iteration on average. This is consistent with our theory that argues that Lloyd terminates after the first iteration because the next iteration cannot update the partition. We observe that the phenomenon is even more pronounced when initialization is performed using random partitions.

Figure 6 offers a more direct comparison of the NMI obtained by Lloyd's *k*-means, Hartigan's *k*-means, SDP, and spectral clustering in the numerical experiments performed in Section 4. In low-noise regimes, spectral clustering typically achieves slightly higher NMI than other methods, with the gap rarely exceeding 0.1 when compared to Hartigan's *k*-means. For $K = 2$ and $K = 5$, spectral clustering also appears more robust at the highest noise levels; however, this advantage weakens as $K$ increases, and Hartigan's method becomes competitive and superior in the case of $K = 10$. We emphasize that spectral clustering is generally less scalable and, depending on the graph construction and normalization, need not optimize the same *k*-means objective as Hartigan's algorithm.

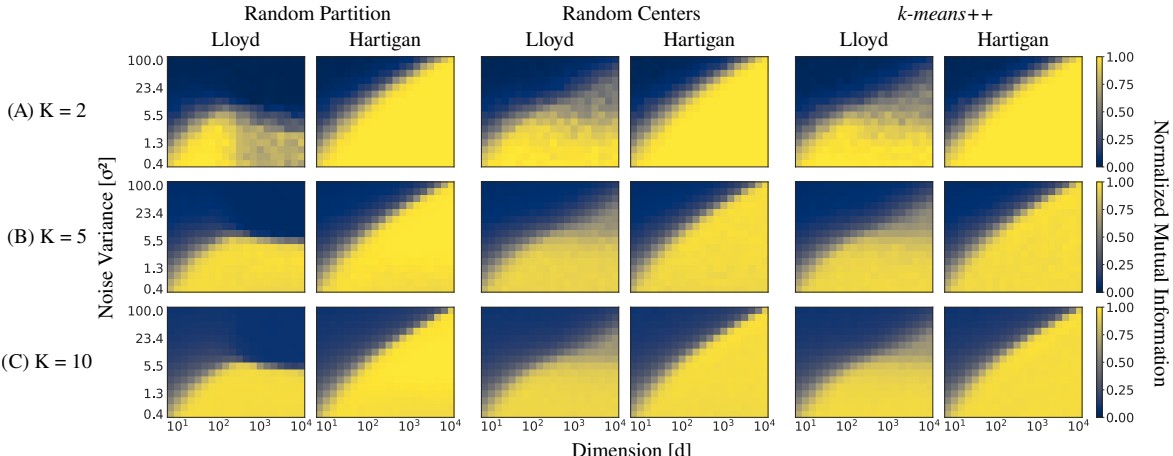

*Figure 3.* Normalized Mutual Information (NMI) between ground-truth clusters and the clusters obtained by Lloyd's and Hartigan's *k*-means across different initialization strategies and numbers of classes. An NMI value of 1 indicates perfect correlation, while a value of 0 signifies no mutual information between two assignments. Each value corresponds to the average of 100 independent experiments, where, for each instance, we sample from the GMM defined in Model 2.1 (generalized to $K \geq 2$) with $\tau^2 = 1$ and 20 samples per class, and run each algorithm until convergence. This figure shows that Lloyd's *k*-means is more sensitive to the initial centers as the data dimension increases, whereas Hartigan's *k*-means is less sensitive. Details for each initialization strategy are available in Section 4.

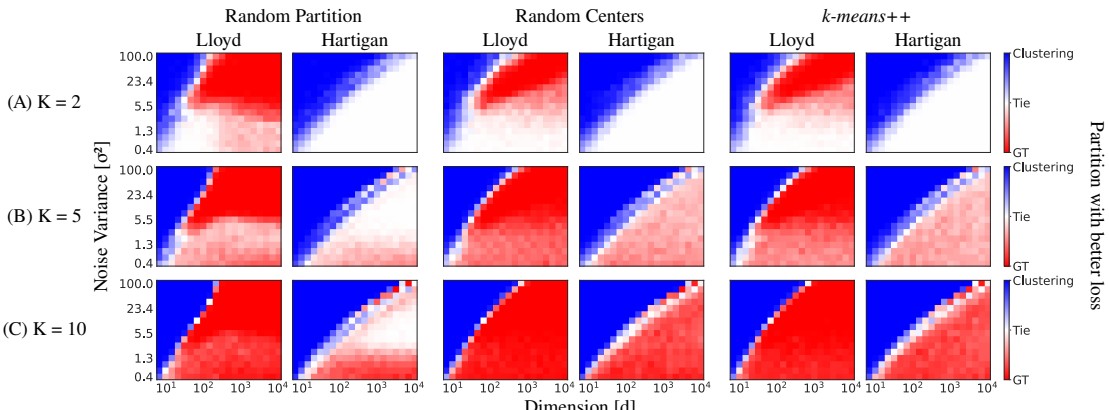

*Figure 4.* *k*-Means Win Rate (see Section F.1) comparison of Lloyd's and Hartigan's *k*-means across different initialization strategies and numbers of classes. Each value corresponds to the average of 100 independent experiments where, for each instance, we sample data from the Gaussian mixture model defined in Model 2.1 (generalized to $K \geq 2$) with $\tau^2 = 1$ and 20 samples per class, and run each algorithm until convergence.

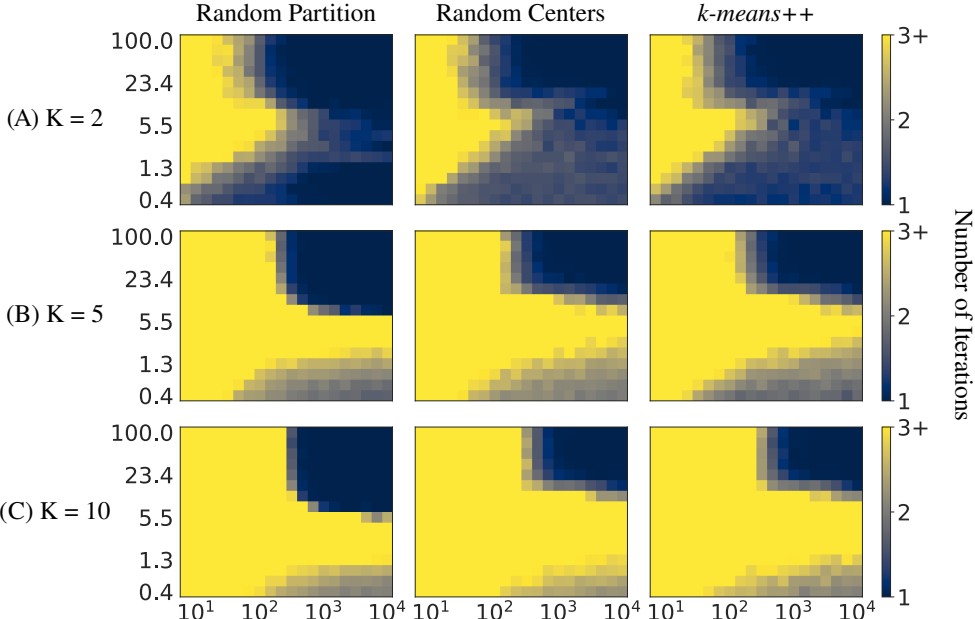

*Figure 5.* Number of iterations performed by Lloyd's $k$-means for different initialization strategies and number of classes, $K$. Each value corresponds to the average of 100 independent experiments, where, for each instance, we sample data from the Gaussian mixture model defined in Model 2.1 (generalized to $K \geq 2$) with $\tau^2 = 1$ and 20 samples per class, and run Lloyd's $k$-means until convergence. We observe that, in the case of random partition initialization, we exclude the scenario in which the initialization itself might constitute a fixed point. Consequently, the number of iterations will be 1 even if the partition remains unchanged after the first iteration.

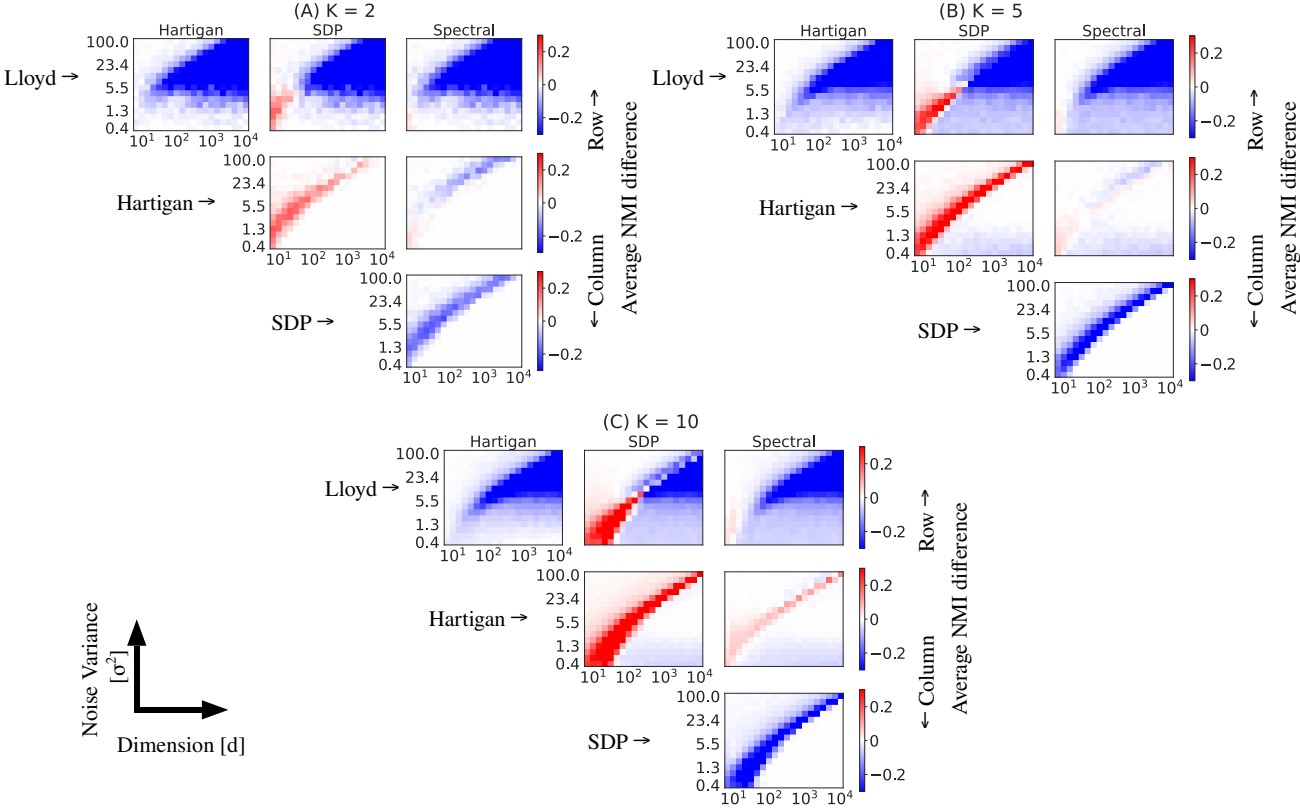

*Figure 6.* Comparison of Normalized Mutual Information (NMI) values between different clustering approaches. Each value represents the average of 100 independent experiments, where, for each experiment, data is sampled from the Gaussian Mixture Model defined in Section 2.1 with $\tau^2 = 1$ and 20 samples per class. For each pair of clustering methods, we calculate the difference in NMI and average it across all experiments. The colorbars include arrows indicating which color represents better performance: the row method outperforms the column method in red regions, and the column method outperforms the row method in blue regions.

### F.3. Lloyd's *k*-Means Algorithm Fails on "Easy Problems"

To illustrate that the clustering problem can become "easy" in certain regimes, we introduce a simplified clustering algorithm based on Principal Component Analysis (PCA), to which we refer as "PCA + Split". This algorithm is specifically designed for the simple case of two clusters, which is the focus of this paper. In the "PCA + Split" algorithm, we compute the first principal component of the data and partition the samples based on the sign of their principal component coefficient. Figure 7 compares the partition obtained by Lloyd's *k*-means and "PCA + Split" and the ground-truth partition. In Figure 7 (A), the partitions are compared through the *k*-Means Win Rate (see Section F.1); while in Figure 7 (B), the partitions are compared through the Normalized Mutual Information (see Definition A.13).

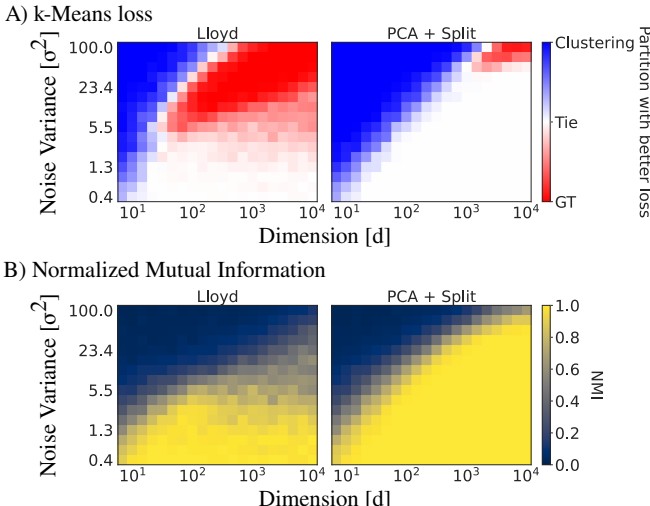

*Figure 7.* Comparison of the partitions obtained by Lloyd's *k*-means and "PCA + Split" and the ground-truth partition via (A) the *k*-Means Win Rate (see Section F.1) and (B) the Normalized Mutual Information (NMI, see Definition A.13). Each value corresponds to the average of 100 independent experiments, where for each instance we sample data from the GMM defined in Model 2.1 with $\tau^2 = 1.0$ and 20 samples per class. We sample the data such that the true clusters are balanced. Lloyd's *k*-means is initialized with *k-means++* and is run until convergence.

## F.4. Computational Performance Comparison of Lloyd's and Hartigan's $k$-Means

Although the timing analysis is beyond the primary scope of this paper, we present a limited set of comparisons to illustrate that Hartigan's algorithm performs similarly to Lloyd's algorithm in terms of execution time. We chose not to include spectral and SDP algorithms in this analysis, as they are often impractical for the scale of problems considered here. For this comparison, both algorithms were implemented in Numba (Lam et al., 2015). For Lloyd's algorithm, distance computations and centroid updates were vectorized using NumPy broadcasting. For Hartigan's algorithm, distance computations were similarly vectorized; in addition, rather than recomputing the affected centroids from scratch after each reassignment, we update them incrementally from their previous values and the reassigned sample, following the local center update introduced by Bottou & Bengio (1994). All experiments were conducted in a single CPU thread, with no GPU acceleration.

We evaluated the algorithms across different dimensions (d), sample sizes (n), and numbers of clusters (K). Data was sampled from the Gaussian Mixture Model (GMM) specified in Model 2.1 (generalized to $K \geq 2$) with $\tau^2 = 1$ and $\sigma^2 = 10$. In this regime, Lloyd's algorithm typically runs for a few iterations before terminating. For each experiment, a new dataset was generated, and each clustering algorithm was initialized with random samples as centers and executed once. The experiments were repeated 10 times for each combination of $d$, $n$, and $K$ values. Figure 8 (A) and Figure 8 (B) show the computation times for running each algorithm for a single iteration and until convergence, respectively. We observe that Lloyd's and Hartigan's algorithms have comparable per-iteration computational costs in this implementation. The differences observed when running each algorithm until convergence are driven by the number of iterations required: at high dimensions, Lloyd's algorithm becomes trapped in fixed points after a few iterations and terminates quickly, whereas Hartigan's algorithm continues to update. It is important to note that these results may vary across implementations, especially when using parallel computing resources such as GPUs. Figure 8 (C) reports the maximum number of iterations required for convergence across the 10 trials. The two algorithms exhibit divergent behavior: the number of iterations required by Lloyd's algorithm decreases with dimension, whereas the opposite holds for Hartigan's algorithm. This is consistent with our theoretical results, as higher dimensions are expected to cause Lloyd's algorithm to become trapped in fixed points more quickly.

(A) Compute time single iteration

(B) Compute time until convergence

(C) Maximum iterations until convergence

*Figure 8.* **Computation Time for Lloyd's and Hartigan's $k$-Means Algorithms**. Each value represents the mean over 10 independent trials, where data is sampled from the Gaussian Mixture Model (GMM) described in Model 2.1 (generalized for $K \geq 2$) with $\tau^2 = 1$ and $\sigma^2 = 10$. Panel (A) shows the average computation time for a single iteration of each algorithm, while Panel (B) depicts the average computation time required for each algorithm to reach convergence. Panel (C) reports the maximum number of iterations until convergence across the 10 trials for each configuration.

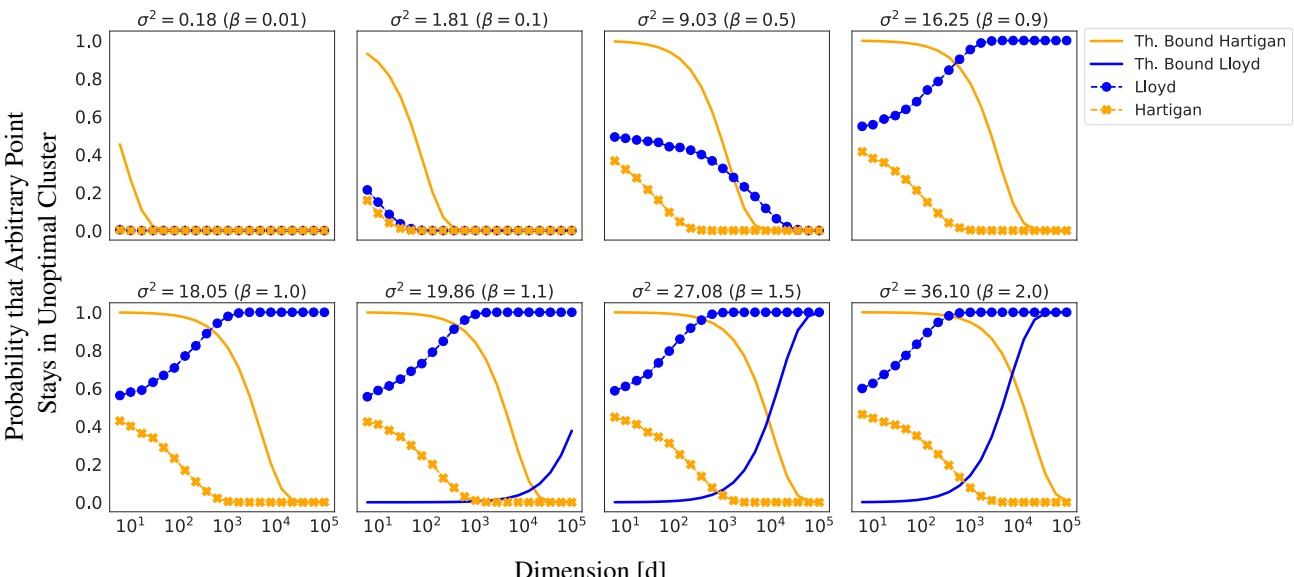

*Figure 9.* Numerical experiments for Theorems 3.4 and 3.9. For each experiment, data (centers and samples) are sampled from Model 2.1 for the special case where the ground-truth partition is balanced, $n = 40$ and $\tau^2 = 1$. Additionally, we fix the "current clusters" such that, given an arbitrary sample $i$, the purity coefficients are $R_j^\ell = 0.25$ and $R_{\bar{j}}^\ell = 0.75$; where the sample has true assignment $i \in S_\ell^\star$ and current assignment $i \in C_j$. In the plots, we show the ratio of instances in which sample $i$ remains in cluster $C_j$ after a step of Lloyd's $k$-means (blue, circles) or Hartigan's $k$-means (orange, crosses). The theoretical bounds for this ratio corresponding to each algorithm are plotted as solid lines, with the bound from Theorem 3.4 shown only for $\sigma^2 > 18.05$, where its conditions are satisfied. We observe that, in the relevant regime, as the dimension increases, Lloyd's algorithm rarely moves the sample to the other cluster, in sharp contrast with Hartigan's algorithm.

## F.5. Divergent Behaviors of Fixed Points of the Algorithm

In this section, we present results for numerical experiments demonstrating Theorems 3.4 and 3.9. Each instance is an independent experiment where the data (centroids and samples) are sampled from the GMM defined in Model 2.1, with $n = 40$ and $\tau^2 = 1$. To substitute concrete values into the expression in Theorems 3.4 and 3.9 and conform to their assumptions, we consider the case when both the ground-truth clusters and the current clusters are balanced ($R^\ell = R^{\bar{\ell}} = 0.5$ and $|C_1| = |C_2|$), but the current clusters are defined such that $R_j^\ell = 0.25$ and $R_{\bar{j}}^\ell = 0.75$. That is, we define the current clusters such that switching the sample from $C_j$ to $C_{\bar{j}}$ would improve the $k$-means loss (Equation (5)). In each instance, we examine whether Lloyd's $k$-means or Hartigan's $k$-means would move sample $i$ to the other cluster.

We repeat the experiment $10^4$ times for each of several values of $\sigma^2$ and $d$, and plot in Figure 9 the ratio of instances where the sample $i$ remains in its current cluster under the criteria of each clustering algorithm. The error interval, which is difficult to see, is Wilson's interval (See Definition A.12). The values of $\sigma^2$ are defined such that $\sigma^2 = \beta\,\sigma_0^2$, where $\sigma_0^2$ is the value required for the assumptions of Theorem 3.4 to hold (Equation (13)).

## F.6. Experiments on Larger Datasets

In this section, we present results for larger versions of some of the datasets analyzed in the previous experiments: a larger version of the GMM data used to compare the clustering algorithms in Figure 1, and a larger version of the 20 Newsgroups datasets, where all available samples were used instead of random subsets.

Figures 10, 11, 12, 13, and 14 present results for the larger GMM dataset corresponding to Figures 1, 2, 3, 4, and 6, respectively, with 100 samples per class (compared to 20 in the previous experiments). Due to the increased computational cost, we restrict these experiments to Lloyd's *k*-means, "PCA+Lloyd", Hartigan's *k*-means, and spectral clustering. The overall behavior of the algorithms remains consistent with the smaller-scale experiments, with the main difference being that the larger sample size makes the algorithms more robust to noise: a higher noise level is required before they fail to recover the true clusters.

Table 2 reports the results for the large-scale GMM experiments; we fix $\sigma^2 = 10$ and $d = 200$, and run experiments where the number of samples ranges from 20,000 to 100,000, with $K$ ranging from 200 to 1,000. Across all configurations, Hartigan's *k*-means achieves both a lower *k*-means loss and a higher NMI than Lloyd's *k*-means.

Finally, we evaluate the algorithms on the full 20 Newsgroups datasets, using all available samples in each group that is included (instead of random subsets). Table 3 reports the *k*-means loss and NMI obtained by Lloyd's *k*-means, Hartigan's *k*-means, and spectral clustering. Hartigan's algorithm consistently achieves a lower *k*-means loss than the other methods.

*Table 2.* Normalized *k*-means loss and NMI for clustering on large-scale GMM datasets, with Lloyd's *k*-means initialized using *k*-means++ and Hartigan's *k*-means initialized with a random partition. For each configuration, we sample data from the GMM defined in Model 2.1 (generalized to $K \geq 2$) with $\tau^2 = 1$ and the specified value of $\sigma^2$, and run each algorithm until convergence. The *k*-means loss (Equation (5)) is divided by $d \cdot n$ for readability, given the scale of the raw values. The "GT" column reports the normalized *k*-means loss of the ground-truth partition. Hartigan's *k*-means consistently achieves a lower *k*-means loss and a higher NMI than Lloyd's *k*-means across all configurations.

| | DATASET PARAMETERS | | | | *k*-MEANS LOSS / $(d \cdot n)$ | | | NMI | |
| --- | --- | --- | --- | --- | --- | --- | --- | --- | --- |
| | $n$ | $d$ | $K$ | $\sigma^2$ | LLOYD | HARTIGAN | GT | LLOYD | HARTIGAN |
| GMM | 20000 | 200 | 200 | 10 | 10.137 | **9.875** | 9.895 | 0.52 | **0.90** |
| GMM | 50000 | 200 | 500 | 10 | 10.203 | **9.961** | 9.902 | 0.34 | **0.61** |
| GMM | 100000 | 200 | 500 | 10 | 10.124 | **9.921** | 9.950 | 0.54 | **0.87** |
| GMM | 100000 | 200 | 1000 | 10 | 10.178 | **10.029** | 9.900 | 0.34 | **0.40** |

*Table 3.* Clustering results for the full 20 Newsgroups datasets using Lloyd's and Hartigan's *k*-means and spectral clustering. 50 random initializations per dataset for each algorithm, reduced from 500 in Table 1 due to the increased computational cost. As in Table 1, we report the *k*-means loss (Equation (5)) and the NMI (see Definition A.13) corresponding to the partition that achieves the lowest *k*-means loss. SDP is omitted as it does not scale to this regime. Hartigan's *k*-means consistently achieves the lowest *k*-means loss across all datasets.

| | DATASET PARAMETERS | | | *k*-MEANS LOSS | | | NMI | | |
| --- | --- | --- | --- | --- | --- | --- | --- | --- | --- |
| | $n$ | $d$ | $K$ | LLOYD | HARTIGAN | SPECTRAL | LLOYD | HARTIGAN | SPECTRAL |
| 20NG-A (F) | 1727 | 5000 | 2 | 1684.84 | **1684.80** | 1686.26 | 0.61 | **0.63** | 0.02 |
| 20NG-B (F) | 4590 | 5000 | 5 | 4458.09 | **4457.68** | 4460.86 | **0.49** | 0.46 | 0.46 |
| 20NG-C (F) | 9367 | 5000 | 10 | 9036.94 | **9036.58** | 9049.33 | **0.41** | 0.40 | 0.32 |
| 20NG-D (F) | 18246 | 5000 | 20 | 17432.27 | **17423.16** | 17446.23 | **0.33** | 0.32 | 0.26 |

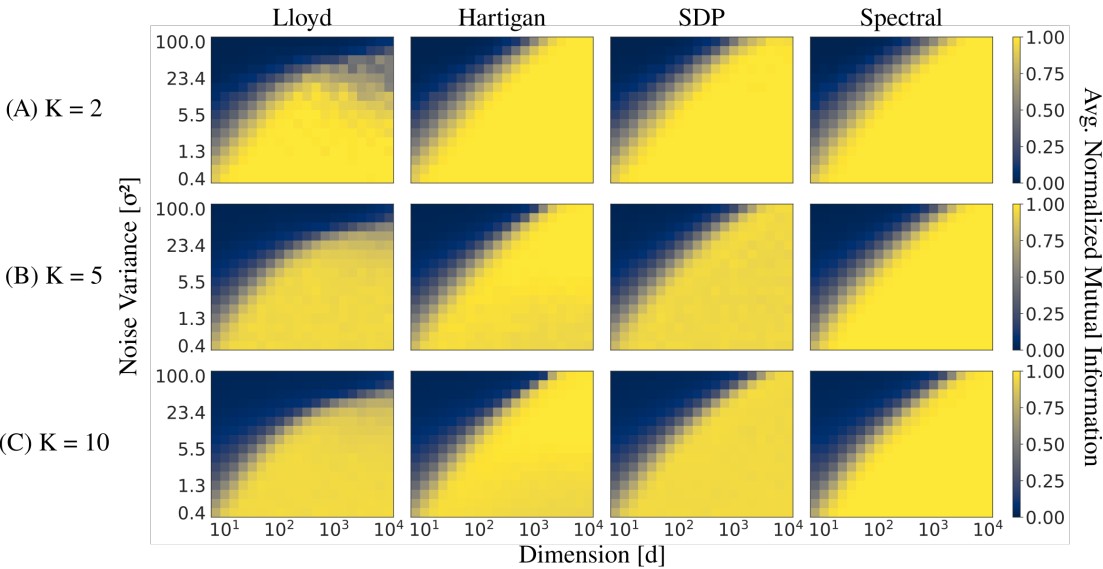

*Figure 10.* Normalized mutual information (NMI; see Definition A.13) between the ground-truth partition and the output of each clustering algorithm on the larger GMM dataset. Each entry reports the mean over 100 independent trials: in each trial, we sample data from the Gaussian mixture model (GMM) in Model 2.1 (generalized to $K \geq 2$) with $\tau^2 = 1.0$ and 100 samples per class, and run each algorithm until convergence, with Lloyd's algorithm initialized using $k$-means++ and Hartigan's algorithm initialized with a random partition. SDP is omitted as it does not scale to this regime. The results are consistent with those in Figure 1 obtained at 20 samples per class, with the main difference being that the larger sample size makes the algorithms more robust to noise.

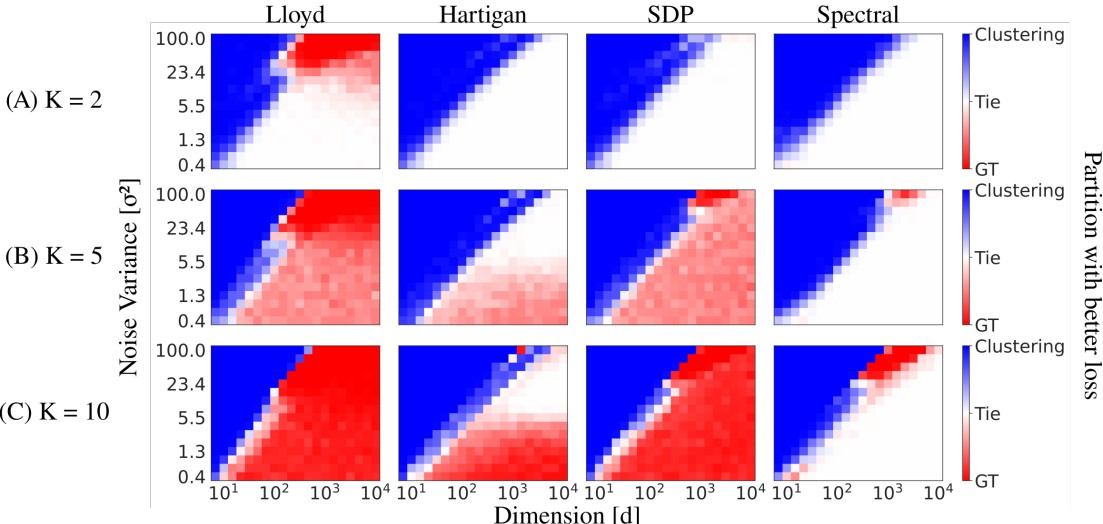

*Figure 11.* Comparison of the $k$-Means Win Rate (see Section F.1) obtained with each algorithm on the larger GMM dataset for different values of $K$. Each value corresponds to the average of 100 independent experiments, where, for each instance, we sample data from the Gaussian mixture model defined in Model 2.1 (generalized to $K \geq 2$) with $\tau^2 = 1$ and 100 samples per class, and run each algorithm until convergence using the same initialization strategy as in Figure 10.

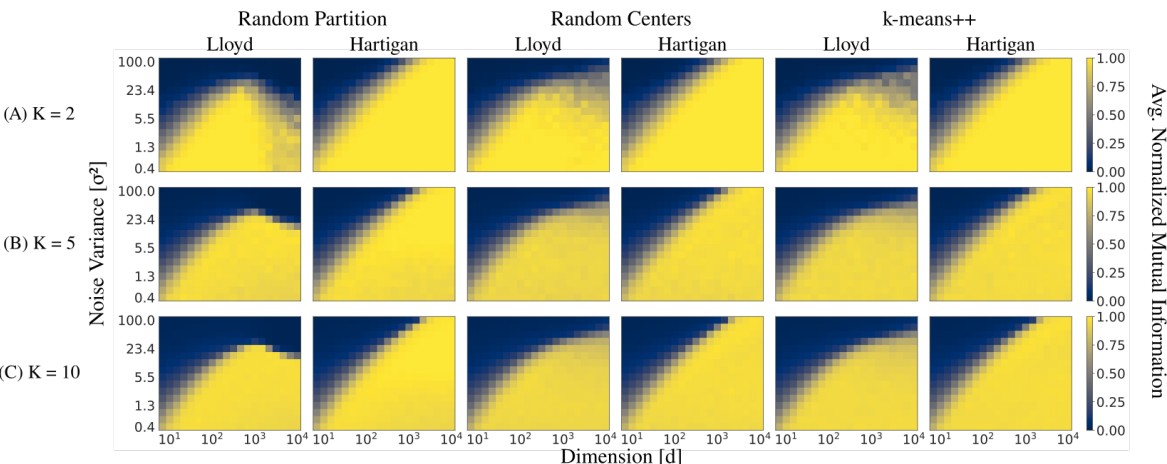

*Figure 12.* Normalized Mutual Information (NMI) between ground-truth clusters and the clusters obtained from Lloyd's and Hartigan's *k*-means on the larger GMM dataset. An NMI value of 1 indicates perfect correlation, while a value of 0 signifies no mutual information between two assignments. Each value corresponds to the average of 100 independent experiments, where, for each instance, we sample data from the GMM defined in Model 2.1 (generalized to $K \geq 2$) with $\tau^2 = 1$ and 100 samples per class, and run each algorithm until convergence. The results are consistent with those in Figure 3, showing that Lloyd's *k*-means remains sensitive to the initial centers as the data dimension increases, even at the larger sample size. Details for each initialization strategy are available in Section 4.

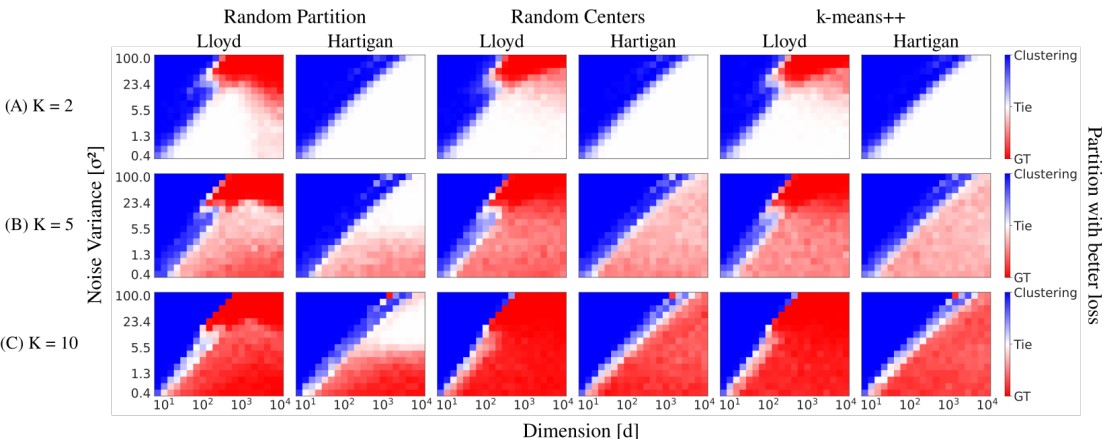

*Figure 13.* *k*-Means Win Rate (see Section F.1) comparison of Lloyd's and Hartigan's *k*-means on the larger GMM dataset. Each value corresponds to the average of 100 independent experiments where, for each instance, we sample data from the Gaussian mixture model defined in Model 2.1 (generalized to $K \geq 2$) with $\tau^2 = 1$ and 100 samples per class, and run each algorithm until convergence.

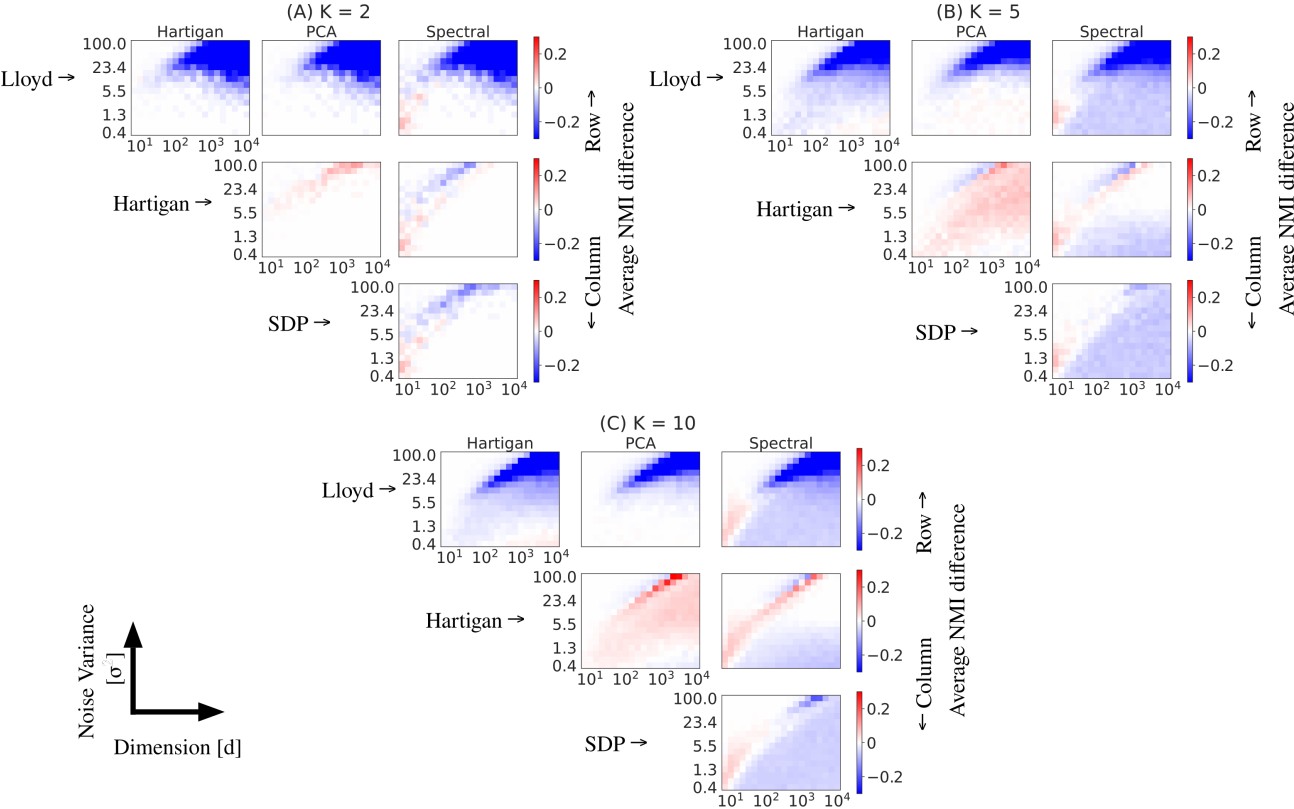

*Figure 14.* Comparison of Normalized Mutual Information (NMI) values between different clustering approaches on the larger GMM dataset. Each value represents the average of 100 independent experiments, where, for each experiment, data is sampled from the Gaussian Mixture Model defined in Section 2.1 with $\tau^2 = 1$ and 100 samples per class. For each pair of clustering methods, we calculate the difference in NMI and average it across all experiments. The colorbars include arrows indicating which color represents better performance: the row method outperforms the column method in red regions, and the column method outperforms the row method in blue regions.

## F.7. Additional Spectral Clustering Results

In this section, we present additional results for spectral clustering under two strategies for constructing the affinity graph: (i) a fully connected graph with a radial basis function (RBF) kernel with $\gamma = 1.0$, corresponding to the default scikit-learn configuration used in the main text, and (ii) a nearest-neighbors graph with the number of neighbors set to $\lceil \log n \rceil$, following the heuristic of von Luxburg (2007). Although a wide range of alternative graph constructions exists, a thorough optimization of spectral clustering is beyond the scope of this paper; we include these comparisons for completeness, as they illustrate that alternative graph constructions can improve performance in some cases. The results are summarized in Figures 15 and 16, and Table 4.

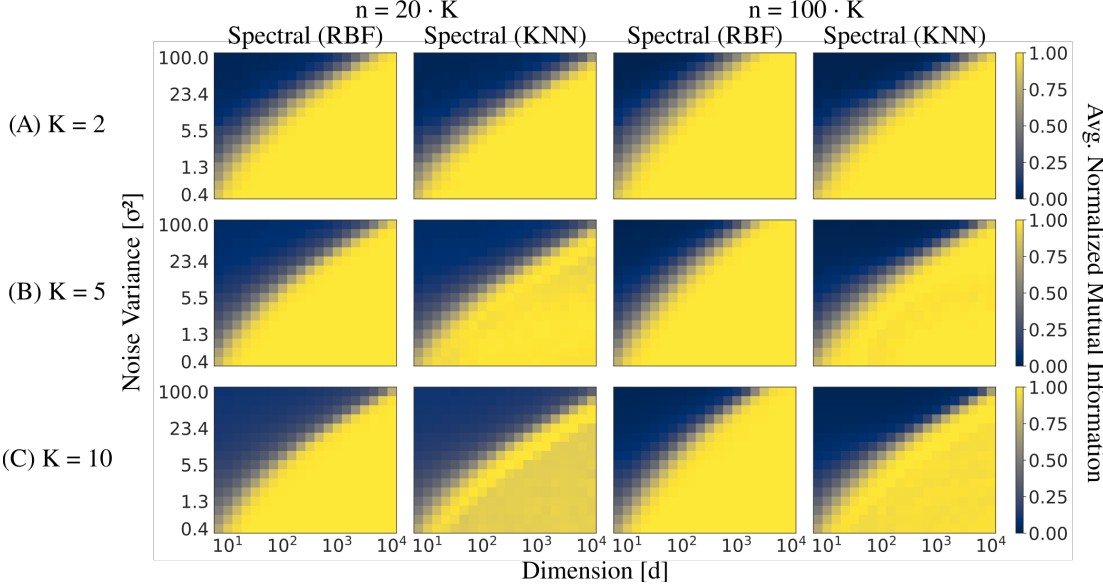

*Figure 15.* Normalized mutual information (NMI; see Definition A.13) between the ground-truth partition and the output of each spectral clustering strategy on the synthetic GMM dataset. Each entry reports the mean over 100 independent trials, in which we sample data from the Gaussian mixture model (GMM) in Model 2.1 (generalized to $K \geq 2$) with $\tau^2 = 1.0$, and run each algorithm until convergence. The first two columns correspond to $n = 20 \cdot K$ samples (20 per class) and the last two to $n = 100 \cdot K$ samples (100 per class). The results show that constructing the affinity graph with the nearest-neighbors strategy degrades the performance of spectral clustering in the high-noise regime.

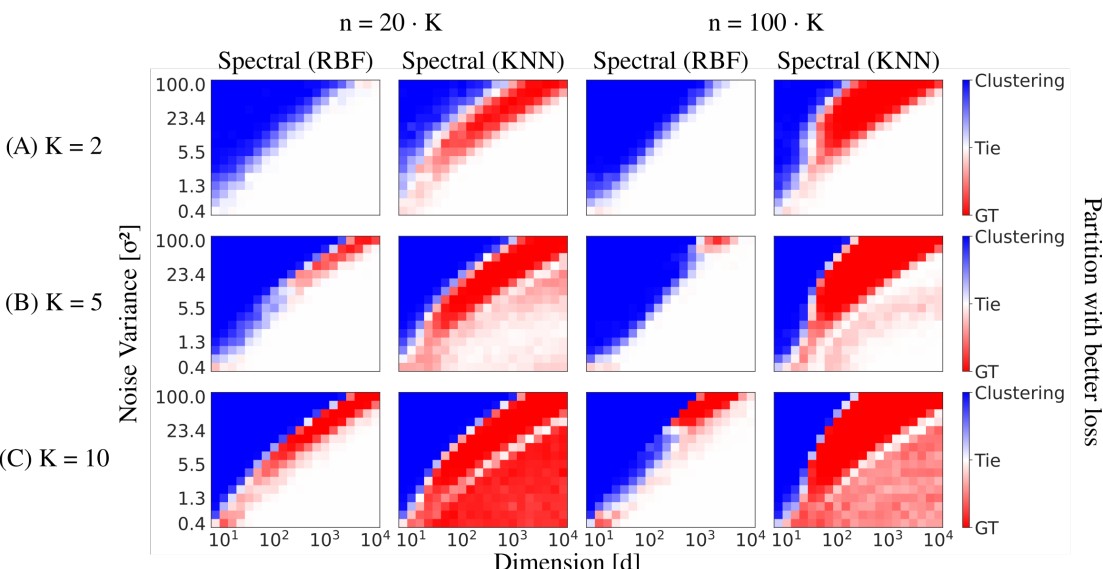

*Figure 16.* Comparison of the *k*-Means Win Rate (see Section F.1) obtained with each spectral clustering strategy on the synthetic GMM dataset for different values of *K*. Each value corresponds to the average of 100 independent experiments, where, for each instance, we sample data from the Gaussian mixture model defined in Model 2.1 (generalized to $K \geq 2$) with $\tau^2 = 1$, and run each algorithm until convergence. The first two columns correspond to $n = 20 \cdot K$ samples (20 per class) and the last two to $n = 100 \cdot K$ samples (100 per class).

*Table 4.* Spectral clustering results for all real-world datasets under two strategies for constructing the affinity graph: a fully connected graph with an RBF kernel ($\gamma = 1.0$), and a nearest-neighbors graph with the number of neighbors set to $\lceil \log n \rceil$. For each strategy, we report the *k*-means loss (Equation (5)) and the NMI (see Definition A.13) corresponding to the partition that achieves the lowest *k*-means loss across 500 random initializations (50 for the full 20NG datasets, denoted by the suffix "(f)"). Neither strategy is uniformly better across datasets and metrics, motivating our use of the default RBF configuration as the more consistent choice for the experiments in the main text.

| DATASET PARAMETERS | | | | *k*-MEANS LOSS | | NMI | |
|---|---|---|---|---|---|---|---|
| | $n$ | $d$ | $K$ | SPECTRAL (RBF) | SPECTRAL (KNN) | SPECTRAL (RBF) | SPECTRAL (KNN) |
| OLIVETTI | 400 | 4096 | 40 | 166.20 | **165.37** | 0.83 | **0.84** |
| AMAZON | 1500 | 10000 | 50 | **1363.28** | 1378.10 | 0.38 | **0.51** |
| 20NG-A | 200 | 5000 | 2 | **193.48** | 193.57 | **0.52** | 0.50 |
| 20NG-B | 500 | 5000 | 5 | 482.89 | **482.28** | 0.37 | **0.39** |
| 20NG-C | 1000 | 5000 | 10 | **953.67** | 956.88 | 0.27 | 0.27 |
| 20NG-D | 2000 | 5000 | 20 | **1889.99** | 1902.52 | **0.26** | 0.21 |
| 20NG-A (F) | 1727 | 5000 | 2 | 1686.26 | **1685.50** | 0.02 | **0.72** |
| 20NG-B (F) | 4590 | 5000 | 5 | **4460.86** | 4487.56 | **0.46** | 0.26 |
| 20NG-C (F) | 9367 | 5000 | 10 | **9049.33** | 9101.78 | 0.32 | **0.36** |
| 20NG-D (F) | 18246 | 5000 | 20 | **17446.23** | 17633.04 | 0.26 | **0.36** |

