# OpenReview forum: "The Catastrophic Failure of *the* k-Means Algorithm in High Dimensions, and How Hartigan's Algorithm Avoids It"
_ICML.cc/2026/Conference — ICML 2026 regular_

### Official Review · Reviewer_jcJT · 2026-03-09

**Soundness:** 3
**Presentation:** 3
**Significance:** 3
**Originality:** 3
**Overall Recommendation:** 3
**Confidence:** 5

**Summary:**

This paper demonstrates that Lloyd's k-means algorithm exhibits catastrophic failure in high-dimensional, high-noise environments: and almost every data partition is highly likely to be a fixed point. Hartigan's sk-means algorithm, however, does not suffer from this flaw.

**Compliance With Llm Reviewing Policy:**

Affirmed.

**Final Justification:**

I am very grateful to the author for responding to my questions in the rebuttal, and some of them were well resolved. However, some concerns about this paper remain. In terms of experiments, most of the datasets in this paper are artificially generated, and I believe their settings are unreasonable. Furthermore, they were not tested on large-scale datasets. The importance of $k$-means lies in its ability to quickly process large-scale data; for small-scale data, many methods offer better clustering results than $k$-means. In terms of theory, this paper only analyzes the case of $k=2$, which is very limiting. Moreover, $k=2$ is biased towards Hartigan's $k$-means. Although the authors claim that more settings are needed for the case of $k>2$, and therefore this paper does not discuss it (I understand this is true in many cases, and a paper is not required to solve all problems), only by including the case of $k>2$ can this paper be considered complete and sufficiently contributive (because the problems mentioned earlier exist with $k=2$), rather than being split into multiple papers. After reading other reviewers' comments, I found that reviewer 74pU and I recognized the same question (which I raised in Q4): the paper requires $d$ to be extremely large, which is almost unreasonable in real-world scenarios, and most of the experiments in this paper construct such extreme examples on artificial datasets. After reading the authors' responses to reviewer 74pU and to me, I feel that the authors have not addressed the aforementioned problem. In summary, this paper does not propose a new method, but rather makes a theoretical contribution. However, it only discusses the case of $k=2$, which inherently biases towards Hartigan's $k$-means (meaning that the comparison is inherently unfair when $k=2$). Moreover, in real-world scenarios, $k$ is often greater than 2. Furthermore, this paper demonstrates the concept by constructing an extreme example (where $d$ is extremely large). Therefore, I tend to maintain my score.

**Key Questions For Authors:**

1. This paper does not propose a new algorithm, but rather provides proofs for existing methods to explain why Hartigan's k-means outperforms Lloyd's k-means. However, it only considers the case where k=2, while in reality, k is often greater than 2, thus failing to explain the reasons in practical situations. Furthermore, k=2 itself biases towards Hartigan's k-means.

2. Why are the spectral clustering results in Table 1 inferior to k-means? This seems to contradict the common understanding.

3. Hartigan's k-means and Lloyd's k-means use different initialization methods. Does this affect the experimental results? What are the experimental results under the two different initialization methods?

4. The synthetic dataset has only 20 points per class. If k is equal to 20, the total number of points is only 40, but the dimensionality is very high. Is this too extreme and deviates from reality?

5. Is the setting of a noise variance from 0 to 100, with only 20 points per class, reasonable?

6. The datasets used in Table 1 are all small, with a maximum of only 1000 points, while k-means is adopted because of its high efficiency in scaling to large-scale datasets, especially those with tens of millions of points.

**Limitations:**

No, the case where k > 2 was not analyzed.

**Strengths And Weaknesses:**

Strengths

This paper addresses a crucial issue: the shortcomings of Lloyd's k-means algorithm. A proof is provided for k=2. The paper is logically clear, well-organized, and easy to read.

Weaknesses

This paper does not propose a new solution and only analyzes the case when k=2. The synthesized dataset is also not very suitable.

---

> ### Author Rebuttal · Authors · 2026-03-30
>
> Q1. We are pleased that the reviewer recognizes the importance of the issue. The results for Lloyd's algorithm are easily generalizable to $k>2$. This requires some additional notation and accounting and obscures the basic argument. It was therefore "left on the editing floor" for the longer form version of this paper, in favor of presenting results for Hartigan's algorithm. The results for Hartigan's algorithm for $k>2$ are more nuanced and require further investigation; the general idea holds, but it is not as clean or general. The reviewer might consider this a bias towards Hartigan's algorithm, but the fundamental failure of Lloyd's algorithm persists, and the overwhelming advantage of Hartigan's algorithm does not appear to go away even if it is not as clean as $k=2$. The self-influence mechanism we discuss in the paper certainly holds for $k>2$.
> Our experiments with $k=5$ and $k=10$ convincingly indicate that this is not a special case of $k=2$.
>
>
>
> Q2. Spectral clustering does not optimize for the same k-means criterion. It is indeed often better in some metrics, even when it is worse than the k-means loss, simply because the k-means loss is not the best measure for every purpose; in the OLIVETTI dataset example, we observe that spectral clustering achieves a better NMI even though the k-means loss is worse.
> It is also sometimes better, even in the k-means criteria, than Lloyd's algorithm because of the issues in Lloyd's algorithm.
> Evidently, there are cases (20NG datasets) where Lloyd's algorithm is simply not very good, and Hartigan's algorithm outperforms spectral clustering when Lloyd's does not; we do not claim this is a universal phenomenon.
> An in-depth analysis of the differences between spectral clustering and the k-means criterion is beyond the scope of this paper; we present the spectral clustering results for completeness and reference. We note, however, that spectral clustering is generally less scalable than Lloyd's algorithm or k-means (without special tricks).
>
>
> Q3. The most straightforward initialization for Lloyd's algorithm in this context is a random partition. However, since k-means++ is a better, widely used initialization method, we used it with Lloyd's algorithm. A comparison of initialization is provided in the appendix, showing that k-means++ is advantageous for Lloyd's algorithm.  In other words, the comparison is favorable to Lloyd: it receives the stronger initialization (k-means++) and still fails in the high-dimensional regime. Hartigan's algorithm outperforms Lloyd's algorithm in almost all cases. We have noticed a degradation in certain cases of Hartigan with k-means++ initialization (Figures 3,4), so we did not impose this initialization on it in Figure 1. This observation is in line with our theoretical results because it is in lower noise and lower dimension cases than where our conservative theory applies. This is an interesting lead in further investigation of algorithms' dynamics in broader regimes than our initial results cover.
>
>
>
> Q4/Q5. We had 20 samples per class, so in the largest k in the synthetic experiments, there were 200 samples. We will add a larger experiment with $\geq100$ samples per class to the final version (the results are very similar). The noise range spans the transitions between the different regimes. Indeed, when noise is high and the number of samples is small, the recovered mean is very noisy, but, as the results demonstrate, Hartigan's algorithm can still recover the correct classification.
>
> Q6. Dataset sizes were chosen to allow comparison across all parameters with SDP and spectral methods, which don't scale well beyond a few thousand samples. We will add results with larger datasets (excluding some methods) in the appendix. We note that Hartigan's iterations have the same complexity as Lloyd's iterations, so they have similar basic scaling properties (further investigation of the dynamics and computational issues like parallelization is still needed - but preliminary results are surprisingly encouraging).

---

> > ### Author Rebuttal · Reviewer_jcJT · 2026-04-01
> >
> > Thanks for your rebuttal.
> >
> > My concerns about question 1 remain.
> >
> > Does your response to question 2 suggest that using these evaluation metrics is inappropriate? The NMI of spectral clustering on all three datasets is lower than that of kmeans. How did you construct the spectral clustering graph and how did you set the parameters?
> >
> > I still think Q4/5 settings are unreasonable for real-world scenarios. Can you show results on larger-scale data? The advantage of $k$-means is that it can run quickly on very large datasets.

---

> > > ### Author Response · Authors · 2026-04-07
> > >
> > > **Q1 (scope, and algorithmic novelty).**
> > >
> > > The $K=2$ case is the minimal setting. It is clean, concise and easy to generalize (for Lloyd's algorithm), but the generalization requires additional notation and would exceed the page limit. The experiments with $K=5$ and $K=10$ show that the phenomenon is not limited to $K=2$ (also see larger $K$ below). The mechanisms of Lloyd's failure (self-influence) and the proof (concentration) do not depend on $K$.
> > >
> > > The ICML 2026 guidelines state (in bold): *"Originality does not necessarily require introducing an entirely new method. Rather, a work that provides novel insights by evaluating existing methods, or demonstrates improved understanding is also equally valuable."* We respectfully submit that a paper about a fundamental failure of the best-known clustering algorithm fits this definition.
> > >
> > >
> > >
> > >
> > >
> > > **Q2 (spectral clustering and metrics).**
> > >
> > > The spectral clustering experiments are presented for completeness. It does not minimize the k-means objective and is therefore not the natural benchmark for the algorithms we analyze.
> > >
> > > On metrics: No. The metrics are standard, with natural limitations, and they reflect different aspects of the problem. The NMI and spectral clustering give a broader perspective on performance.
> > >
> > > On parameters: we use scikit-learn's `SpectralClustering` with default parameters (RBF affinity, $\gamma=1$, etc.), with a larger number of initializations (500). The 20NG text was processed using TF-IDF with English stop words and 5000 features. The default scikit-learn parameters arguably represent a classic, "unbiased" choice of parameters. When we constructed a $k$-nearest-neighbors graph for spectral clustering instead, it outperformed Lloyd and Hartigan on the 20NG datasets, but its performance on Olivetti was degraded.
> > >
> > > Spectral clustering outperforms on the Olivetti dataset and happens to underperform on the 20NG datasets in default parameters. The surprising thing about the experiments is that Hartigan's algorithm is empirically comparable to state-of-the-art $k$-means-loss algorithms such as SDP and Ndaoud's algorithm — exceeding the theoretical guarantees.
> > >
> > > In order to address the concerns, we will draw a distinction between the $k$-means-loss algorithms and spectral clustering, clarify that spectral clustering is presented for completeness, and reiterate the limitation of default parameters. We will clarify that the results invite an interesting question about instances where spectral clustering is known to outperform Lloyd's algorithm but Hartigan's fixes Lloyd's limitation - a question for future work that does not imply general superiority of Hartigan's algorithm over spectral clustering on general clustering tasks, if only for the reason that they do not minimize the same loss.
> > >
> > > The main story is the issue with Lloyd's algorithm and the comparison within the $k$-means-loss family of algorithms. **If the reviewers find that including spectral clustering is distracting, we are open to removing the spectral clustering experiments**.
> > >
> > >
> > > **Q4/Q5 (scale).**
> > >
> > > The divergence between Lloyd's and Hartigan's algorithms is observed at higher $n$, where spectral clustering is more challenging computationally. We are adding larger experiments with larger $n$; in these cases, as the theory hints, the same phenomena are observed at a higher-noise level or larger $k$.
> > >
> > > Examples:
> > >
> > > | Dataset |       N |     d |    K |  σ² |  Loss: Lloyd | Loss: Hartigan |     Loss: GT | NMI: Lloyd | NMI: Hartigan |
> > > |---------|--------:|------:|-----:|----:|-------------:|---------------:|-------------:|-----------:|--------------:|
> > > | GMM     |  20,000 |   200 |  200 |  10 |  40660822.50 |    39519691.33 |  39605549.59 |       0.47 |          0.89 |
> > > | GMM     |  50,000 |   200 |  500 |  10 | 102171559.71 |    99462022.62 |  99032389.59 |       0.34 |          0.65 |
> > > | GMM     | 100,000 |   200 |  500 |  10 | 203138814.12 |   198328048.66 | 198900437.99 |       0.47 |          0.87 |
> > > | GMM     | 100,000 |   200 | 1000 |  10 | 203477040.75 |   200230702.33 | 198023763.50 |       0.34 |          0.41 |
> > > | GMM     | 100,000 |   200 | 2000 |  10 | 200250501.96 |   195073485.22 | 195933242.10 |       0.43 |          0.50 |
> > > * Hartigan's loss can fall below the GT loss because the GT labelling is not the loss-optimal partition in this regime.
> > >
> > > In our 74pU response we report parallel large-scale results on the full 20 Newsgroups (n up to 18,246, K up to 20) and the Amazon Commerce Reviews corpus (n=1,500, d=10,000, K=50); Hartigan's algorithm achieves better loss, which translates to better NMI in most, but not all cases. Spectral clustering (with kNN graph due to scale) outperforms Lloyd's and Hartigan's algorithms in terms of NMI.

---

### Official Review · Reviewer_2Xrb · 2026-03-10

**Soundness:** 3
**Presentation:** 3
**Significance:** 3
**Originality:** 3
**Overall Recommendation:** 5
**Confidence:** 4

**Summary:**

The paper concerns the famous k-means problem and the classical Lloyds algorithm and Hartigans algorithm to solve it.
More precisely, the paper considers settings with a high number of dimensions and high noise.
The paper shows that the two algorithms show substantial differences in that setting:
While Lloyds algorithm returns the initial partition of the data points (even if the underlying clusters are trivially recoverable by other algorithms), Hartigans algorithm on the other hand does not have this weakness.
Hence, the paper contributes in explaining the empirical difficulties observed with k-means in the high dimensional setting.
Also, the paper provides an extensive experimental study verifying the theoretical results.

Since k-means is one of the most fundamental clustering problems, this paper clearly is of general interest for ICML.
The results are quite interesting and I found it quite surprising that these two classis algorithms show such a different behavior in that setting.
I only briefely checked the proofs in the appendix, but the overall ideas of the proofs seem sound to me; I also liked to provided intuition such as the one in the beginning of Section 3.
I also liked the extensive experimentes provided in the paper.
My main critique with the experiments is that only one value for the number of samples n was tested. It would have been nice to have at least one further value for n to see whether the results are stable with respect to n. But I see that the existing experiments are already quite extensive and the main focus of the paper is theory (so this is rather a suggestion than a critique).
Hence, overall I think this paper is a nice addition to the conference.

**Compliance With Llm Reviewing Policy:**

Affirmed.

**Final Justification:**

I would like to thank the authors once again for answering my questions. After reading the other reviews as well, I prefer to keep my score.

**Key Questions For Authors:**

Questions:
1) Is there any ituition for the shape of the figures shown in Figure 1 (and later), i.e. why is the upper left part always blue and the lower right part always yellow?
Related to that: l1463: Do you have any explaination for the different shapes of Figures 3 and 5? I would have exptected some correlation in the sense that if Lloyds algorith terminates after 1 iteration than the corresponding area in figure 3 should be yellow?

2) I am wondering how realistic the studied scenario is. Are there any real-world applications to which setting is useful?

3) Some of the formulas appearing in the theorems appear to be 'magic'; for example the computation of sigma in l247 c2; could you provide any intuition for formulas like this?

4) I am wondering why Theorem 3.9 is not symmetric: in l280: 0 < R_^l_j \leq R_\bar{j}^l \leq 1. So there is '0<' and '\leq 1'. Intuitively, I thought that this sould be symmetric, i. e., both sould be '<' or '\leq'. The same with l294 in that theorem. Can you explain why my intuition is wrong?

5) Dosen't figure 8 imply that both algorithms usually only require a constant number of iterations? If yes, this should be discussed.

6) The theoretic results are shown for the case of 2 clusters and in the experiments also more than 2 clusters are considered. Can you extend your theoretic results to the case of more than 2 clusters?


Comment on Lloyd vs Hartigans algorithm related to Telgarsky, M. and Vattani:
I think they also showed Hartigan can improve clusterings found by Lloyds algorithm but the reverse is not true. In some sense your result matches that statement since tou show that with high probability Lloyds algorithm terminates after one iteration. It could also be mentioned that both algorithms in the worst case require a superpolynomial number of iterations (see eg 'K-means Requires Exponentially Many Iterations Even in the Plane' and 'Complexity of Local Search for Euclidean Clustering Problem') but with smoothed complexity it becomes polynomial (see eg 'Smoothed Analysis of the k-Means Method', 'Worst-Case and Smoothed Analysis of the Hartigan-Wong Method for k-Means Clustering'). All these papers also show that solving k-means with these two algorithms is quite difficult but they do not show such a stark difference as the result in that paper.




Comments:
- in the beginning (probably section 1.1) there should be a reference to Appendix A
- For me, reading section 1.1 felt quite overwhelming since directly a notation-heavy theorem was presented. Here, more intuition prior to the theorem could help. Also, some notation in that theorem such as 'Gaussian mixture model' and 'isotropic noise covariance' is not defined (could be done in appendix A). Also, it could be helpful to already provide a reference to section 2 and say that k-means, Lloyds algorithm and Hartigans algorithm are defined there.
- l121 c1: couldn't you write x_i as a single normal distribution?
- l171 c1: I think that should be called a 'partition' too
- l190 c1: make clear that both partite sets have size close to n/2
- l239 c2: recall tau here
- l340 c2: refer to the appendix where NMI is defined
- l340ff c2: Initially I was wondering why you compare two different initialization for Lloyd and Hartigan and that this could be unfair. This was somewhat explained in the next paragraph. I recommend to swap them to avoid this confusion.
- l844: please explain this step a bit more (similar as it was done in the proof of Lemma 3.2)
- l854: '(1-R_j^l)' > '(1-R_j^l)\mu_l^*'
- l905: shouldn't this proof be analogous to the one for Lemma 3.1; you only need to swap the roles? If yes, this proof could then be dropped.
- l952: please provide more intuition why this case is interesting
- l996: this first derivative is written quite strangely, it should be written similar to the second one
- l1120: please mention where the requirement that each cluster contains at least two samples is used
- l1448: do you use the same initialization as for NMI?
- l1453: 'relatively consistent' For me, this only holds for NMI. I think k-means win rate should thus be discussed a bit more.
- l1478: why you only compare these two algorithms here?
- l1532: I think it should be 20 samples instead of 40
- l1677: at least add a reference to that 'standard improvements'
- figure 9: in the first 5 figures it is not clear where the solid line for the bound Lloyd is located since another line is on top of it

**Limitations:**

yes

**Strengths And Weaknesses:**

see summary and questions

---

> ### Author Rebuttal · Authors · 2026-03-30
>
> We are pleased that the reviewer found the results interesting and surprising, and that the reviewer found the proof sound and the experiments extensive. We would like to thank the reviewer for the helpful comments and suggestions. We will add additional experiments with different values of $n$. We have updated the paper with the reviewer's suggestions; due to OpenReview's space limitation, we were not able to answer each item as we would have liked.
>
> 1. Upper left (low-d, high-noise): the correct partition is not recoverable information-theoretically; all algorithms achieve low NMI (but better k-means loss than ground-truth). Lower right (high-d, low-noise): the problem is easy, and all algorithms succeed. The surprising finding is that Lloyd fails in the high-noise, high-dimensional regime where the problem remains easy — everything works except for the most popular algorithm. The results suggest this occurs in regimes far broader than those covered by the theorems. Regarding Figures 3 and 5: the correlation is the opposite of what one might expect — Lloyd terminating in one iteration means it is stuck at its initialization, not that it has converged correctly.
>
> 2. The theoretical results are conservative; experiments show the problem is relevant in a much broader range. Our motivation stems from cryo-electron microscopy (cryo-EM, 2017 Nobel Prize in Chemistry), which is characterized by high noise and high-dimensional data (images). The workhorse algorithm in cryo-EM is Expectation Maximization (a cousin of Lloyd's). Spectral clustering, SDPs, and most initialization methods do not generalize directly to cryo-EM, making the performance of iterative algorithms particularly important.
>
> 3. The most "magical" formulae, like l247, identify the noise level at which the sample is closer (in expectation) to its current cluster than to the alternative cluster and is therefore unlikely to move. This is similar to Remark 3.5, with an additional generalization that yields a uniform value for a broader family of partitions and eliminates parameters.
>
> 4. The subtle asymmetry covers edge cases: we consider a non-pure cluster $j$ and another cluster that may or may not be pure. Unless the partition is "correct," at least one cluster is not pure, though the other may be (hence the asymmetry). Concretely, $R^l_j > 0$ because $x_i$ is in cluster $C_j$ and belongs to class $l$.
>
> 5. The figure does seem to suggest this, but the analysis of the number of iterations requires additional techniques and may not reflect worst-case behavior (as implied by other work cited by the reviewer), so it remains beyond the scope of this paper. An analysis of the dynamics is an interesting candidate for future work.
>
> 6. Lloyd's result generalizes to $k>2$ with additional notation; this was deferred in favor of the Hartigan result. The Hartigan analysis for $k>2$ requires further investigation; the general idea holds, but is less clean. Our experiments indicate convincingly that this is not a special $k=2$ problem.
>
> 7. Telgarsky and Vattani: Yes, our results are consistent with theirs, but our results go further, for example, showing the stark divergence of the algorithms. We have not analyzed dynamics directly (only via fixed points), so we have not commented on iteration counts; we will add the suggested citations. The most interesting aspect is that the problem is very easy to solve in this regime, yet Lloyd's algorithm still fails.
>
> Selected notes on the reviewer's comments:
>
> * Section 1.1: A compromise for a quick, informal reference; we refer readers to the full results.
>
> * l121: $x_i$ and $x_{i'}$ from the same class share the same $\mu$, introducing dependence, so they cannot be written as completely i.i.d. samples.
>
> * l905: Small but important differences — in one case, the sample is included in computing the center; in the other, it is not.
>
> * l1120: The minimum of two samples is an artifact of the proof technique (we have a new theorem without it). Currently, this artifact is technically needed for Theorem 3.6.
>
> * l1448: Yes, same initialization. We will clarify this in the text.
>
> * l1453: A limitation of the win rate is its sensitivity to small misclassification: a single misclassified point may barely affect NMI but cause the experiment to receive a score of $-1$. We updated the text.
>
> * l1677: We follow Scikit-Learn for Lloyd; for Hartigan, we implement the local center update of Bottou and Bengio (1995).
>
> * Figure 9: No solid Lloyd line in the first 5 figures because the theorem does not apply there; we will clarify the caption.

---

> > ### Author Rebuttal · Reviewer_2Xrb · 2026-04-02
> >
> > Thanks for your answers which helped me a lot!
> >
> > I only have one additional comment on Question 5: I think you interpreted my question such that I asked for the general behavior in terms of number of iterations. But I was only referring to the number of required iterations in your instances where this number seems to be a small constant (<10). If this is true, I think this could be mentioned as this shows that the general PLS-hardness results are quite brittle in the sense that such instances do not occur in practice. (Again, in might be that the problem is not PLS-hard in the studied restriction, but even if this is the case your experiments show that this is no issue.)

---

> > > ### Author Response · Authors · 2026-04-07
> > >
> > > We apologize for the confusion. In Figure 8, the maximum number of iterations required for convergence was $64$ for Lloyd and $50$ for Hartigan. For Lloyd, the highest number of iterations was required at $K = 16$, $d = 100$, and $n = 10^4$, while the fewest iterations were needed when $d = 10{,}000$. In contrast, Hartigan required more iterations in high-dimensional settings ($d = 1{,}000$ and $d = 10{,}000$) and when $n = 10^4$, a trend that was consistent across all values of $K$. Following your suggestion, we will add a subplot to Figure 8 reporting the average number of iterations for each experimental configuration. We are happy to add the information if this is what you meant, and you find it useful.

---

### Official Review · Reviewer_qRfv · 2026-03-11

**Soundness:** 3
**Presentation:** 3
**Significance:** 3
**Originality:** 3
**Overall Recommendation:** 4
**Confidence:** 3

**Summary:**

This paper studies the behavior of Lloyd’s $k$-means algorithm in high-dimensional and high-noise settings. The authors show that the algorithm can exhibit catastrophic failure: with high probability, nearly every partition becomes a fixed point, causing Lloyd’s algorithm to simply return its initial partition even when the underlying clusters are easily recoverable. In contrast, the paper proves that Hartigan’s $k$-means algorithm does not suffer from this issue. These results highlight a fundamental difference between the two algorithms and provide a theoretical explanation for the empirical difficulties often observed with $k$-means in high-dimensional data.

**Compliance With Llm Reviewing Policy:**

Affirmed.

**Key Questions For Authors:**

1. What is the main technical novelty of the paper beyond formalizing the difference between Hartigan’s and Lloyd’s $k$-means?
2. There are many clustering algorithms designed for high-dimensional and high-noise settings. Could the authors explain the motivation for focusing the analysis specifically on Hartigan’s $k$-means algorithm?
3. Could the authors compare their results with existing clustering methods designed for high-dimensional or noisy data?

**Limitations:**

yes

**Strengths And Weaknesses:**

Weakness：
1. The paper compares Hartigan’s $k$-means algorithm with the standard Lloyd’s $k$-means algorithm and analyzes their differences. However, it seems that the work mainly reorganizes or formalizes existing observations rather than introducing substantially new technical ideas.
2. The organization of the paper could be improved. A large portion of the paper is devoted to introducing the $k$-means problem and reviewing existing techniques, while the methodology and technical analysis of the proposed work are relatively brief.
3.  The paper focuses on high-dimensional and high-noise settings, yet many clustering methods have been proposed to address these challenges. The paper does not discuss or compare with these existing approaches, which makes it difficult to assess the practical relevance of the results.

---

> ### Author Rebuttal · Authors · 2026-03-30
>
> W1/Q1. We are pleased that the reviewer recognizes the theoretical contributions in our paper.
> Indeed, our contribution is theoretical - proving the failure and extreme divergence in behaviours - not algorithmic. The paper identifies a regime where the (arguably) most popular and best-known clustering algorithm fails catastrophically on very easy problems.
> The closest results (Telgarsky and Vattani, 2010 and Slonim, Aharoni, and Crammer 2013) are substantially more limited; they prove a very narrow difference between the algorithms: for example, that the output of Hartigan's algorithm is also a fixed point of Lloyd's algorithm (Telgarsky and Vattani, 2010), and that a random partition is a fixed point of Lloyd's algorithm in an asymptotic regime where the SNR goes to zero (Slonim et al., 2013) - implicitly, a progressively hard problem - whereas we prove that even at a constant (low) SNR where the problem becomes progressively easier with dimensions, Lloyd's algorithm fails and we prove an extreme divergence between Lloyd's and Hartigan's algorithms.
> While it had been observed that Lloyd's algorithm experiences difficulties in high dimensions (often leading to the use of PCA heuristics in applications), the catastrophic extent of the failure and its occurrence on easy problems had not been explained.
>
> Concretely, our paper establishes several results that were not previously known or conjectured:
>
> (a) Lloyd's algorithm fails on easy problems. All approximately balanced partitions are Lloyd fixed points w.h.p. at constant SNR, in a regime where the problem becomes progressively easier with dimension. Prior work (Slonim et al.) proved this only asymptotically, where SNR -> 0 - an increasingly hard problem where failure is unsurprising.
>
> (b) From fixed-point landscape to algorithm output. Prior results characterize individual fixed points or set relationships between fixed-point sets, but leave open what the algorithms actually do - Lloyd might avoid bad fixed points from typical initializations, and Hartigan's smaller fixed-point set might still contain incorrect partitions. Because our result covers essentially all partitions, it directly characterizes Lloyd's output: the algorithm is trapped at initialization with no possible escape. This is a qualitative leap from "bad fixed points exist" to "the algorithm must fail."
>
> (c) Hartigan's algorithm guarantees. No incorrect partition is a Hartigan fixed point w.h.p. By the same logic as (b), this means Hartigan's algorithm must converge to the correct partition - not just "has fewer fixed points" (Telgarsky and Vattani) or "works in experiments" (Slonim et al.), but is provably correct in this regime.
>
> (d) Extreme divergence. Total failure (Lloyd) vs. total success (Hartigan) for structurally similar algorithms on the same easy problem. This complete phase transition between two variants of the same algorithm was not previously known or conjectured; prior work showed only local, quantitative differences.
>
> While this work does not introduce a new algorithm in itself, it serves as the basis for subsequent work on general algorithms and on algorithms for specific applications that build on these observations (in preparation). Given that we have already had to limit the discussion (for example, to the $k=2$ case) and move a lot of the theory to the appendix, it does not seem feasible to include additional results without compromising the quality and clarity of this paper.
>
> Q2/Q3/W3. We compared Lloyd's algorithm to Hartigan's algorithm, PCA + k-means, SDP, spectral clustering, and Ndaoud's algorithm (in the appendix), which represent the main families of alternative algorithms in general-purpose clustering; we also included multiple initialization methods (in the appendix). The list is broader than that in comparable papers such as Telgarsky 2010, Slonim 2013, and Ndaoud 2022. Subjectively, we have been particularly interested in Hartigan's algorithm because it is very similar in structure to Lloyd's algorithm, and appears to be as scalable, suggesting a direction for improving on Lloyd's algorithm in areas where it (and similar algorithms) is currently used. We acknowledge that these could have been two separate papers (our initial intention), but we find the contrast between the algorithms very interesting and note that the methods used in the two proofs are very similar.
> The reviewer may refer to specialized clustering methods for specialized high-dimensional applications; indeed, we are exploring heavily modified versions of these algorithms for cryo-electron microscopy applications as alternatives to algorithms used in that application (variants of Lloyd and Expectation Maximization), but the discussion of this specialized application is beyond the scope of this paper.

---

> > ### Author Rebuttal · Reviewer_qRfv · 2026-04-01
> >
> > Thank you for the authors’ response. I realize that the paper primarily focuses on the theoretical analysis of algorithmic behavior under specific regimes, rather than on traditional clustering modeling or application-oriented problems, which differs from my initial understanding. The authors’ response has addressed my concerns well, and I will reconsider raising my score.

---

### Official Review · Reviewer_74pU · 2026-03-12

**Soundness:** 2
**Presentation:** 3
**Significance:** 2
**Originality:** 3
**Overall Recommendation:** 4
**Confidence:** 4

**Summary:**

This paper studies Lloyd $k$-means algorithm and Hartigan's algorithm in high dimension, high noise setting under the dadasets generated by Gaussian mixture model. The main result (Theorem 1.1 in the paper) shows that when noise variance parameter $\sigma$ and the dimension $d$ are sufficiently large, every approximately balanced partition of the given dataset is a fixed point of Lloyd's iteration with high probability. As a sequence, the Lloyd's algorithm will stuck to its initialization regardless of the optimal cluster structure. In contrast, this paper points out that Hartigan's greedy single-sample strategy has no incorrect fixed partition from any initialization. This paper also provides experimental support across $k = 2, 5, 10$ and chosen baselines are $k$-means with PCA, SDP based method and spectral clustering, on both synthetic and real-world datasets.

**Compliance With Llm Reviewing Policy:**

Affirmed.

**Final Justification:**

The rebuttal solved most of my concerns. The remain concern is they don't analyse the case when $k>2$.

**Key Questions For Authors:**

1. **JL+Lloyd baseline (mentioned in Weakness part).** Since JL random projection can reduce the dimension to $O(\log n/\epsilon^2)$ while preserving the $k$-means objective, and $k$-means++ provide approximation ratio of $O(\log k)$, it seems that JL plus $k$-means++ could solve high dimensional $k$-means problem with theoretical guarantee. What is your advantage against to JL plus $k$-means++ approach?
2. **What is your PCA target dimension in your experimental part?** In the experimental part, PCA+Lloyd is a baseline. But the authors seem never mention the target dimension using PCA to reduce to.
3. **Why does Hartigan perform better than $k$-mean++ in theory?** This is a curiousity question. In Figure 1, Hartigan's algorithm works better than $k$-means++. But $k$-means++ could provide theoretical guarantee, i.e., approximatio ratio of $O(\log k)$, while Hartigan could not. It would be better if the authors analysis why Hartigan perform better compared to $k$-means++.

**Limitations:**

yes.

**Strengths And Weaknesses:**

## Strengths

1. The contrast between Lloyd and Hartigan is clear and informative. The authors provide clear setting of high dimension and high norise that leads Lloyd fail and Hartigan succeed. Further more, they provide theoretical analysis behind this phenomenon.

## Weakness

1. **(Major concern) Missing discussion of JL-based dimensionality reduction.** The paper does not discuss the Johnson-Lindenstrauss approach, where random projection to $O(\log n / \epsilon^2)$ dimensions preserves the $k$-means objective with high probability. Therefore, the high dimension problem could be solved if $d$ is polynomial of $n$. The paper includes PCA+Lloyd but not JL+Lloyd, and does not discuss how its self-influence mechanism relates to why dimensionality reduction resolves Lloyd's failure.
2.  **The datasets dimensions are too low in experiments.** The dimension thresholds ($d\ge n^3$ for Lloyd and $d\ge n \sigma^4$ for Hartigan) are much higher than experimental datasets dimensions. Some discussion of the gap between the theoretical and empirical thresholds would be helpful.
3.  **Limited discussion on more complicated cases.** The theoretical part of this paper only discusses the case when $k=2$ and under GMM. However, in real world cases, considering larger $k$ and more complex dataset distribution is commonly required.
4.  **Limited real-world datasets.** The chosen real-world datasets in the experimental part are too small. I suggest the authors add more datasets with larger size.

---

> ### Author Rebuttal · Authors · 2026-03-30
>
> W1/Q1. JL dimensionality reduction. We are pleased that the reviewer recognizes the paper's theoretical contribution and clarity. The reviewer correctly identifies dimensionality reduction as one direction for follow-up work on this topic; unfortunately, we are not able to include all these equally important directions. The reviewer observed the counter-intuitive fact that our theory hints at, and the experiments confirm: there are certain regimes where less information (randomly projected data) actually makes Lloyd's algorithm work better than it would work otherwise.
> A random projection behaves the same way as a lower-dimensional instance of the problem, so the plots for random projections are essentially identical to the plots we have for different dimensions (observed theoretically and numerically but currently excluded from the paper).
> Indeed, there are some cases where a careful choice of dimension can improve Lloyd's performance, but it does not appear to compete with Hartigan's algorithm's performance (and the choice depends on the noise).
> More aggressive dimensionality reduction decreases the NMI (a very aggressive reduction corrupts the data to the point where the correct clusters cannot be recovered by any algorithm).
> PCA behaves differently. Intuitively, because PCA optimally preserves pairwise distances (in the appropriate sense) given the data, PCA can indeed help and is used as a heuristic in applications, but it appears to be very suboptimal, as demonstrated in the experiments.
>
>
> W2. Gap between theoretical and empirical thresholds. Our bounds are sufficient conditions based on concentration inequalities and union bounds over all partitions; they certify the *existence* of the phenomenon, not sharp thresholds. The experiments show that the phenomenon manifests at much lower dimensions and noise levels, suggesting that the true thresholds are significantly tighter. Closing this gap is a direction for future work.
>
>
>
> W3. $k=2$ restriction. The results for Lloyd's algorithm generalize to $k>2$ with additional notation and accounting; this was deferred in favor of presenting the Hartigan result. The Hartigan analysis for $k>2$ is more nuanced and requires further investigation, though the general idea holds. Our experiments with $k=5$ and $k=10$ indicate convincingly that this is not a special $k=2$ phenomenon.
>
>
> W4. Limited real-world datasets. The regime studied is specifically high-dimensional, low-sample ($d \gg n$), which arises naturally in gene expression, single-cell RNA-seq, text analysis, and medical imaging. Our datasets (Olivetti: $d=4096$, $n=400$; 20NG: $d=5000$, $n=200$--$1000$) are representative of this regime. We will add a larger experiment with 100 samples per class (preliminary results are very similar).
>
>
> Q2. PCA target dimension in the experiments was 4 for k=2, 5 for k=5, and 10 for k=10 (updated in the paper).
>
>
> Q3: Hartigan vs k-means++. The k-means++ guarantee (Arthur and Vassilvitskii, 2007) states $E[\phi] \leq 8(\ln k + 2)\,\phi_{\mathrm{OPT}}$. This is a bound on the *k-means loss*, not on cluster recovery. In the high-noise regime, the objective is dominated by noise, and this guarantee becomes vacuous for recovery. In fact, in this regime, the guarantee is not very usful even for the k-means loss: using the pairwise identity $W(C) = \frac{1}{2} \sum_{k} \sum_{C(i)=k} \sum_{C(i')=k} \|x_i - x_{i'}\|^2$ (Hastie et al., 2009, Eq.~14.31), a rough estimate gives $E[\phi_{\mathrm{OPT}}] \propto 2d\sigma^2$ and a rough (over)estimate $E[\phi_{\mathrm{BAD}}] \propto 2d(\sigma^2 + \tau^2)$. When $\sigma \gg \tau$, the ratio $\phi_{\mathrm{BAD}}/\phi_{\mathrm{OPT}} \approx 1 + \tau^2/\sigma^2 \ll 8(\ln k + 2)$, so *every* partition---including a random one---trivially satisfies the k-means++ guarantee while being a very bad partition. Hartigan's advantage is about *partition quality* with respect to ground truth (which also implies an advantage in the loss in this regime): its weighted distance avoids the self-influence effect that traps Lloyd, allowing it to distinguish the correct partition in a regime where all the relative differences in k-means costs are small (but differences do exist in absolute terms).

---

> > ### Author Rebuttal · Reviewer_74pU · 2026-04-03
> >
> > Thank you for the detailed explanation.
> >
> > There are several additional questions I would like to ask the authors.
> >
> > 1. In your rebuttal to qRfV (https://openreview.net/forum?id=4CwO8At8Hw&noteId=W289QrrqZE), you mention that you proved that when the SNR is a small constant, it is still possible to recover the ground truth. Here, SNR is defined as
> > $\text{SNR} = ||\mu_1^* - \mu_2^*||^2 / \sigma^2$,
> > right?
> > However, according to your Theorem 1.1, the dimension $d$ must be sufficiently large, i.e., $d \ge n \sigma^4$. Since SNR should depend on $d$. Why can SNR remain a constant when $d$ is so large? It seems to me that your setting implicitly assumes a high SNR, which may make the problem easier.
> >
> > 2. I share the same concern raised by jcJT (https://openreview.net/forum?id=4CwO8At8Hw&noteId=NrddwQfKAP) regarding the construction of the spectral clustering step and the choice of parameters.
> >
> > 3. Could you provide some intuition for why your analysis requires such a large $d$? From an information-theoretic perspective, increasing $d$ indeed increases the SNR, making the problem easier. But from a clustering perspective, lower-dimensional settings are typically considered easier. Why does your method rely on high dimensionality?

---

> > > ### Author Response · Authors · 2026-04-07
> > >
> > > Q1. There are several definitions of SNR, which is why we avoided the ambiguous term in the paper. In the rebuttal we referred to $\tau^2 / \sigma^2$ or equivalently $d \tau^2 / d \sigma^2$ (which is approximately $\|\mu_1^* - \mu_2^* \|^2 / 2 d \sigma^2 $).
> > > This definition is natural from a signal-processing perspective, but it is not unique (a different definition can be found in Ndaoud 2022, for example). We should have stated the definition explicitly.
> > > Our noise model has variance $\sigma^2$ in each dimension, as explicitly defined in the paper (which is worth noting because some papers use different scaling conventions).
> > > Under our definition the ratio is dimension-independent. The theory assumes low SNR by our definition; in high dimensions the problem is statistically easy and most algorithms succeed — the surprise is that Lloyd's does not. The same regime is high-SNR under the reviewer's definition (with $\sigma$ as defined in the paper), with the same surprise.
> > >
> > >
> > > Q2. In the paper, we used scikit-learn's default parameters for `SpectralClustering`, other than `n_init`, which was 500.
> > > The spectral clustering experiments are presented for completeness.
> > > If the reviewers find that including spectral clustering is distracting, we are open to removing the spectral clustering experiments.
> > >
> > > We are adding larger-scale experiments. Examples:
> > >
> > > Real datasets:
> > >
> > > | Dataset |       N |      d |   K | Loss: Lloyd | Loss: Hartigan | Loss: Spect-NN | Loss: GT | NMI: Lloyd | NMI: Hartigan | NMI: Spect-NN |
> > > |---------|--------:|-------:|----:|------------:|---------------:|---------------:|---------:|-----------:|--------------:|--------------:|
> > > |  20NG-A |   1,727 | 10,000 |   2 |     1691.51 |  1691.47 |  1692.02 |  1692.09 |   0.57 |   0.58 |  0.75 |
> > > |  20NG-B |   4,590 | 10,000 |   5 |     4484.76 |  4484.23 |  4497.14 |  4494.14 |   0.45 |   0.46 |  0.49 |
> > > |  20NG-C |   9,367 | 10,000 |  10 |     9104.91 |  9104.55 |  9104.82 |  9279.26 |   0.41 |   0.41 |  0.43 |
> > > |  20NG-D |  18,246 | 10,000 |  20 |    17577.11 | 17574.17 |  17662.30 | 17777.73 |  0.35 |   0.34 |  0.43 |
> > > |  Amazon |   1,500 | 10,000 |  50 |     1386.77 |  1356.26 |  1370.36 |  1384.92 |   0.42 |   0.52 |  0.55 |
> > >
> > > * In these larger experiments, we had to use spectral clustering with kNN preprocessing and fewer inits. Details omitted here due to character limitation.
> > >
> > > * Full 20NG groups, 10k features.
> > >
> > > * Amazon Commerce Reviews dataset (Liu, 2011), which contains $1500$ samples, $10000$ features, and reviews by $K=50$ distinct users. Preprocessing: standardizing each feature using scikit-learn's StandardScaler, followed by normalization of each sample to unit norm.
> > >
> > >
> > > Synthetic examples:
> > >
> > > | Dataset |       N |     d |    K |  σ² |  Loss: Lloyd | Loss: Hartigan |     Loss: GT | NMI: Lloyd | NMI: Hartigan |
> > > |---------|--------:|------:|-----:|----:|-------------:|---------------:|-------------:|-----------:|--------------:|
> > > | GMM     |  20,000 |   200 |  200 |  10 |  40660822.50 |    39519691.33 |  39605549.59 | 0.47 | 0.89 |
> > > | GMM     |  50,000 |   200 |  500 |  10 | 102171559.71 |    99462022.62 |  99032389.59 | 0.34 | 0.65 |
> > > | GMM     | 100,000 |   200 |  500 |  10 | 203138814.12 |   198328048.66 | 198900437.99 | 0.47 | 0.87 |
> > > | GMM     | 100,000 |   200 | 1000 |  10 | 203477040.75 |   200230702.33 | 198023763.50 | 0.34 | 0.41 |
> > > | GMM     | 100,000 |   200 | 2000 |  10 | 200250501.96 |   195073485.22 | 195933242.10 | 0.43 | 0.50 |
> > > * Hartigan's loss can fall below the GT loss because the GT labelling is not the loss-optimal partition in this regime.
> > >
> > >
> > > Q3. For Lloyd's algorithm, this is exactly the surprising fact: the problem becomes easier, but Lloyd's algorithm fails.
> > >
> > > Part of the reason clustering is typically considered easier in low dimensions is that Lloyd's popular algorithm fails in high dimensions.
> > >
> > > The proof technique uses the expected distance of samples to their current cluster center vs. alternative cluster centers. In the high-noise regime, the expected difference between these distances is small. In low dimensions, the actual distances have high variance around their expectations, so despite the small expected difference, samples can still "hop" to the correct cluster with non-negligible probability. In high dimensions, the distances concentrate tightly around their means. This concentration means the small expected advantage of the incorrect cluster becomes a *deterministic* barrier: every sample is trapped near its current assignment with high probability, and Lloyd's algorithm cannot move any point. This is the self-influence effect — each sample contributes to its own cluster center, pulling the center toward itself and creating a spurious attractive force.

---

### Decision · Program_Chairs · 2026-04-30

**Decision:**

Accept (regular)

**Comment:**

This paper presents a theoretical and empirical comparison of Lloyd’s and Hartigan’s $k$-means algorithms in high-dimensional, high-noise regimes, demonstrating that Lloyd’s algorithm is prone to catastrophic failure while Hartigan’s remains robust.

Strengths
- Offers an insightful comparison between two foundational clustering algorithms.
- The theoretical analysis is clear and well-written.

Weaknesses
- The theoretical analysis is limited for general $k$.
- Experimental results have room for improvement, though reviewers noted this is acceptable given the paper's primary focus on theory.

Decision
Accept